# ROBUST NAS UNDER ADVERSARIAL TRAINING: BENCHMARK, THEORY, AND BEYOND

**Yongtao Wu**
LIONS, EPFL
`yongtao.wu@epfl.ch`

**Fanghui Liu**[*]
University of Warwick
`fanghui.liu@warwick.ac.uk`

**Carl-Johann Simon-Gabriel**[†]
Mirelo AI
`cjsg@mirelo.ai`

**Grigorios G Chrysos**[*]
University of Wisconsin-Madison
`chrysos@wisc.edu`

**Volkan Cevher**
LIONS, EPFL
`volkan.cevher@epfl.ch`

## ABSTRACT

Recent developments in neural architecture search (NAS) emphasize the significance of considering robust architectures against malicious data. However, there is a notable absence of benchmark evaluations and theoretical guarantees for searching these robust architectures, especially when adversarial training is considered. In this work, we aim to address these two challenges, making twofold contributions. First, we release a comprehensive data set that encompasses both clean accuracy and robust accuracy for a vast array of *adversarially trained* networks from the NAS-Bench-201 search space on image datasets. Then, leveraging the neural tangent kernel (NTK) tool from deep learning theory, we establish a generalization theory for searching architecture in terms of clean accuracy and robust accuracy under multi-objective adversarial training. We firmly believe that our benchmark and theoretical insights will significantly benefit the NAS community through reliable reproducibility, efficient assessment, and theoretical foundation, particularly in the pursuit of robust architectures. [1]

## 1 INTRODUCTION

The success of deep learning can be partly attributed to the expert-designed architectures, e.g., ResNet (He et al., 2016), Vision Transformer (Dosovitskiy et al., 2021), and GPT (Brown et al., 2020), which spurred research in the field of Neural Architecture Search (NAS) (Baker et al., 2017; Zoph & Le, 2017; Suganuma et al., 2017; Real et al., 2019; Liu et al., 2019). The target of NAS is to automate the process of discovering powerful network architectures from a search space using well-designed algorithms. This automated approach holds the promise of unveiling highly effective architectures with promising performance that might have been overlooked in manually crafted designs (Zela et al., 2020; Wang et al., 2021; Ye et al., 2022).

Nevertheless, merely pursuing architectures with high clean accuracy, as the primary target of NAS, is insufficient due to the vulnerability of neural networks to adversarial attacks, where even small perturbations can have a detrimental impact on performance (Szegedy et al., 2013). Consequently, there is a growing interest in exploring robust architectures through the lens of NAS (Guo et al., 2020; Mok et al., 2021; Hosseini et al., 2021; Huang et al., 2022). These approaches aim to discover architectures that exhibit both high performance and robustness under existing adversarial training strategies (Goodfellow et al., 2015; Madry et al., 2018).

When studying the topic of robust neural architecture search, we find that there are some remaining challenges unsolved both empirically and theoretically. From a practical point of view, the accuracy of architecture found by NAS algorithms can be directly evaluated using existing benchmarks through a look-up approach, which significantly facilitates the evolution of the NAS community (Ying et al.,

---

[*]Partially done at LIONS, EPFL.
[†]Partially done at AWS Lablets.
[1]Our project page is available at `https://tt2408.github.io/nasrobbench201hp`.

2019; Duan et al., 2021; Dong & Yang, 2020; Hirose et al., 2021; Zela et al., 2022; Bansal et al., 2022; Jung et al., 2023). However, these benchmarks are constructed under standard training, leaving the adversarially trained benchmark missing. This requirement is urgent within the NAS community because 1) it would accelerate and standardize the process of delivering robust NAS algorithms (Guo et al., 2020; Mok et al., 2021), since our benchmark can be used as a look-up table; 2) the ranking of robust architectures shows some inconsistency based on whether adversarial training is involved, as we show in Table 1. Therefore, one main target of this work is to build a NAS benchmark tailored for adversarial training, which would be beneficial to reliable reproducibility and efficient assessment.

Beyond the need for a benchmark, the theoretical guarantees for architectures obtained through NAS under adversarial training remain elusive. Prior literature (Oymak et al., 2021; Zhu et al., 2022a) establishes the generalization guarantee of the searched architecture under standard training based on neural tangent kernel (NTK) (Jacot et al., 2018). However, when involving adversarial training, it is unclear *how to derive NTK under the multi-objective objective with standard training and adversarial training. Which NTK(s) can be employed to connect and further impact clean accuracy and robust accuracy?* These questions pose an intriguing theoretical challenge to the community as well.

Based on our discussions above, we summarize the contributions and insights as follows:

- We release the first adversarially trained NAS benchmark called *NAS-RobBench-201*. This benchmark evaluates the robust performance of architectures within the commonly used NAS-Bench-201 (Dong & Yang, 2020) search space in NAS community, which includes 6,466 unrepeated network architectures. 107k GPU hours are required to build the benchmark on three datasets (CIFAR-10/100, ImageNet-16-120) under adversarial training. The entire results of all architectures are included in the supplementary material and will be public to foster the development of NAS algorithms for robust architecture.

- We provide a comprehensive assessment of the benchmark, e.g., the performance of different NAS algorithms, the analysis of selected nodes from top architectures, and the correlation between clean accuracy and robust accuracy. We also test the correlation between various NTK metrics and accuracies, which demonstrates the utility of NTK(s) for architecture search.

- We consider a general theoretical analysis framework for the searched architecture under a multi-objective setting, including standard training and adversarial training, from a broad search space, *cf.* Theorem 1. Our framework allows for fully-connected neural networks (FCNNs) and convolutional neural networks (CNNs) with activation function search, skip connection search, and filter size search (in CNNs). Our results indicate that clean accuracy is determined by a joint NTK that includes partly a clean NTK and a robust NTK while the robust accuracy is always influenced by a joint NTK with the robust NTK and its "twice" perturbed version[2]. For a complete theoretical analysis, we provide the estimation of the lower bound of the minimum eigenvalue of such joint NTK, which significantly affects the (robust) generalization performance with guarantees.

By addressing these empirical and theoretical challenges, our benchmark comprehensively evaluates robust architectures under adversarial training for practitioners. Simultaneously, our theory provides a solid foundation for designing robust NAS algorithms. Overall, this work aims to contribute to the advancement of NAS, particularly in the realm of robust architecture search, as well as broader architecture design.

## 2 RELATED WORK

In this section, we present a brief summary of the related literature, while a comprehensive overview and discussion of our contributions with respect to prior work are deferred to Appendix B.

**Adversarial example and defense.** Since the seminal work of Szegedy et al. (2013) illustrated that neural networks are vulnerable to inputs with small perturbations, several approaches have emerged to defend against such attacks (Goodfellow et al., 2015; Madry et al., 2018; Xu et al., 2018; Xie et al., 2019; Zhang et al., 2019; Xiao et al., 2020). Adversarial training (Goodfellow et al., 2015; Madry et al., 2018) is one of the most effective defense mechanisms that minimizes the empirical training loss based

---

[2]We call "clean NTK" the standard NTK w.r.t clean input while "robust NTK" refers to the NTK with adversarial input. The "twice" perturbed version refers to the input where adversarial noise is added twice.

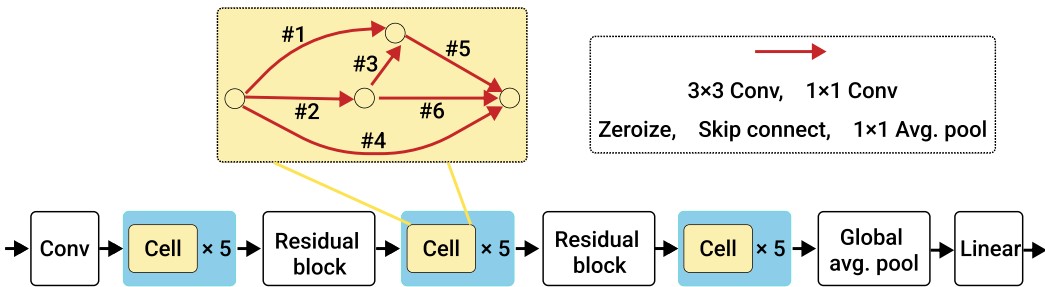

Figure 1: Visualization of the NAS-Bench-201 search space. **Top left**: A neural cell with 4 nodes and 6 edges. **Top right**: 5 predefined operations that can be selected as the edge in the cell. **Bottom**: Macro structure of each candidate architecture in the benchmark.

on the adversarial data, which is obtained by solving a maximum problem. Goodfellow et al. (2015) use Fast Gradient Sign Method (FGSM), a one-step method, to generate the adversarial data. However, FGSM relies on the linearization of loss around data points and the resulting model is still vulnerable to other more sophisticated adversaries. Multi-step methods are subsequently proposed to further improve the robustness, e.g., multi-step FGSM (Kurakin et al., 2018), multi-step PGD (Madry et al., 2018). To mitigate the effect of hyper-parameters in PGD and the overestimation of robustness (Athalye et al., 2018), Croce & Hein (2020b) propose two variants of parameter-free PGD attack, namely, $APGD_{CE}$ and $APGD_{DLR}$, where CE stands for cross-entropy loss and DLR indicates difference of logits ratio (DLR) loss. Both $APGD_{CE}$ and $APGD_{DLR}$ attacks dynamically adapt the step-size of PGD based on the loss at each step. Furthermore, to enhance the diversity of robust evaluation, Croce & Hein (2020b) introduce Auto-attack, which is the integration of $APGD_{CE}$, $APGD_{DLR}$, Adaptive Boundary Attack (FAB) (Croce & Hein, 2020a), and black-box Square Attack (Andriushchenko et al., 2020). Other methods of generating adversarial examples include L-BFGS (Szegedy et al., 2013), C&W attack (Carlini & Wagner, 2017).

**NAS and benchmarks.** Over the years, significant strides have been made towards developing NAS algorithms from various perspectives, e.g., differentiable search with weight sharing (Liu et al., 2019; Zela et al., 2020; Ye et al., 2022), NTK-based methods (Chen et al., 2021; Xu et al., 2021; Mok et al., 2022; Zhu et al., 2022a). Most recent work on NAS for robust architecture belongs to the first category (Guo et al., 2020; Mok et al., 2021). Along with the evolution of NAS algorithms, the development of NAS benchmarks is also important for an efficient and fair comparison. Regarding clean accuracy, several tabular benchmarks, e.g., NAS-Bench-101 (Ying et al., 2019), TransNAS-Bench-101 (Duan et al., 2021), NAS-Bench-201 (Dong & Yang, 2020) and NAS-Bench-301 (Zela et al., 2022) have been proposed that include the performance under standard training on image classification tasks. More recently, Jung et al. (2023) extend NAS-Bench-201 towards robustness by evaluating the performance in NAS-Bench-201 space in terms of various attacks. However, all of these benchmarks are under standard training, which motivates us to release the first NAS benchmark that contains the performance under adversarial training.

**NTK.** Originally proposed by Jacot et al. (2018), NTK connects the training dynamics of over-parameterized neural networks to kernel regression. In theory, NTK provides a tractable analysis for several phenomena in deep learning. For example, the generalization guarantee of over-parameterized FCNN under standard training has been established in Cao & Gu (2019); Zhu et al. (2022a). In this work, we extend the scope of NTK-based generalization guarantee to multi-objective training, which covers the case of both standard training and adversarial training.

## 3 NAS-ROBBENCH-201

In this section, we first describe the construction of the benchmark, including details on search space, datasets, training setup, and evaluation metrics. Next, we present a comprehensive statistical analysis of the built benchmark. More details can be found in the supplementary.

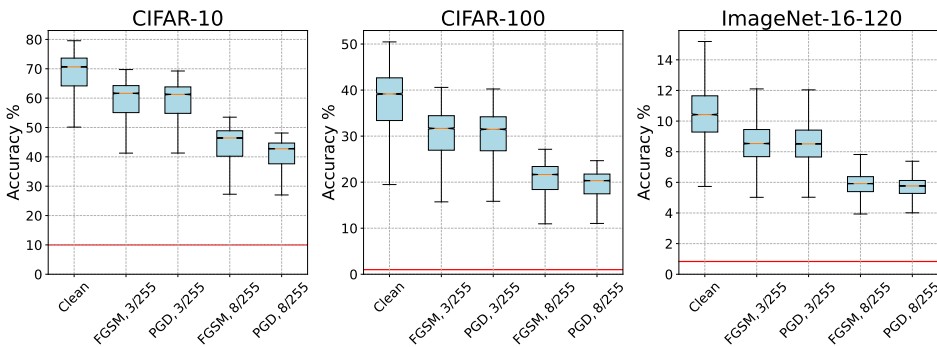

Figure 2: Boxplots for both clean and robust accuracy of all 6466 non-isomorphic architectures in the considered search space. Red line indicates the accuracy of a random guess.

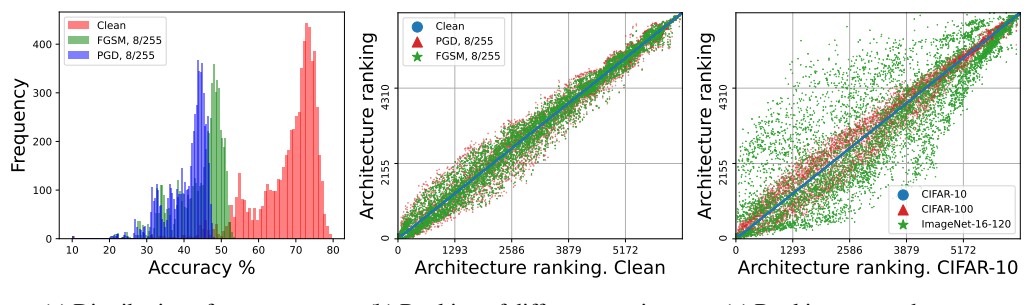

(a) Distribution of accuracy.   (b) Ranking of different metrics.   (c) Ranking across datasets.

Figure 3: (a) Distribution of accuracy on CIFAR-10. The peak in the distribution of clean accuracy is much more notable than that of FGSM and PGD. (b) The architecture ranking on CIFAR-10 sorted by robust metric and clean metric correlate well for lower ranking (see larger x-axis) but there still exists a difference for higher ranking. Both (a) and (b) motivate the NAS for robust architecture in terms of robust accuracy instead of clean accuracy. (c) Architecture ranking of average robust accuracy on 3 datasets, sorted by the average robust accuracy on CIFAR-10. The architectures present similar performance across different datasets, which motivates transferable NAS under adversarial training.

Table 1: Spearman coefficient between various accuracies on CIFAR-10 on NAS-RobBench-201 and the benchmark of Jung et al. (2023). Specifically, the **first three columns/rows** (without *) indicate the clean, PGD (8/255), and FGSM (8/255) accuracies in NAS-RobBench-201, while the **last three columns/rows** (with *) indicate the corresponding accuracies in Jung et al. (2023).

|        | Clean | PGD   | FGSM  | Clean* | PGD*  | FGSM* |
|--------|-------|-------|-------|--------|-------|-------|
| Clean  | 1.000 | 0.985 | 0.989 | 0.977  | 0.313 | 0.898 |
| PGD    | 0.985 | 1.000 | 0.998 | 0.970  | 0.382 | 0.937 |
| FGSM   | 0.989 | 0.998 | 1.000 | 0.974  | 0.371 | 0.931 |
| Clean* | 0.977 | 0.970 | 0.974 | 1.000  | 0.322 | 0.891 |
| PGD*   | 0.313 | 0.382 | 0.371 | 0.322  | 1.000 | 0.487 |
| FGSM*  | 0.898 | 0.937 | 0.931 | 0.891  | 0.487 | 1.000 |

## 3.1 BENCHMARK CONSTRUCTION

**Search space.** We construct our benchmark based on a commonly-used cell-based search space in the NAS community: NAS-Bench-201 (Dong & Yang, 2020), which consists of 6 edges and 5 operators as depicted in Figure 1. Each edge can be selected from the following operator: $\{3 \times 3 \text{ Convolution}, 1 \times 1 \text{ Convolution}, \text{Zeroize}, \text{Skip connect}, 1 \times 1 \text{ Average pooling}\}$, which results in $5^6 = 15625$ architectures while only 6466 architectures are non-isomorphic. Therefore, we only train and evaluate 6466 architectures.

**Dataset.** We evaluate each network architecture on (a) CIFAR-10 (Krizhevsky et al., 2009), (b) CIFAR-100 (Krizhevsky et al., 2009), and (c) ImageNet-16-120 (Chrabaszcz et al., 2017). Both CIFAR-10

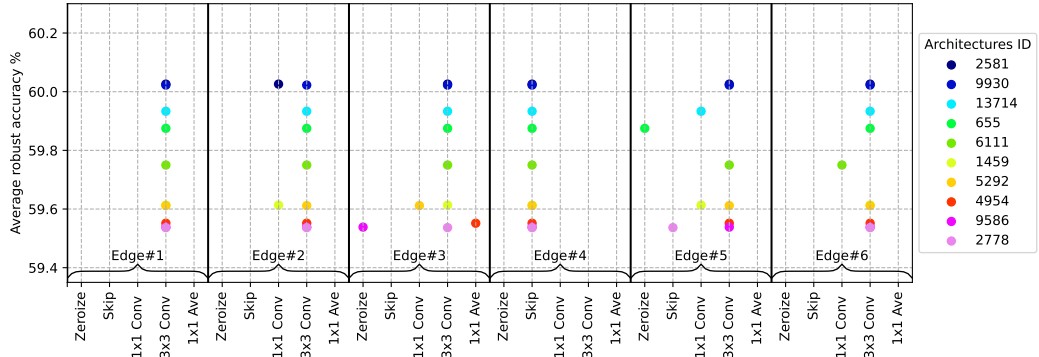

Figure 4: The operators of each edge in the top-10 architectures (average robust accuracy) on NAS-RobBench-201. The definition of edge number ($\#1 \sim \#6$) and operators are illustrated in Figure 1.

and CIFAR-100 contain $60,000$ RGB images of resolution $32 \times 32$, where $50,000$ images are in the training set and $10,000$ images are in the test set. Each image in CIFAR-10 is annotated with $1$ out of the $10$ categories while there are $100$ categories in CIFAR-100. ImageNet-16-120 is a variant of ImageNet that down-samples the images to resolution $16 \times 16$ and selects images within the classes $[1,120]$. In total, there are $151,700$ images in the training set and $6,000$ images in the test set. Data augmentation is applied for the training set. On CIFAR-10 and CIFAR-100, we apply a random crop for the original patch with 4 pixels padding on the border, a random flip with a probability of $0.5$, and standard normalization. On ImageNet-16-120, we apply similar augmentations via random crop with 2 pixels padding. The data is normalized before the attack, following the setup in Zhang et al. (2019); Rice et al. (2020).

**Training procedure.** We adopt a standard adversarial training setup via mini-batch SGD with a step-size of $0.05$, momentum of $0.9$, weight decay of $10^{-4}$, and batch size of $256$. We train each network for $50$ epochs where one-cycle step-size schedule with maximum step-size $0.1$ (Smith & Topin, 2019), which is proposed for faster convergence. Each run is repeated for 3 different seeds. Regarding the adversarial attack during training, we follow a common setting, i.e., 7 steps of projected gradient descent (PGD) with step-size $2/255$ and perturbation radius $\rho = 8/255$ (Madry et al., 2018). We apply the same setup for all of the aforementioned datasets. In total, we adversarially train and evaluate $6466 \times 3 \times 3 \approx 58k$ *architectures* by a number of NVIDIA T4 Tensor Core GPUs. One seed for one dataset consumes approximately 34 hours on 350 GPUs. Consequently, 3 datasets and 3 seeds take around $34 \times 350 \times 3 \times 3 \approx 107k$ *GPU hours*.

**Evaluation metrics.** We evaluate the clean accuracy and robust accuracy of each architecture after training. Specifically, we measure the robust accuracy based on fast gradient sign method (FGSM) (Goodfellow et al., 2015) and PGD attack with $\ell_\infty$ constraint attack with perturbation radius $\rho \in \{3/255, 8/255\}$. For PGD attack, we adopt 20 steps with step-size $2.5 \times \rho/20$. Additionally, we evaluate each architecture under AutoAttack with perturbation radius $\rho = 8/255$.

### 3.2 STATISTICS OF THE BENCHMARK

**Overall preview of the benchmark.** In Figure 2, we show the boxplots of the clean accuracy and robust accuracy of all architectures in the search space, respectively. Notice that there exists a non-negligible gap between the performance of different architectures, e.g., ranging from $40\% \sim 70\%$ accuracy under FGSM attack on CIFAR-10. Therefore, designing the architecture holds immense significance given the wide spectrum of achievable accuracy values. In Figure 3a, we plot the distribution of the clean accuracy and robust accuracy on CIFAR-10 in the proposed NAS-RobBench-201. We observe that the distribution of FGSM accuracy and PGD accuracy is similar while the peak in the distribution of clean accuracy is much more notable than that of FGSM and PGD. Overall, the architecture ranking sorted by robust metric and clean metric correlate well, as shown in Figure 3b.

**Effect of operators selection on robustness.** To see the impact of operators in robust architecture design, in Figure 4, we present the selected operators at each edge of the top-10 architectures in

Table 2: Result of different NAS algorithms on the proposed NAS-RobBench-201. "Optimal" refers to the architecture with the highest average robust accuracy among the benchmark. "Attack scheme" in the first column indicates how the accuracy is measured during the searching phase of these baseline methods.

| Attack scheme | Method | Clean | FGSM (3/255) | PGD (3/255) | FGSM (8/255) | PGD (8/255) |
|---|---|---|---|---|---|---|
| - | Optimal | 0.794 | 0.698 | 0.692 | 0.537 | 0.482 |
| Clean | Regularized Evolution | 0.791 | 0.693 | 0.688 | 0.530 | 0.476 |
| | Random Search | 0.779 | 0.682 | 0.676 | 0.520 | 0.470 |
| | Local Search | 0.796 | 0.697 | 0.692 | 0.533 | 0.478 |
| FGSM (8/255) | Regularized Evolution | 0.790 | 0.693 | 0.688 | 0.532 | 0.478 |
| | Random Search | 0.774 | 0.679 | 0.674 | 0.521 | 0.471 |
| | Local Search | 0.794 | 0.695 | 0.689 | 0.535 | 0.481 |
| PGD (8/255) | Regularized Evolution | 0.789 | 0.692 | 0.686 | 0.531 | 0.478 |
| | Random Search | 0.771 | 0.676 | 0.671 | 0.520 | 0.471 |
| | Local Search | 0.794 | 0.695 | 0.689 | 0.535 | 0.481 |

CIFAR-10 dataset. The top-10 criterion is the average robust accuracy, which is the mean of all robust metrics mentioned in Section 3.1. Overall, these top architectures have similar operator selections at each edge. We can see there exists a frequently selected operator for each edge. For instance, the $3 \times 3$ convolution operation appears to be the best choice for the majority of edges, except for edge 4, where the skip-connection operation demonstrates its optimality.

**Architecture ranking across different datasets.** Figure 3c depicts the architecture ranking based on the average robust accuracy across CIFAR-10/100 and ImageNet-16. The result reveals a high correlation across various datasets, thereby inspiring the exploration of searching on a smaller dataset to find a robust architecture for larger datasets.

**Comparison with the existing benchmark (Jung et al., 2023).** The closest benchmark to ours is built by Jung et al. (2023), where the robust evaluation of each architecture under standard training is given. In Table 1, we present the Spearman coefficient among different accuracies. The observation underscores the significance of adversarial training within our benchmark. Specifically, it suggests that employing Jung et al. (2023) to identify a resilient architecture may not guarantee its robustness under adversarial training. Moreover, in Figure 8 of the appendix, we can see a notable difference between these two benchmarks in terms of top architectures id and selected nodes.

**NAS algorithms on the benchmark.** Let us illustrate how to use the proposed NAS-RobBench-201 for NAS algorithms. As an example, we test several NAS algorithms on CIFAR-10. The search-based NAS algorithms include regularized evolution (Real et al., 2019), local search (White et al., 2021) and random search (Li & Talwalkar, 2020). We run each algorithm 100 times with 150 queries and report the mean accuracy in Table 2. We can observe that by searching the robust metrics, e.g., FGSM and PGD, these methods are able to find a more robust architecture than using clean accuracy metrics. Additionally, local search performs better than other methods. Evaluation results on more NAS approaches including differentiable search approaches (Liu et al., 2019; Mok et al., 2021) and train-free approaches (Xu et al., 2021; Zhu et al., 2022a; Shu et al., 2022; Mellor et al., 2021) are deferred to Table 7 of the appendix.

## 4 ROBUST GENERALIZATION GUARANTEES UNDER NAS

Till now, we have built NAS-RobBench-201 to search robust architectures under adversarial training. To expedite the search process, NTK-based NAS algorithms allow for a train-free style in which neural architectures can be initially screened based on the minimum eigenvalue of NTK as well as its variants, e.g., Chen et al. (2021); Xu et al. (2021); Mok et al. (2022); Zhu et al. (2022a). Admittedly, it's worth noting that these methods build upon early analysis on FCNNs under standard training within the learning theory community (Du et al., 2019c; Cao & Gu, 2019; Lee et al., 2019). Consequently, our subsequent theoretical results aim to provide a solid groundwork for the development of NTK-based robust NAS algorithms. Below, the problem setting is introduced in Section 4.1. The (robust)

generalization guarantees of FCNNs as well as CNNs are present in Section 4.2. Detailed definitions for the notations can be found in Appendix A.

## 4.1 PROBLEM SETTING

We consider a search space suitable for residual FCNNs and residual CNNs. To be specific, the class of $L$-layer ($L \geq 3$) residual FCNN with input $\boldsymbol{x} \in \mathbb{R}^d$ and output $f(\boldsymbol{x}, \boldsymbol{W}) \in \mathbb{R}$ is defined as follows:

$$
\begin{aligned}
\boldsymbol{f}_1 &= \sigma_1(\boldsymbol{W}_1 \boldsymbol{x}), \qquad \boldsymbol{f}_l = \frac{1}{L}\sigma_l(\boldsymbol{W}_l \boldsymbol{f}_{l-1}) + \alpha_{l-1} \boldsymbol{f}_{l-1}, \quad 2 \leq l \leq L-1, \\
f_L &= \langle \boldsymbol{w}_L, \boldsymbol{f}_{L-1} \rangle, \quad f(\boldsymbol{x}, \boldsymbol{W}) = f_L,
\end{aligned}
\tag{1}
$$

where $\boldsymbol{W}_1 \in \mathbb{R}^{m \times d}$, $\boldsymbol{W}_l \in \mathbb{R}^{m \times m}$, $l = 2, ..., L-1$ and $\boldsymbol{w}_L \in \mathbb{R}^m$ are learnable weights under Gaussian initialization i.e., $\boldsymbol{W}_l^{(i,j)} \sim \mathcal{N}(0, 1/m)$, for $l \in [L]$. This is the typical NTK initialization to ensure the convergence of the NTK (Allen-Zhu et al., 2019; Zhu et al., 2022a). We use $\boldsymbol{W} := (\boldsymbol{W}_1, ..., \boldsymbol{w}_L) \in \mathcal{W}$ to represent the collection of all weights. The binary parameter $\alpha_l \in \{0, 1\}$ determines whether a skip connection is used and $\sigma_l(\cdot)$ represents an activation function in the $l^{\text{th}}$ layer. Similarly, we define $L$-layer ($L \geq 3$) residual CNNs with input $\boldsymbol{X} \in \mathbb{R}^{d \times p}$, where $d$ denotes the input channels and $p$ represents the pixels, and output $f(\boldsymbol{X}, \boldsymbol{W}) \in \mathbb{R}$ as follows:

$$
\begin{aligned}
\boldsymbol{F}_1 &= \sigma_1(\boldsymbol{\mathcal{W}}_1 * \boldsymbol{X}), \qquad \boldsymbol{F}_l = \frac{1}{L}\sigma_l(\boldsymbol{\mathcal{W}}_l * \boldsymbol{F}_{l-1}) + \alpha_{l-1}\boldsymbol{F}_{l-1}, \quad 2 \leq l \leq L-1, \\
\boldsymbol{F}_L &= \langle \boldsymbol{W}_L, \boldsymbol{F}_{L-1} \rangle, \qquad f(\boldsymbol{X}, \boldsymbol{W}) = \boldsymbol{F}_L,
\end{aligned}
\tag{2}
$$

where $\boldsymbol{\mathcal{W}}_1 \in \mathbb{R}^{\kappa \times m \times d}, \boldsymbol{\mathcal{W}}_l \in \mathbb{R}^{\kappa \times m \times m}$, $l = 2, ..., L-1$, and $\boldsymbol{W}_L \in \mathbb{R}^{m \times p}$ are learnable weights with $m$ channels and $\kappa$ filter size. We define the convolutional operator between $\boldsymbol{X}$ and $\boldsymbol{\mathcal{W}}_1$ as:

$$
(\boldsymbol{\mathcal{W}}_1 * \boldsymbol{X})^{(i,j)} = \sum_{u=1}^{\kappa}\sum_{v=1}^{d} \boldsymbol{\mathcal{W}}_1^{(u,i,v)} \boldsymbol{X}^{(v, j+u-\frac{\kappa+1}{2})}, \text{for } i \in [m], j \in [p],
$$

where we use zero-padding, i.e., $\boldsymbol{X}^{(v,c)} = 0$ for $c < 1$ or $c > p$.

Firstly, we introduce the following two standard assumptions to establish the theoretical result.

**Assumption 1** (Normalization of inputs). *We assume the input space of FCNN is:* $\mathcal{X} \subseteq \{\boldsymbol{x} \in \mathbb{R}^d : \|\boldsymbol{x}\|_2 = 1\}$. *Similarly, the input space of CNN is:* $\mathcal{X} \subseteq \{\boldsymbol{X} \in \mathbb{R}^{d \times p} : \|\boldsymbol{X}\|_{\text{F}} = 1\}$.

**Assumption 2** (Lipschitz of activation functions). *We assume there exist two positive constants $C_{l,\text{Lip}}$ and $C_{\text{Lip}}$ such that $|\sigma_l(0)| \leq C_{l,\text{Lip}} \leq C_{\text{Lip}}$ with $l \in [L]$ and for any $z, z' \in \mathbb{R}$:*

$$
|\sigma_l(z) - \sigma_l(z')| \leq C_{l,\text{Lip}}|z - z'| \leq C_{\text{Lip}}|z - z'|,
$$

*where $C_{\text{Lip}}$ is the maximum value of the Lipschitz constants of all activation functions in the networks.*

**Remarks:** *a)* The first assumption is standard in deep learning theory and attainable in practice as we can always scale the input (Zhang et al., 2020). *b)* The second assumption covers a range of commonly used activation functions, e.g., ReLU, LeakyReLU, or sigmoid (Du et al., 2019a).

Based on our description, the **search space** for the class of FCNNs and CNNs includes: *a)* Activation function search: any activation function $\sigma_l$ that satisfies Assumption 2. *b)* Skip connection search: whether $\alpha_l$ is zero or one. *c)* Convolutional filter size search: the value of $\kappa$.

Below we describe the adversarial training of FCNN while the corresponding one for CNN can be defined in the same way by changing the input as $\boldsymbol{X}$.

**Definition 1** ($\rho$-Bounded adversary). An adversary $\mathcal{A}_\rho : \mathcal{X} \times \mathbb{R} \times \mathcal{W} \to \mathcal{X}$ is $\rho$-bounded for $\rho > 0$ if it satisfies: $\mathcal{A}_\rho(\boldsymbol{x}, y, \boldsymbol{W}) \in \hat{\mathcal{B}}(\boldsymbol{x}, \rho)$. We denote by $\mathcal{A}_\rho^*$ the worst-case $\rho$-bounded adversary given a loss function $\ell$: $\mathcal{A}_\rho^*(\boldsymbol{x}, y, \boldsymbol{W}) := \text{argmax}_{\hat{\boldsymbol{x}} \in \hat{\mathcal{B}}(\boldsymbol{x}, \rho)} \ell(y f(\hat{\boldsymbol{x}}, \boldsymbol{W}))$.

For notational simplicity, we simplify the above notations as $\mathcal{A}_\rho(\boldsymbol{x}, \boldsymbol{W})$ and $\mathcal{A}_\rho^*(\boldsymbol{x}, \boldsymbol{W})$ by omitting the label $y$. Without loss of generality, we restrict our input $\boldsymbol{x}$ as well as its perturbation set $\hat{\mathcal{B}}(\boldsymbol{x}, \rho)$ within the surface of the unit ball $\mathcal{S} := \{\boldsymbol{x} \in \mathbb{R}^d : \|\boldsymbol{x}\|_2 = 1\}$ (Gao et al., 2019).

We employ the cross-entropy loss $\ell(z) := \log[1 + \exp(-z)]$ for training. We consider the following multi-objective function involving with the standard training loss $\mathcal{L}^{\text{clean}}(\boldsymbol{W})$ under empirical risk minimization and adversarial training loss $\mathcal{L}^{\text{robust}}(\boldsymbol{W})$ with adversary in Definition 1:

$$\min_{\boldsymbol{W}} (1-\beta) \underbrace{\frac{1}{N} \sum_{i=1}^{N} \ell[y_i f(\boldsymbol{x}_i, \boldsymbol{W})]}_{:= \mathcal{L}^{\text{clean}}(\boldsymbol{W})} + \beta \underbrace{\frac{1}{N} \sum_{i=1}^{N} \ell[y_i f(\mathcal{A}_\rho(\boldsymbol{x}_i, \boldsymbol{W}), \boldsymbol{W})]}_{:= \mathcal{L}^{\text{robust}}(\boldsymbol{W})}, \tag{3}$$

where the regularization parameter $\beta \in [0,1]$ is for a trade-off between the standard training and adversarial training. The $\beta = 0$ case refers to the standard training and $\beta = 1$ corresponds to the adversarial training. Based on our description, the neural network training by stochastic gradient descent (SGD) with a step-size $\gamma$ is shown in Algorithm 1.

## 4.2 GENERALIZATION BOUNDS

NTK plays an important role in understanding the generalization of neural networks (Jacot et al., 2018). Specifically, the NTK is defined as the inner product of the network Jacobian w.r.t the weights at initialization with infinite width, i.e., $k(\boldsymbol{x}_i, \boldsymbol{x}_j) = \lim_{m \to \infty} \left\langle \frac{\partial f(\boldsymbol{x}_i, \boldsymbol{W}^{[1]})}{\partial \text{vec}(\boldsymbol{W})}, \frac{\partial f(\boldsymbol{x}_j, \boldsymbol{W}^{[1]})}{\partial \text{vec}(\boldsymbol{W})} \right\rangle$. To present the generalization theory, let us denote the NTK matrix for clean accuracy as:

$$\boldsymbol{K}_{\text{all}} := (1-\beta)^2 \boldsymbol{K} + \beta(1-\beta)(\bar{\boldsymbol{K}}_\rho + \bar{\boldsymbol{K}}_\rho^\top) + \beta^2 \hat{\boldsymbol{K}}_\rho, \tag{4}$$

where $\boldsymbol{K}^{(i,j)} = k(\boldsymbol{x}_i, \boldsymbol{x}_j)$ is called the clean NTK, $\bar{\boldsymbol{K}}_\rho^{(i,j)} = k(\boldsymbol{x}_i, \mathcal{A}_\rho^*(\boldsymbol{x}_j, \boldsymbol{W}^{[1]}))$ is the *cross* NTK, and $\hat{\boldsymbol{K}}_\rho^{(i,j)} = k(\mathcal{A}_\rho^*(\boldsymbol{x}_i, \boldsymbol{W}^{[1]}), \mathcal{A}_\rho^*(\boldsymbol{x}_j, \boldsymbol{W}^{[1]}))$ is the robust NTK, for any $i,j \in [N]$. Similarly, denote the NTK for robust accuracy as:

$$\widetilde{\boldsymbol{K}}_{\text{all}} = (1-\beta)^2 \hat{\boldsymbol{K}}_\rho + \beta(1-\beta)(\bar{\boldsymbol{K}}_{2\rho} + \bar{\boldsymbol{K}}_{2\rho}^\top) + \beta^2 \hat{\boldsymbol{K}}_{2\rho}, \tag{5}$$

where the robust NTK $\hat{\boldsymbol{K}}_\rho$ has been defined in Eq. (4), $\bar{\boldsymbol{K}}_{2\rho}^{(i,j)} = k(\mathcal{A}_\rho^*(\boldsymbol{x}_i, \boldsymbol{W}^{[1]}), \hat{\boldsymbol{x}}_j)$, and $\hat{\boldsymbol{K}}_{2\rho}^{(i,j)} = k(\hat{\boldsymbol{x}}_i, \hat{\boldsymbol{x}}_j)$, for $i,j \in [N]$, with $\hat{\boldsymbol{x}}_j = \mathcal{A}_\rho^*(\mathcal{A}_\rho^*(\boldsymbol{x}_j, \boldsymbol{W}^{[1]}), \boldsymbol{W}^{[1]})$ under "twice" perturbation. Note that the formulation of "twice" perturbation allows seeking a perturbed point outside the radius of $\rho$, indicating stronger adversarial data is used. Such perturbation is essentially different from doubling the step size under a single $\rho$. Interestingly, such a scheme has the benefit of avoiding catastrophic overfitting (de Jorge et al., 2022). Accordingly, we have the following theorem on generalization bounds for clean/robust accuracy with the proof deferred to Appendix C.

**Theorem 1** (Generalization bound of FCNN by NAS). *Denote the expected clean* 0-1 *loss as* $\mathcal{L}_{0-1}^{\text{clean}}(\boldsymbol{W}) := \mathbb{E}_{(\boldsymbol{x},y)}[\mathbb{1}\{yf(\boldsymbol{x}, \boldsymbol{W}) < 0\}]$, *and expected robust* 0-1 *loss as* $\mathcal{L}_{0-1}^{\text{robust}}(\boldsymbol{W}) := \mathbb{E}_{(\boldsymbol{x},y)}[\mathbb{1}\{yf(\mathcal{A}_\rho(\boldsymbol{x}, \boldsymbol{W}), \boldsymbol{W}) < 0\}]$. *Consider the residual FCNNs in Eq.* (1) *by activation function search and skip connection search, under Assumption 1 and 2 with* $C_{\text{Lip}}$. *If one runs Algorithm 1 with a step-size* $\gamma = \nu \sqrt{\min\{\boldsymbol{y}^\top(\boldsymbol{K}_{\text{all}})^{-1}\boldsymbol{y}, \boldsymbol{y}^\top(\widetilde{\boldsymbol{K}}_{\text{all}})^{-1}\boldsymbol{y}\}} / (\sqrt{C_{\text{Lip}}} e^{C_{\text{Lip}}} m\sqrt{N})$ *for small enough absolute constant* $\nu$ *and width* $m \geq m^\star$, *where* $m^\star$ *depends on* $(N, L, C_{\text{Lip}}, \lambda_{\min}(\boldsymbol{K}_{\text{all}}), \lambda_{\min}(\widetilde{\boldsymbol{K}}_{\text{all}}), \beta, \rho, \delta)$, *then for any* $\rho \leq 1$, *with probability at least* $1 - \delta$ *over the randomness of* $\boldsymbol{W}^{[1]}$, *we obtain:*

$$\mathbb{E}_{\bar{\boldsymbol{W}}}\left(\mathcal{L}_{0-1}^{\text{clean}}(\bar{\boldsymbol{W}})\right) \lesssim \tilde{\mathcal{O}}\left(\sqrt{\frac{L^2 \boldsymbol{y}^\top \boldsymbol{K}_{\text{all}}^{-1} \boldsymbol{y}}{N}}\right) + \mathcal{O}\left(\sqrt{\frac{\log(1/\delta)}{N}}\right),$$

$$\mathbb{E}_{\bar{\boldsymbol{W}}}\left(\mathcal{L}_{0-1}^{\text{robust}}(\bar{\boldsymbol{W}})\right) \lesssim \tilde{\mathcal{O}}\left(\sqrt{\frac{L^2 \boldsymbol{y}^\top \widetilde{\boldsymbol{K}}_{\text{all}}^{-1} \boldsymbol{y}}{N}}\right) + \mathcal{O}\left(\sqrt{\frac{\log(1/\delta)}{N}}\right), \tag{6}$$

*where* $\lambda_{\min}(\cdot)$ *indicates the minimum eigenvalue of the NTK matrix, the expectation in the LHS is taken over the uniform sample of* $\bar{\boldsymbol{W}}$ *from* $\{\boldsymbol{W}^{[1]}, ..., \boldsymbol{W}^{[N]}\}$, *as illustrated in Algorithm 1.*

**Remarks:** *a)* By courant minimax principle (Golub & Van Loan, 1996), we can further have $\boldsymbol{y}^\top \boldsymbol{K}_{\text{all}}^{-1} \boldsymbol{y} \leq \frac{\boldsymbol{y}^\top \boldsymbol{y}}{\lambda_{\min}(\boldsymbol{K}_{\text{all}})}$, and $\boldsymbol{y}^\top \hat{\boldsymbol{K}}_{\text{all}}^{-1} \boldsymbol{y} \leq \frac{\boldsymbol{y}^\top \boldsymbol{y}}{\lambda_{\min}(\hat{\boldsymbol{K}}_{\text{all}})}$. Cao & Gu (2019); Zhu et al. (2022b) prove

---

**Algorithm 1:** Multi-objective adversarial training with stochastic gradient descent

---

**Input:** data distribution $\mathcal{D}$, adversary $\mathcal{A}_\rho$, step size $\gamma$, iteration $N$, initialized weight $\boldsymbol{W}^{[1]}$.

**for** $i = 1$ **to** $N$ **do**

Sample $(\boldsymbol{x}_i, y_i)$ from $\mathcal{D}$.

$\boldsymbol{W}^{[i+1]} = \boldsymbol{W}^{[i]} - \gamma \cdot (1-\beta) \nabla_{\boldsymbol{W}} \ell\big(y_i f(\boldsymbol{x}_i, \boldsymbol{W}^{[i]})\big) - \gamma \cdot \beta \nabla_{\boldsymbol{W}} \ell\big(y_i f(\mathcal{A}_\rho(\boldsymbol{x}_i, \boldsymbol{W}^{[i]}), \boldsymbol{W}^{[i]})\big)$ .

**end for**

Randomly choose $\bar{\boldsymbol{W}}$ uniformly from $\big\{\boldsymbol{W}^{[1]}, ..., \boldsymbol{W}^{[N]}\big\}$.

**Output**: $\bar{\boldsymbol{W}}$.

---

for FCNN under standard training that the standard generalization error is affected by only $\lambda_{\min}(\boldsymbol{K})$. As a comparison, our result exhibits that under multi-objective training, the generalization error is controlled by a mixed kernel $\boldsymbol{K}_{\text{all}}$ with regularization factor $\beta$. We can recover their result by taking $\beta = 0$, i.e., standard training under the clean NTK. Taking $\beta = 1$ falls in the case of adversarial training with robust NTK. *c)* Our theory demonstrates that the clean accuracy is affected by both the clean NTK and robust NTK, but the robust generalization bound is influenced by the robust NTK and its "twice" perturbation version.

Here we extend our generalization bound to CNN with proof postponed to Appendix E.

**Corollary 1.** *Consider the residual CNN defined in Eq. (2). If one applies Algorithm 1 with a step-size $\gamma = \nu \min\{\boldsymbol{y}^\top (\boldsymbol{K}_{\text{all}})^{-1} \boldsymbol{y}, \ \boldsymbol{y}^\top (\widetilde{\boldsymbol{K}}_{\text{all}})^{-1} \boldsymbol{y}\} / (\sqrt{C_{\text{Lip}} p} \kappa e^{C_{\text{Lip}} \sqrt{\kappa}} m \sqrt{N})$ for some small enough absolute constant $\nu$ and the width $m \geq m^\star$, where $m^\star$ depends on $(N, L, p, \kappa, C_{\text{Lip}}, \lambda_{\min}(\boldsymbol{K}_{\text{all}}), \lambda_{\min}(\widetilde{\boldsymbol{K}}_{\text{all}}), \beta, \rho, \delta)$, then under Assumption 1 and 2, for any $\rho \leq 1$, with probability at least $1 - \delta$, one has the generalization result as in Eq. (6).*

**Remarks:** Notice that the filter size $\kappa$ affects the step-size and the required over-parameterization condition. A larger filter size enables larger step-sizes to achieve a faster convergence rate, which aligns with practical insights (Ding et al., 2022).

## 4.3 Correlation between NTK and accuracy on NAS-RobBench-201

In Figure 5, we plot the Spearman correlation between different NTK scores and various accuracy metrics. Specifically, we use adversarial data with PGD attack to construct the kernel $\hat{\boldsymbol{K}}_\rho$ defined in Eq. (4). Then, to efficiently compute $\lambda_{\min}$, we estimate it by its Frobenius norm as in Xu et al. (2021), and we simply name it NTK-score. Specifically, we focus on 5 distinct NTK variants, including clean NTK, robust NTKs with $\rho \in \{3/255, 8/255\}$, and the corresponding robust twice NTKs. Regarding the robust twice NTK, we first perform one PGD attack on the raw image data to generate the adversarial data, and then we perform the same attack on this adversarial data again and use it to construct the NTK. Our findings reveal that robust NTKs exhibit a notably stronger correlation when compared to standard NTK under adversarial training. More interestingly, we observe that the correlation is increasing for metrics with larger perturbation. Such findings persist when examining NTKs with FGSM attack, as we elaborate in Appendix F.

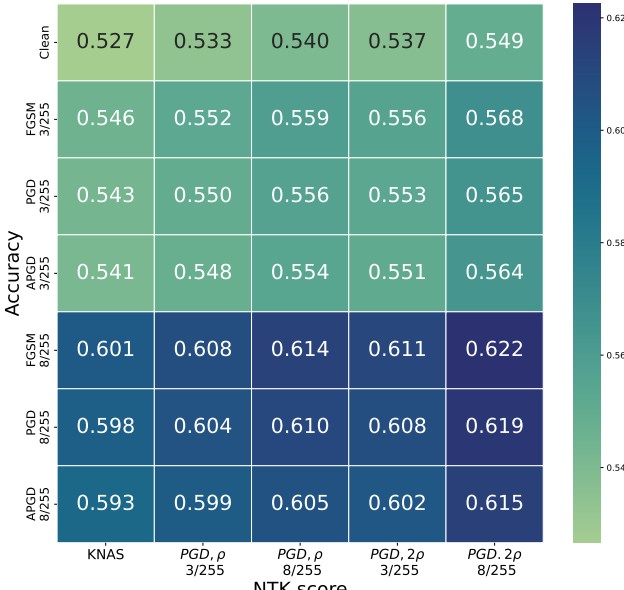

Figure 5: Spearman coefficient between NTK-scores and various metrics. Labels with $2\rho$ in the x-axis indicate the scores w.r.t the robust twice NTK while labels with $\rho$ indicates the score w.r.t the robust NTK.

## 5 CONCLUSION

In this paper, to facilitate the research of the NAS community, we release NAS-RobBench-201 that includes the robust evaluation result of $6466$ architectures on $3$ image datasets under adversarial training. Furthermore, we study the robust generalization guarantee of both FCNN and CNN for NAS. Our result reveals how various NTKs can affect the robustness of neural networks, which can motivate new designs of robust NAS approaches. The first limitation is that our benchmark and theoretical results do not explore the cutting-edge (vision) Transformers. Additionally, our NTK-based analysis studies the neural network in the linear regime, which can not fully explain its success in practice (Allen-Zhu et al., 2019; Yang & Hu, 2021). Nevertheless, we believe both our benchmark and theoretical analysis can be useful for practitioners in exploring robust architectures.

### ACKNOWLEDGEMENTS

We thank the reviewers for their constructive feedback. We are grateful to Kailash Budhathoki from Amazon for the assistance. This work has received funding from the Swiss National Science Foundation (SNSF) under grant number 200021_205011. This work was supported by Hasler Foundation Program: Hasler Responsible AI (project number 21043). Corresponding author: Fanghui Liu.

## ETHICS STATEMENT

On one hand, in the pursuit of facilitating innovation in robust neural architecture design, we build a benchmark by using around 107k GPU hours. We acknowledge that such a substantial computational footprint has environmental consequences, e.g., energy consumption and carbon emissions. On the other hand, the core target of robust NAS is to guarantee that AI models excel not only in accuracy but also in robustness towards malicious data. Therefore, we anticipate positive societal consequences stemming from our research efforts.

## REPRODUCIBILITY STATEMENT

Details on the construction of the benchmark can be found in Section 3.1, including the description of search space, dataset, and hyperparameters setting. Furthermore, in the supplementary material, we provide the evaluation results of all architectures in the proposed benchmark. The code and the pre-trained weight will be publicly released after the paper's acceptance. Regarding the theoretical result, all the proofs can be found in Appendix C to E.

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

- The proof for the lower bound on the minimum eigenvalue of the NTK is present in Appendix D.
- The extension of theoretical results to CNN can be found at Appendix E.
- Further details on the experiment are developed in Appendix F.
- Limitations and societal impact of this work are discussed in Appendix G and Appendix H, respectively.

## A  SYMBOLS AND NOTATION

Vectors/matrices/tensors are symbolized by lowercase/uppercase/calligraphic boldface letters, e.g., $\boldsymbol{w}, \boldsymbol{W}, \boldsymbol{\mathcal{W}}$. We use $\|\cdot\|_{\mathrm{F}}$ and $\|\cdot\|_2$ to represent the Frobenius norm and the spectral norm of a matrix, respectively. The Euclidean norm of a vector is symbolized by $\|\cdot\|_2$. The superscript with brackets is used to represent a particular element of a vector/matrix/tensor, e.g., $\boldsymbol{w}^{(i)}$ is the $i$-th element of $\boldsymbol{w}$. The superscript with brackets symbolizes the variables at different training step, e.g., $\boldsymbol{W}^{[t]}$. Regarding the subscript, we denote by $\boldsymbol{W}_l$ the learnable weights at $l$-th layer, $\boldsymbol{x}_i$ the $i$-th input data. $[L]$ is defined as a shorthand of $\{1, 2, ..., L\}$ for any positive integer $L$. We denote by $\mathcal{X} \subset \mathbb{R}^d$ and $\mathcal{Y} \subset \mathbb{R}$ the input space and the output space, respectively. The training data $(\boldsymbol{x}_i, y_i)$ is drawn from a probability measure $\mathcal{D}$ on $\mathcal{X} \times \mathcal{Y}$. For an input $\boldsymbol{x} \in \mathcal{X}$, we symbolize its $\rho$-neighborhood by

$$\hat{\mathcal{B}}(\boldsymbol{x}, \rho) := \{\hat{\boldsymbol{x}} \in \mathcal{X} : \|\hat{\boldsymbol{x}} - \boldsymbol{x}\|_2 \leq \rho\} \cap \mathcal{X}.$$

For any $\boldsymbol{W} \in \mathcal{W}$, we define its $\tau$-neighborhood as follows:

$$\mathcal{B}(\boldsymbol{W}, \tau) := \{\boldsymbol{W}' \in \mathcal{W} : \|\boldsymbol{W}'_l - \boldsymbol{W}_l\|_{\mathrm{F}} \leq \tau, l \in [L]\}.$$

We summarize the core symbols and notation in Table 3.

## B  COMPLETE RELATED WORK AND OUR CONTRIBUTION

### B.1  COMPLETE RELATED WORK

**Adversarial example and defense.** Since the seminal work of (Szegedy et al., 2013) illustrated that neural networks are vulnerable to inputs with small perturbations, several approaches have emerged to defend against such attacks, e.g., adversarial training (Goodfellow et al., 2015; Madry et al., 2018), feature denoising (Xu et al., 2018; Xie et al., 2019), modifying network architecture (Xiao et al., 2020), regularizing Lipschitz constant (Cisse et al., 2017) or input gradient (Ross & Doshi-Velez, 2018). Adversarial training is one of the most effective defense mechanisms that minimizes the empirical training loss based on the adversarial data, which is obtained by solving a maximum problem. Goodfellow et al. (2015) use Fast Gradient Sign Method (FGSM), a one-step method, to generate the adversarial data. However, FGSM relies on the linearization of loss around data points and the resulting model is still vulnerable to other more sophisticated adversaries. Multi-step methods are subsequently proposed to further improve the robustness, e.g., multi-step FGSM (Kurakin et al., 2018), multi-step PGD (Madry et al., 2018). Other methods of generating adversarial examples include L-BFGS (Szegedy et al., 2013), C&W attack (Carlini & Wagner, 2017).

**NAS and benchmarks.** Over the years, significant strides have been made towards developing NAS algorithms, which have been explored from various angles, e.g., differentiable search with weight-sharing (Liu et al., 2019; Zela et al., 2020; Ye et al., 2022), reinforcement learning (Baker et al., 2017; Zoph & Le, 2017; Zoph et al., 2018), evolutionary algorithm (Suganuma et al., 2017; Real et al., 2017; 2019), NTK-based methods (Chen et al., 2021; Xu et al., 2021; Mok et al., 2022; Zhu et al.,

Table 3: Core symbols and notations used in this paper.

| Symbol | Dimension(s) | Definition |
|---|---|---|
| $\mathcal{N}(\mu,\sigma)$ | - | Gaussian distribution with mean $\mu$ and variance $\sigma$ |
| $[L]$ | - | Shorthand of $\{1,2,...,L\}$ |
| $\mathcal{O}, o, \Omega$ and $\Theta$ | - | Standard Bachmann–Landau order notation |
| $\|\boldsymbol{w}\|_2$ | - | Euclidean norms of vector $\boldsymbol{w}$ |
| $\|\boldsymbol{W}\|_2$ | - | Spectral norms of matrix $\boldsymbol{W}$ |
| $\|\boldsymbol{W}\|_{\mathrm{F}}$ | - | Frobenius norms of matrix $\boldsymbol{W}$ |
| $\lambda_{\min}(\boldsymbol{W}), \lambda_{\max}(\boldsymbol{W})$ | - | Minimum and maximum eigenvalues of matrix $\boldsymbol{W}$ |
| $\sigma_{\min}(\boldsymbol{W}), \sigma_{\max}(\boldsymbol{W})$ | - | Minimum and Maximum singular values of matrix $\boldsymbol{W}$ |
| $\boldsymbol{w}^{(i)}$ | - | $i$-th element of vector $\boldsymbol{w}$ |
| $\boldsymbol{W}^{(i,j)}$ | - | $(i,j)$-th element of matrix $\boldsymbol{W}$ |
| $\boldsymbol{W}^{[t]}$ | - | matrix $\boldsymbol{W}$ at time step $t$ |
| $N$ | - | Number of training data points |
| $d$ | - | Dimension (channel) of the input in FCNN (CNN) |
| $m$ | - | Width (Channel) of intermediate layers in FCNN (CNN) |
| $p$ | - | Pixel of the input in CNN |
| $\kappa$ | - | Filter size in CNN |
| $\boldsymbol{x}_i$ | $\mathbb{R}^d$ | The $i$-th data point |
| $y_i$ | $\mathbb{R}$ | The $i$-th target |
| $\mathcal{D}_X$ | - | Input data distribution |
| $\mathcal{D}_Y$ | - | Target data distribution |
| $\sigma_l(\cdot)$ | - | Element-wise activation function at $l$-th layer in FCNN |
| $\boldsymbol{W}_1, \boldsymbol{W}_l, \boldsymbol{w}_L$ | $\mathbb{R}^{m \times d}, \mathbb{R}^{m \times m}, \mathbb{R}^m$ | Learnable weights in FCNN, $l = 2,...,L-1$ |
| $\boldsymbol{W} := (\boldsymbol{W}_1,...,\boldsymbol{w}_L)$ | | Collection of all learnable weights in FCNN |
| $\boldsymbol{D}_l$ | $\mathbb{R}^{m \times m}$ | Diagonal matrix, $\boldsymbol{D}_l = \mathrm{Diag}(\sigma_l{}'(\boldsymbol{W}_l \boldsymbol{f}_{l-1}))$, for $l \in [L-1]$ in FCNN |

2022a). Most recent work on NAS for robust architecture belongs to the first category. Guo et al. (2020) first trains a huge network that contains every edge in the cell of the search space and then obtains a specific robust architecture based on this trained network. Mok et al. (2021) propose a robust NAS method by imposing a regularized term in the DARTS objective to ensure a smoother loss landscape. These algorithms are different from the NTK-based methods, where we can select a robust architecture based on the NTK metrics without training a huge network.

Along with the evolution of NAS algorithms, the development of NAS benchmarks is also important for an efficient and fair comparison. Regarding clean accuracy, several tabular benchmarks, e.g., NAS-Bench-101 (Ying et al., 2019), TransNAS-Bench-101 (Duan et al., 2021), NAS-Bench-201 (Dong & Yang, 2020) and NAS-Bench-301 (Zela et al., 2022), have been proposed that include the evaluation performance of architectures under standard training on image classification tasks. More recently, Jung et al. (2023) extend the scope of the benchmark towards robustness by evaluating the performance of the pre-trained architecture in NAS-Bench-201 in terms of various adversarial attacks. However, all of these benchmarks are under standard training, which motivates us to release the first NAS benchmark that contains the performance of each architecture under adversarial training.

**Neural tangent kernel.** Neural tangent kernel (NTK) was first proposed by Jacot et al. (2018) that connects the training dynamics of over-parameterized neural network, where the number of parameters is greater than the number of training data, to kernel regression. In theory, NTK provides a tractable analysis for several phenomena in deep learning, e.g., convergence (Allen-Zhu et al., 2019), generalization (Huang et al., 2020), and spectral bias (Cao et al., 2019; Choraria et al., 2022). The study on the NTK has been expanded to a range of neural networks including, fully-connected networks (Jacot et al., 2018), convolutional networks (Bietti & Mairal, 2019), and graph networks (Du et al., 2019b). The analysis has also been extended from standard training to adversarial training. In particular, Gao et al. (2019) prove the convergence of FCNN with quadratic ReLU activation function, relying on the NTK theory, which suggests that the weights of the network are close to their initialization (Jacot et al., 2018). Zhang et al. (2020) provide a refined analysis for FCNN with ReLU activation functions and prove that polynomial width suffices instead of exponential width. More recently, Wang et al. (2022) present a convergence analysis convergence of 2-layer FCNN under certified robust training. From the generalization perspective, the guarantee of over-parameterized FCNN under standard training has been established in Cao & Gu (2019); Zhu et al. (2022a). In this work, we extend the scope of generalization guarantee of over-parameterized networks to multi-objective training, which covers the case of both standard training and adversarial training. Meanwhile, Huang et al. (2021) studies

the impact of network width and depth on the robustness of adversarially trained DNNs. From the theoretical side, Huang et al. (2021) is based on the Lipschitz constant for robustness evaluation while our analysis builds a general framework for the generalization guarantees on both clean and robust accuracy

NTK has brought insights into several practical applications. For example, in NAS, (Xu et al., 2021; Shu et al., 2022; Zhu et al., 2022a) show that one can use NTK as a guideline to select architectures without training. As a kernel, NTK can be used in supervised learning and has demonstrated remarkable performance over neural networks on small datasets (Arora et al., 2020). Lastly, NTK has also shown its potential in virtual drug screening, image impainting (Radhakrishnan et al., 2022), and MRI reconstruction (Tancik et al., 2020).

Let us now provide further intuition on why training neural networks via gradient descent can be connected to the NTK. Let us denote the loss as $\ell(\boldsymbol{W}) = \frac{1}{2} \sum_{n=1}^{N} (f(\boldsymbol{x}_n; \boldsymbol{W}) - y_n)^2$. By choosing an infinitesimally small learning rate, one has the following gradient flow: $\frac{d\boldsymbol{W}^{[t]}}{dt} = -\nabla \ell(\boldsymbol{W}^{[t]})$. By simple chain rule, we can observe that the network outputs admit the following dynamics: $\frac{df(\boldsymbol{W}^{[t]})}{dt} = -\boldsymbol{K}^{[t]}(f(\boldsymbol{W}^{[t]}) - \boldsymbol{y})$, where $\boldsymbol{K}^{[t]} = \left(\frac{\partial f(\boldsymbol{W}^{[t]})}{\partial \boldsymbol{W}}\right)\left(\frac{\partial f(\boldsymbol{W}^{[t]})}{\partial \boldsymbol{W}}\right)^{\top} \in \mathbb{R}^{N \times N}$ is the NTK at $t$ step. Hence, NTK can be used to characterize the training process of neural networks.

## B.2 CONTRIBUTIONS AND RELATIONSHIP TO PRIOR WORK

In this section, we clarify the contribution of our work when compared to the prior work on robust NAS. Specifically, our work makes contributions both theoretically and empirically.

From the theoretical side, our work differs from previous theoretical work (Zhu et al., 2022a; Cao & Gu, 2019) in the following aspects:

- Problem setting: Cao & Gu (2019) build the generalization bound of FCNN via NTK and (Zhu et al., 2022a) extend this result under the activation function search and skip connection search. Instead, our work studies the robust generalization bound of both FCNN/CNN under multi-objective adversarial training. How to handle the multi-objective, clean/robust accuracy, more general architecture search (CNN) is still unknown in prior theoretical work.
- Results: Differently from Zhu et al. (2022a); Cao & Gu (2019) that are based on the sole NTK for generalization guarantees, our result demonstrates that, under adversarial training, the generalization performance (clean accuracy and robust accuracy) is affected by several NTKs. Concretely, the clean accuracy is determined by one clean NTK and robust NTK; while robust accuracy is determined by robust NTK and its "twice" perturbation version.
- The technical difficulties lie in a) how to build the proof framework under multi-objective training by the well-designed joint of NTKs and b) how to tackle the coupling relationship among several NTKs and derive the lower bound of the minimum eigenvalue of NTKs. Accordingly, our work builds the generalization guarantees of the searched architecture under multi-objective adversarial training. Our results demonstrate the effect of different search schemes, perturbation radius, and the balance parameter, which doesn't exist in previous literature.

From the application side, we release an adversarially trained benchmark on the NAS-bench201 search space, which differs from previous benchmark in the following aspects:

- Motivation: As a comparison, existing benchmarks on the NAS-bench201 search space are built on standard training (Dong & Yang, 2020; Jung et al., 2023). Since robust NAS algorithms usually include the evaluation results under adversarial training (Guo et al., 2020; Mok et al., 2021), our benchmark facilitates a direct retrieval of these results for reliable reproducibility and efficient assessment.
- Statistical result: In Table 1, we present the rank correlation among different accuracies within our proposed benchmark as well as the benchmark in (Jung et al., 2023). The observation underscores the significance of adversarial training within our benchmark. Specifically, it suggests that employing Jung et al. (2023) to identify a resilient architecture may not guarantee its robustness under the context of adversarial training. Additionally, in Figure 8, we can see a notable difference between these two benchmarks in terms of top architectures id and selected nodes.

- Lastly, we investigate the correlation between various NTK metrics and the robust accuracy of the architecture in the benchmark. By integrating this result with our theoretical generalization bound, we can inspire practitioners to craft more robust and potent algorithms.

## C  PROOF OF THE GENERALIZATION OF RESIDUAL FCNN

### C.1  SOME AUXILIARY LEMMAS

**Lemma 1** (Corollary 5.35 in Vershynin (2012)). *Given a matrix $\boldsymbol{W} \in \mathbb{R}^{m_1 \times m_2}$ where each element is sampled independently from $\mathcal{N}(0,1)$, for every $\zeta \geq 0$, with probability at least $1 - 2\exp(-\zeta^2/2)$ one has:*

$$\sqrt{m_1} - \sqrt{m_2} - \zeta \leq \sigma_{\min}(\boldsymbol{W}) \leq \sigma_{\max}(\boldsymbol{W}) \leq \sqrt{m_1} + \sqrt{m_2} + \zeta,$$

*where $\sigma_{\max}(\boldsymbol{W})$ and $\sigma_{\min}(\boldsymbol{W})$ represents the maximum singular value and the minimum singular value of $\boldsymbol{W}$, respectively.*

**Lemma 2** (Upper bound of spectral norms of initial weight). *Given a weight matrix $\boldsymbol{W}_l^{[1]} \in \mathbb{R}^{m_1 \times m_2}$ where $m_1 \geq m_2$ and each element is sampled independently from $\mathcal{N}(0, \frac{1}{m_1})$, then with probability at least $1 - 2\exp(-m_1/2)$, one has:*

$$\|\boldsymbol{W}_l^{[1]}\|_2 \leq 3.$$

*Proof of Lemma 2.* Following Lemma 1, w.p. at least $1 - 2\exp(-\zeta^2/2)$, one has:

$$\|\boldsymbol{W}_l^{[1]}\|_2 \leq \sqrt{\frac{1}{m_1}}(\sqrt{m_1} + \sqrt{m_2} + \zeta).$$

Setting $\zeta = \sqrt{m_1}$ and using the fact that $m_1 \geq m_2$ completes the proof. $\square$

**Lemma 3** (The order of the network output at initialization). *Fix any $l \in [1, L-1]$ and $\boldsymbol{x}$, when the width satisfies $m = \Omega(N/\delta)$, with probability at least $1 - 2l\exp(-m/2) - \delta$ over the randomness of $\{\boldsymbol{W}_i^{[1]}\}_{i=1}^l$, we have:*

$$C_{\text{fmin}} \leq \left\|\boldsymbol{f}_l(\boldsymbol{x}, \boldsymbol{W}^{[1]})\right\|_2 \leq C_{\text{fmax}},$$

*where $C_{\text{fmax}}$ and $C_{\text{fmax}}$ are some $\Theta(1)$ constant.*

*Proof.* The result can be obtained by simply applying Lemma 2 for the initial weights and Lemma C.1 in Du et al. (2019a). $\square$

The following lemma shows that the perturbation of $\boldsymbol{x}$ can be considered as an equivalent perturbation of the weights.

**Lemma 4.** *Given any fixed input $\boldsymbol{x} \in \mathcal{X}$, with probability at least $1 - 2e^{-m/2}$ over random initialization, for any $\hat{\boldsymbol{x}} \in \mathcal{X}$ satisfying $\|\boldsymbol{x} - \hat{\boldsymbol{x}}\|_2 \leq \rho$, and any $\boldsymbol{W} \in \hat{\mathcal{B}}(\boldsymbol{W}^{[1]}, \tau)$, there exists $\widetilde{\boldsymbol{W}} \in B(\boldsymbol{W}^{[1]}, \tau + 3\rho + \tau\rho)$ such that:*

$$\begin{aligned}
\boldsymbol{f}_l(\hat{\boldsymbol{x}}, \boldsymbol{W}) &= \boldsymbol{f}_l(\boldsymbol{x}, \widetilde{\boldsymbol{W}}), \quad 1 \leq l \leq L-1, \\
\boldsymbol{D}_l(\hat{\boldsymbol{x}}, \boldsymbol{W}) &= \boldsymbol{D}_l(\boldsymbol{x}, \widetilde{\boldsymbol{W}}), \quad 1 \leq l \leq L-1, \\
f(\hat{\boldsymbol{x}}, \boldsymbol{W}) &= f(\boldsymbol{x}, \widetilde{\boldsymbol{W}}).
\end{aligned}$$

*Proof.* Let us define:

$$\widetilde{\boldsymbol{W}}_1 = \boldsymbol{W}_1 + \frac{\boldsymbol{W}_1(\hat{\boldsymbol{x}} - \boldsymbol{x})\boldsymbol{x}^\top}{\|\boldsymbol{x}\|_2^2}.$$

Obviously, $\widetilde{\boldsymbol{W}}_1$ satisfies $\widetilde{\boldsymbol{W}}_1\boldsymbol{x} = \boldsymbol{W}_1\hat{\boldsymbol{x}}$. Then setting $\widetilde{\boldsymbol{W}}_2, ..., \widetilde{\boldsymbol{W}}_L$ equal to $\boldsymbol{W}_2, ..., \boldsymbol{W}_L$ will make the output of the following layers equal. Besides, by Lemma 2, with probability $1 - 2e^{-m/2}$, one has $\left\|\boldsymbol{W}_1^{[1]}\right\|_2 \leq 3$ and thus

$$\left\|\widetilde{W}_1 - W_1\right\|_F = \left\|\frac{W_1(\hat{x} - x)x^\top}{\|x\|_2^2}\right\|_F \leq \frac{\|W_1(\hat{x} - x)\|_2 \|x\|_2}{\|x\|_2^2} = \|W_1\|_2 \|\hat{x} - x\|_2$$

$$\leq \left(\left\|W_1^{[1]}\right\|_2 + \tau\right)\rho \leq 3\rho + \rho\tau,$$

which indicates that $\left\|\widetilde{W} - W\right\|_F \leq 3\rho + \rho\tau$ w.p. Lastly, by triangle inequality, with probability $1 - 2e^{-m/2}$, we have:

$$\left\|\widetilde{W} - \widetilde{W}^{[1]}\right\|_F \leq \left\|\widetilde{W} - W\right\|_F + \left\|W - \widetilde{W}^{[1]}\right\|_F \leq \tau + 3\rho + \tau\rho,$$

which completes the proof. □

The following lemma shows that the network output does not change too much if the weights are close to that in initialization.

**Lemma 5.** *For $\widetilde{W} \in \mathcal{B}(W^{[1]}, \tau)$ with $\tau \leq 3$, if the width satisfies $m = \Omega(N/\delta)$, with probability at least $1 - 2(L-1)\exp(-m/2) - \delta$, one has:*

$$\left\|f_{L-1}(x, \widetilde{W}) - f_{L-1}(x, W^{[1]})\right\|_2 \leq e^{6C_{\text{Lip}}}(C_{\text{Lip}} + C_{\text{fmax}})\tau.$$

*Proof.* To simplify the notation, in the following proof, the variable with $\widetilde{\cdot}$ is related to $\widetilde{W}$, and without $\widetilde{\cdot}$ is related to $W^{[1]}$. For the output of the first layer, we have:

$$\left\|\widetilde{f}_1 - f_1\right\|_2 = \left\|\sigma_1(\widetilde{W}_1 x) - \sigma_1(W_1 x)\right\|_2 \leq C_{\text{Lip}} \left\|\widetilde{W}_1 - W_1\right\|_2 \|x\|_2 \leq \tau C_{\text{Lip}}.$$

For the $l$-th layer ($l \in \{2, 3, ..., L-1\}$), we have:

$$\left\|\widetilde{f}_l - f_l\right\|_2$$

$$\left\|\frac{1}{L}\sigma_l(\widetilde{W}_l \widetilde{f}_{l-1}) + \alpha_{l-1}\widetilde{f}_{l-1} - \frac{1}{L}\sigma_l(W_l f_{l-1}) - \alpha_{l-1} f_{l-1}\right\|_2$$

$$\leq \frac{1}{L}\left\|\sigma_l(\widetilde{W}_l \widetilde{f}_{l-1}) - \sigma_l(W_l f_{l-1})\right\|_2 + \alpha_{l-1}\left\|\widetilde{f}_{l-1} - f_{l-1}\right\|_2 \quad \text{(By triangle inequality)}$$

$$\leq \frac{C_{\text{Lip}}}{L}\left\|\widetilde{W}_l \widetilde{f}_{l-1} - W_l f_{l-1}\right\|_2 + \left\|\widetilde{f}_{l-1} - f_{l-1}\right\|_2 \quad \text{(By the Lipschitz continuity of } \sigma_l\text{)}$$

$$= \frac{C_{\text{Lip}}}{L}\left\|W_l(\widetilde{f}_{l-1} - f_{l-1}) + (\widetilde{W}_l - W_l)\widetilde{f}_{l-1}\right\|_2 + \left\|\widetilde{f}_{l-1} - f_{l-1}\right\|_2$$

$$\leq \frac{C_{\text{Lip}}}{L}\left\{\|W_l\|_2 \left\|\widetilde{f}_{l-1} - f_{l-1}\right\|_2 + \left\|\widetilde{W}_l - W_l\right\|_2 \left\|\widetilde{f}_{l-1}\right\|_2\right\} + \left\|\widetilde{f}_{l-1} - f_{l-1}\right\|_2$$

$$\leq (\frac{C_{\text{Lip}}}{L}\|W_l\|_2 + 1)\left\|\widetilde{f}_{l-1} - f_{l-1}\right\|_2 + \frac{C_{\text{Lip}}}{L}\tau\left(\left\|\widetilde{f}_{l-1} - f_{l-1}\right\|_2 + \|f_{l-1}\|_2\right)$$

$$= \left\{\frac{C_{\text{Lip}}}{L}(\|W_l\|_2 + \tau) + 1\right\}\left\|\widetilde{f}_{l-1} - f_{l-1}\right\|_2 + \frac{C_{\text{Lip}}}{L}\tau\|f_{l-1}\|_2.$$

Therefore, by the inequality recursively and Lemmas 2 and 3, with probability at least $1-2(L-1)\exp(-m/2)-\delta$, we have:

$$\left\|\widetilde{\boldsymbol{f}}_{L-1}-\boldsymbol{f}_{L-1}\right\|_2$$

$$\leq[\frac{C_{\mathrm{Lip}}}{L}(3+\tau)+1]\left\|\widetilde{\boldsymbol{f}}_{L-2}-\boldsymbol{f}_{L-2}\right\|_2+\frac{C_{\mathrm{Lip}}}{L}\tau C_{\mathrm{fmax}}, \quad\text{(By Lemmas 2 and 3)}$$

$$\leq[\frac{C_{\mathrm{Lip}}}{L}(3+\tau)+1]^{L-2}\left\|\widetilde{\boldsymbol{f}}_1-\boldsymbol{f}_1\right\|_2+\sum_{i=0}^{L-3}[\frac{C_{\mathrm{Lip}}}{L}(3+\tau)+1]^i\frac{C_{\mathrm{Lip}}}{L}\tau C_{\mathrm{fmax}} \quad\text{(By recursion)}$$

$$\leq\left(\frac{6C_{\mathrm{Lip}}}{L}+1\right)^{L-2}\tau C_{\mathrm{Lip}}+\sum_{i=0}^{L-3}[\frac{6C_{\mathrm{Lip}}}{L}+1]^i\frac{C_{\mathrm{Lip}}}{L}\tau C_{\mathrm{fmax}}$$

$$=\left(\frac{6C_{\mathrm{Lip}}}{L}+1\right)^{L-2}\tau C_{\mathrm{Lip}}+\frac{1-(6C_{\mathrm{Lip}}/L+1)^{L-2}}{1-6C_{\mathrm{Lip}}/L-1}\frac{C_{\mathrm{Lip}}}{L}\tau C_{\mathrm{fmax}}$$

$$\leq e^{6C_{\mathrm{Lip}}}(C_{\mathrm{Lip}}+C_{\mathrm{fmax}})\tau.$$

$\square$

**Lemma 6.** *For $\widetilde{\boldsymbol{W}},\boldsymbol{W}\in\mathcal{B}(\boldsymbol{W}^{[1]},\tau)$ with $\tau\leq3$, when the width satisfies $m=\Omega(N/\delta)$ and $\rho\leq1$, with probability at least $1-2L\exp(-m/2)-\delta$, one has:*

$$\left|f(\boldsymbol{x},\widetilde{\boldsymbol{W}})-f(\boldsymbol{x},\boldsymbol{W})-\left\langle(1-\beta)\nabla_{\boldsymbol{W}}f(\boldsymbol{x},\boldsymbol{W})+\beta\nabla_{\boldsymbol{W}}f(\mathcal{A}_\rho(\boldsymbol{x},\boldsymbol{W}),\boldsymbol{W}),\widetilde{\boldsymbol{W}}-\boldsymbol{W}\right\rangle\right|$$

$$\leq18e^{6C_{\mathrm{Lip}}}(1+\beta)(C_{\mathrm{Lip}}+C_{\mathrm{fmax}})\tau.$$

$$\left|f(\mathcal{A}_\rho(\boldsymbol{x},\boldsymbol{W}),\widetilde{\boldsymbol{W}})-f(\mathcal{A}_\rho(\boldsymbol{x},\boldsymbol{W}),\boldsymbol{W})-\left\langle(1-\beta)\nabla_{\boldsymbol{W}}f(\boldsymbol{x},\boldsymbol{W})+\beta\nabla_{\boldsymbol{W}}f(\mathcal{A}_\rho(\boldsymbol{x},\boldsymbol{W}),\boldsymbol{W}),\widetilde{\boldsymbol{W}}-\boldsymbol{W}\right\rangle\right|$$

$$\leq18e^{6C_{\mathrm{Lip}}}(2-\beta)(C_{\mathrm{Lip}}+C_{\mathrm{fmax}})\tau.$$

*Proof.* To simplify the notation, in the following proof, the variable with $\widetilde{\cdot}$ is related to $\widetilde{\boldsymbol{W}}$, and without $\widetilde{\cdot}$ is related to $\boldsymbol{W}$. The variable with $\hat{\cdot}$ is related to input $\mathcal{A}_\rho(\boldsymbol{x},\boldsymbol{W})$, and without is related to input $\boldsymbol{x}$. For instance, we denote by:

$$\boldsymbol{f}_l:=\boldsymbol{f}_l(\boldsymbol{x},\boldsymbol{W}), \quad \boldsymbol{D}_l=\mathrm{Diag}(\sigma_l'(\boldsymbol{W}_l\boldsymbol{f}_{l-1})),$$

$$\widetilde{\boldsymbol{f}}_l:=\boldsymbol{f}_l(\boldsymbol{x},\widetilde{\boldsymbol{W}}), \quad \widetilde{\boldsymbol{D}}_l=\mathrm{Diag}(\sigma_l'(\widetilde{\boldsymbol{W}}_l\widetilde{\boldsymbol{f}}_{l-1})),$$

$$\hat{\boldsymbol{f}}_l:=\boldsymbol{f}_l(\hat{\boldsymbol{x}},\boldsymbol{W}), \quad \hat{\boldsymbol{D}}_l=\mathrm{Diag}(\sigma_l'(\boldsymbol{W}_l\hat{\boldsymbol{f}}_{l-1})).$$

Then, let us prove the first inequality. We have:

$$\left|\widetilde{f}-f-\left\langle(1-\beta)\nabla_{\boldsymbol{W}}f+\beta\nabla_{\boldsymbol{W}}\hat{f},\widetilde{\boldsymbol{W}}-\boldsymbol{W}\right\rangle\right|$$

$$=\left|\left\langle\widetilde{\boldsymbol{w}}_L,\widetilde{\boldsymbol{f}}_{L-1}\right\rangle-\langle\boldsymbol{w}_L,\boldsymbol{f}_{L-1}\rangle-\sum_{l=1}^{L}\left\langle(1-\beta)\nabla_{\boldsymbol{W}_l}f+\beta\nabla_{\boldsymbol{W}_l}\hat{f},\widetilde{\boldsymbol{W}}_l-\boldsymbol{W}_l\right\rangle\right|$$

$$=\left|\left\langle\widetilde{\boldsymbol{w}}_L,\widetilde{\boldsymbol{f}}_{L-1}-\boldsymbol{f}_{L-1}\right\rangle+\langle\boldsymbol{f}_{L-1},\widetilde{\boldsymbol{w}}_L-\boldsymbol{w}_L\rangle-\sum_{l=1}^{L-1}\left\langle(1-\beta)\nabla_{\boldsymbol{W}_l}f+\beta\nabla_{\boldsymbol{W}_l}\hat{f},\widetilde{\boldsymbol{W}}_l-\boldsymbol{W}_l\right\rangle\right. \tag{7}$$

$$\left.-\left\langle(1-\beta)\boldsymbol{f}_{L-1}+\beta\hat{\boldsymbol{f}}_{L-1},\widetilde{\boldsymbol{w}}_L-\boldsymbol{w}_L\right\rangle\right|$$

$$\leq\|\widetilde{\boldsymbol{w}}_L\|_2\left\|\widetilde{\boldsymbol{f}}_{L-1}-\boldsymbol{f}_{L-1}\right\|_2+\beta\left\|\boldsymbol{f}_{L-1}-\hat{\boldsymbol{f}}_{L-1}\right\|_2\|\widetilde{\boldsymbol{w}}_L-\boldsymbol{w}_L\|_2$$

$$+\left|\sum_{l=1}^{L-1}\left\langle(1-\beta)\nabla_{\boldsymbol{W}_l}f,\widetilde{\boldsymbol{W}}_l-\boldsymbol{W}_l\right\rangle\right|+\left|\sum_{l=1}^{L-1}\left\langle\beta\nabla_{\boldsymbol{W}_l}\hat{f},\widetilde{\boldsymbol{W}}_l-\boldsymbol{W}_l\right\rangle\right|.$$

The first term can be bounded by Lemmas 2 and 5 as follows:

$$\|\widetilde{\boldsymbol{w}}_L\|_2\left\|\widetilde{\boldsymbol{f}}_{L-1}-\boldsymbol{f}_{L-1}\right\|_2$$

$$\leq\left(\left\|\boldsymbol{w}_L^{[1]}\right\|_2+\left\|\widetilde{\boldsymbol{w}}_L-\boldsymbol{w}_L^{[1]}\right\|_2\right)\left(\left\|\widetilde{\boldsymbol{f}}_{L-1}-\boldsymbol{f}_{L-1}^{[1]}\right\|_2+\left\|\boldsymbol{f}_{L-1}-\boldsymbol{f}_{L-1}^{[1]}\right\|_2\right)$$

$$\leq(3+\tau)2e^{6C_{\mathrm{Lip}}}(C_{\mathrm{Lip}}+C_{\mathrm{fmax}})\tau.$$

The second term can be bounded by Lemmas 3 to 5 with $\rho \leq 1$ as follows:

$$
\begin{aligned}
&\beta\left\|\hat{\boldsymbol{f}}_{L-1}-\boldsymbol{f}_{L-1}\right\|_2\|\widetilde{\boldsymbol{w}}_L-\boldsymbol{w}_L\|_2 \\
&\leq \beta\left(\left\|\boldsymbol{f}_{L-1}^{[1]}-\boldsymbol{f}_{L-1}\right\|_2+\left\|\hat{\boldsymbol{f}}_{L-1}-\hat{\boldsymbol{f}}_{L-1}^{[1]}\right\|_2+\left\|\boldsymbol{f}_{L-1}^{[1]}-\hat{\boldsymbol{f}}_{L-1}^{[1]}\right\|_2\right)\left(\left\|\boldsymbol{w}_L-\boldsymbol{w}_L^{[1]}\right\|_2+\left\|\widetilde{\boldsymbol{w}}_L-\boldsymbol{w}_L^{[1]}\right\|_2\right) \\
&\leq \beta e^{6C_{\text{Lip}}}(C_{\text{Lip}}+C_{\text{fmax}})(2\tau+3\rho)2\tau.
\end{aligned}
\tag{8}
$$

The third term can be bounded by Lemmas 2 and 3 as follows:

$$
\begin{aligned}
&(1-\beta)\left|\sum_{l=1}^{L-1}\left\langle\nabla_{\boldsymbol{W}_l}f,\widetilde{\boldsymbol{W}}_l-\boldsymbol{W}_l\right\rangle\right| \\
&=(1-\beta)\left|\sum_{l=1}^{L-1}\left[\boldsymbol{w}_L^\top\prod_{r=l+1}^{L-1}(\boldsymbol{D}_r\boldsymbol{W}_r+\alpha_{r-1}\boldsymbol{I}_{m\times m})\boldsymbol{D}_l(\widetilde{\boldsymbol{W}}_l-\boldsymbol{W}_l)\boldsymbol{f}_{l-1}\right]\right| \\
&\leq (1-\beta)\sum_{l=1}^{L-1}\left[\|\boldsymbol{w}_L\|_2\prod_{r=l+1}^{L-1}(\|\boldsymbol{D}_r\|_2\|\boldsymbol{W}_r\|_2+1)\|\boldsymbol{D}_l\|_2\left\|\widetilde{\boldsymbol{W}}_l-\boldsymbol{W}_l\right\|_2\|\boldsymbol{f}_{l-1}\|_2\right] \\
&\leq (1-\beta)\sum_{l=1}^{L-1}C_{\text{Lip}}(3+\tau)C_{\text{fmax}}(\frac{C_{\text{Lip}}}{L}(3+\tau)+1)^{L-l-1}\tau \\
&\leq (1-\beta)C_{\text{fmax}}e^{6C_{\text{Lip}}}\tau.
\end{aligned}
\tag{9}
$$

Similarly, the fourth term can be upper bounded by $\beta C_{\text{fmax}}e^{6C_{\text{Lip}}}\tau$. Plugging back Eq. (7) yields:

$$
\begin{aligned}
&\left|f(\boldsymbol{x},\widetilde{\boldsymbol{W}})-f(\boldsymbol{x},\boldsymbol{W})-\left\langle(1-\beta)\nabla_{\boldsymbol{W}}f+\beta\nabla_{\boldsymbol{W}}\hat{f},\widetilde{\boldsymbol{W}}-\boldsymbol{W}\right\rangle\right| \\
&\leq (3+\tau)2e^{6C_{\text{Lip}}}(C_{\text{Lip}}+C_{\text{fmax}})\tau+\beta e^{6C_{\text{Lip}}}(C_{\text{Lip}}+C_{\text{fmax}})(2\tau+3\rho)2\tau+C_{\text{fmax}}e^{6C_{\text{Lip}}}\tau \\
&\leq 18e^{6C_{\text{Lip}}}(1+\beta)(C_{\text{Lip}}+C_{\text{fmax}})\tau.
\end{aligned}
$$

which completes the proof of the first inequality in the lemma. Next, we prove the second inequality in the lemma following the same method.

$$
\begin{aligned}
&\left|\tilde{\hat{f}}-\hat{f}-\left\langle(1-\beta)\nabla_{\boldsymbol{W}}\hat{f}+\beta\nabla_{\boldsymbol{W}}\hat{f},\widetilde{\boldsymbol{W}}-\boldsymbol{W}\right\rangle\right| \\
&=\left|\left\langle\widetilde{\boldsymbol{w}}_L,\tilde{\hat{\boldsymbol{f}}}_{L-1}\right\rangle-\left\langle\boldsymbol{w}_L,\hat{\boldsymbol{f}}_{L-1}\right\rangle-\sum_{l=1}^{L}\left\langle(1-\beta)\nabla_{\boldsymbol{W}_l}f+\beta\nabla_{\boldsymbol{W}_l}\hat{f},\widetilde{\boldsymbol{W}}_l-\boldsymbol{W}_l\right\rangle\right| \\
&=\left|\left\langle\widetilde{\boldsymbol{w}}_L,\tilde{\hat{\boldsymbol{f}}}_{L-1}-\hat{\boldsymbol{f}}_{L-1}\right\rangle+\left\langle\hat{\boldsymbol{f}}_{L-1},\widetilde{\boldsymbol{w}}_L-\boldsymbol{w}_L\right\rangle-\sum_{l=1}^{L-1}\left\langle(1-\beta)\nabla_{\boldsymbol{W}_l}f+\beta\nabla_{\boldsymbol{W}_l}\hat{f},\widetilde{\boldsymbol{W}}_l-\boldsymbol{W}_l\right\rangle\right. \\
&\hspace{6cm}\left.-\left\langle(1-\beta)\boldsymbol{f}_{L-1}+\beta\hat{\boldsymbol{f}}_{L-1},\widetilde{\boldsymbol{w}}_L-\boldsymbol{w}_L\right\rangle\right| \\
&\leq \|\widetilde{\boldsymbol{w}}_L\|_2\left\|\tilde{\hat{\boldsymbol{f}}}_{L-1}-\hat{\boldsymbol{f}}_{L-1}\right\|_2+(1-\beta)\left\|\boldsymbol{f}_{L-1}-\hat{\boldsymbol{f}}_{L-1}\right\|_2\|\widetilde{\boldsymbol{w}}_L-\boldsymbol{w}_L\|_2 \\
&\hspace{3cm}+\left|\sum_{l=1}^{L-1}\left\langle(1-\beta)\nabla_{\boldsymbol{W}_l}f,\widetilde{\boldsymbol{W}}_l-\boldsymbol{W}_l\right\rangle\right|+\left|\sum_{l=1}^{L-1}\left\langle\beta\nabla_{\boldsymbol{W}_l}\hat{f},\widetilde{\boldsymbol{W}}_l-\boldsymbol{W}_l\right\rangle\right| \\
&\leq (3+\tau)2e^{6C_{\text{Lip}}}(C_{\text{Lip}}+C_{\text{fmax}})\tau+(1-\beta)e^{6C_{\text{Lip}}}(C_{\text{Lip}}+C_{\text{fmax}})(2\tau+3\rho)2\tau+C_{\text{fmax}}e^{6C_{\text{Lip}}}\tau \\
&\leq 18e^{6C_{\text{Lip}}}(2-\beta)(C_{\text{Lip}}+C_{\text{fmax}})\tau,
\end{aligned}
\tag{10}
$$

$$\square$$

**Lemma 7.** *There exists an absolute constant $C_1$ such that, for any $\epsilon>0$ and any $\boldsymbol{W},\widetilde{\boldsymbol{W}}\in\mathcal{B}(\boldsymbol{W}^{[1]},\tau)$ with $\tau\leq C_1\epsilon(\beta+1)^{-1}(C_{\text{Lip}}+C_{\text{fmax}})^{-1}e^{-6C_{\text{Lip}}}$, with probability at least $1-2L\exp(-m/2)-\delta$, when the width satisfies $m=\Omega(N/\delta)$ and $\rho\leq 1$, one has:*

$$
\mathcal{L}(\boldsymbol{x},\widetilde{\boldsymbol{W}})\geq\mathcal{L}(\boldsymbol{x},\boldsymbol{W})+\left\langle(1-\beta)\nabla_{\boldsymbol{W}}\mathcal{L}(\boldsymbol{x},\boldsymbol{W})+\beta\nabla_{\boldsymbol{W}}\mathcal{L}(\mathcal{A}_\rho(\boldsymbol{x},\boldsymbol{W}),\boldsymbol{W}),\widetilde{\boldsymbol{W}}-\boldsymbol{W}\right\rangle-\epsilon,
$$

$$
\mathcal{L}(\mathcal{A}_\rho(\boldsymbol{x},\boldsymbol{W}),\widetilde{\boldsymbol{W}})\geq\mathcal{L}(\mathcal{A}_\rho(\boldsymbol{x},\boldsymbol{W}),\boldsymbol{W})+\left\langle(1-\beta)\nabla_{\boldsymbol{W}}\mathcal{L}(\boldsymbol{x},\boldsymbol{W})+\beta\nabla_{\boldsymbol{W}}\mathcal{L}(\mathcal{A}_\rho(\boldsymbol{x},\boldsymbol{W}),\boldsymbol{W}),\widetilde{\boldsymbol{W}}-\boldsymbol{W}\right\rangle-\epsilon.
$$

*Proof.* Firstly, we will prove the first inequality. Recall that the cross-entropy loss is written as $\ell(z) = \log(1 + \exp(-z))$ and we denote by $\mathcal{L}(\boldsymbol{x}, \boldsymbol{W}) := \ell[y \cdot f(\boldsymbol{x}, \boldsymbol{W})]$. Then one has:

$$
\begin{aligned}
&\mathcal{L}(\boldsymbol{x}, \widetilde{\boldsymbol{W}}) - \mathcal{L}(\boldsymbol{x}, \boldsymbol{W}) \\
&= \ell[y\widetilde{f}] - \ell[yf] \\
&\geq \ell'[yf] \cdot y \cdot (\widetilde{f} - f) \quad \text{(By convexity of } \ell(z)). \\
&\geq \ell'[yf] \cdot y \cdot \left\langle (1-\beta)\nabla f, \widetilde{\boldsymbol{W}} - \boldsymbol{W} \right\rangle + \ell'[y\hat{f}] \cdot y \cdot \left\langle \beta \nabla \hat{f}, \widetilde{\boldsymbol{W}} - \boldsymbol{W} \right\rangle - \kappa_1 \\
&= \left\langle (1-\beta)\nabla_{\boldsymbol{W}}\mathcal{L}(\boldsymbol{x}, \boldsymbol{W}) + \beta \nabla_{\boldsymbol{W}}\mathcal{L}(\mathcal{A}_\rho(\boldsymbol{x}, \boldsymbol{W}), \boldsymbol{W}), \widetilde{\boldsymbol{W}} - \boldsymbol{W} \right\rangle - \kappa_1 \quad \text{(By chain rule)},
\end{aligned}
$$

where we define:

$$
\kappa_1 := \left| \ell'[yf] \cdot y \cdot (\widetilde{f} - f) - \ell'[yf] \cdot y \cdot \left\langle (1-\beta)\nabla f, \widetilde{\boldsymbol{W}} - \boldsymbol{W} \right\rangle - \ell'[y\hat{f}] \cdot y \cdot \left\langle \beta \nabla \hat{f}, \widetilde{\boldsymbol{W}} - \boldsymbol{W} \right\rangle \right|.
$$

Thus, it suffices to show that $\kappa_1$ can be upper bounded by $\epsilon$:

$$
\begin{aligned}
\kappa_1 &\leq \left| \ell'[yf] \cdot y \left\{ \widetilde{f} - f - \left\langle (1-\beta)\nabla_{\boldsymbol{W}} f + \beta \nabla_{\boldsymbol{W}} \hat{f}, \boldsymbol{W}), \widetilde{\boldsymbol{W}} - \boldsymbol{W} \right\rangle \right\} \right| + \left| \left\{ \ell'[yf] \cdot y - \ell'[y\hat{f}] \cdot y \right\} \beta \left\langle \nabla_{\boldsymbol{W}} \hat{f}, \widetilde{\boldsymbol{W}} - \boldsymbol{W} \right\rangle \right| \\
&\leq 18 e^{6C_{\text{Lip}}}(1+\beta)(C_{\text{Lip}} + C_{\text{fmax}})\tau + 2\beta \left| \left\langle \nabla_{\boldsymbol{W}} \hat{f}, \widetilde{\boldsymbol{W}} - \boldsymbol{W} \right\rangle \right| \\
&\leq 18 e^{6C_{\text{Lip}}}(1+\beta)(C_{\text{Lip}} + C_{\text{fmax}})\tau + 2\beta \left| \sum_{l=1}^{L-1} \left\langle \nabla_{\boldsymbol{W}_l} f(\boldsymbol{x}, \boldsymbol{W}), \widetilde{\boldsymbol{W}}_l - \boldsymbol{W}_l \right\rangle \right| + 2\beta |\langle \boldsymbol{f}_{L-1}, \widetilde{\boldsymbol{w}_L} - \boldsymbol{w}_L \rangle| \\
&\leq 18 e^{6C_{\text{Lip}}}(1+\beta)(C_{\text{Lip}} + C_{\text{fmax}})\tau + 2\beta C_{\text{fmax}} e^{6C_{\text{Lip}}}\tau + 4\beta C_{\text{fmax}}\tau \\
&\leq 24(1+\beta)e^{6C_{\text{Lip}}}(C_{\text{Lip}} + C_{\text{fmax}})\tau \\
&\leq \epsilon,
\end{aligned}
\tag{11}
$$

where the first and the third inequality is by triangle inequality, the second inequality is by Lemma 6 and the fact that $|\ell'[yf(\boldsymbol{x}, \boldsymbol{W})] \cdot y| \leq 1$, , the fourth inequality follows the same proof as in Eqs. (8) and (9), and the last inequality is by the condition that if $\tau \leq C_2 \epsilon (1+\beta)^{-1}(C_{\text{Lip}} + C_{\text{fmax}})^{-1} e^{-6C_{\text{Lip}}}$ for some absolute constant $C_2$.

Now we will prove the second inequality of the lemma.

$$
\begin{aligned}
&\mathcal{L}(\mathcal{A}_\rho(\boldsymbol{x}, \boldsymbol{W}), \widetilde{\boldsymbol{W}}) - \mathcal{L}(\mathcal{A}_\rho(\boldsymbol{x}, \boldsymbol{W}), \boldsymbol{W}) \\
&= \ell[y\widetilde{\hat{f}}] - \ell[y\hat{f}] \\
&\geq \ell'[y\hat{f}] \cdot y \cdot (\widetilde{\hat{f}} - \hat{f}) \quad \text{(By convexity of } \ell(z)). \\
&\geq \ell'[yf] \cdot y \cdot \left\langle (1-\beta)\nabla f, \widetilde{\boldsymbol{W}} - \boldsymbol{W} \right\rangle + \ell'[y\hat{f}] \cdot y \cdot \left\langle \beta \nabla \hat{f}, \widetilde{\boldsymbol{W}} - \boldsymbol{W} \right\rangle - \kappa_2 \\
&= \left\langle (1-\beta)\nabla_{\boldsymbol{W}}\mathcal{L}(\boldsymbol{x}, \boldsymbol{W}) + \beta \nabla_{\boldsymbol{W}}\mathcal{L}(\mathcal{A}_\rho(\boldsymbol{x}, \boldsymbol{W}), \boldsymbol{W}), \widetilde{\boldsymbol{W}} - \boldsymbol{W} \right\rangle - \kappa_2 \quad \text{(By chain rule)},
\end{aligned}
$$

where we define:

$$
\kappa_2 := \left| \ell'[y\hat{f}] \cdot y \cdot (\widetilde{\hat{f}} - \hat{f}) - \ell'[yf] \cdot y \cdot \left\langle (1-\beta)\nabla f, \widetilde{\boldsymbol{W}} - \boldsymbol{W} \right\rangle - \ell'[y\hat{f}] \cdot y \cdot \left\langle \beta \nabla \hat{f}, \widetilde{\boldsymbol{W}} - \boldsymbol{W} \right\rangle \right|.
$$

Thus, it suffices to show that $\kappa_2$ can be upper bounded by $\epsilon$. Following by the same method in Eq. (11) with Lemma 6, we have:

$$
\begin{aligned}
\kappa_2 &\leq 18 e^{6C_{\text{Lip}}}(2-\beta)(C_{\text{Lip}} + C_{\text{fmax}})\tau + 2\beta C_{\text{fmax}} e^{6C_{\text{Lip}}}\tau + 4\beta C_{\text{fmax}}\tau \\
&\leq 12(3-\beta)e^{6C_{\text{Lip}}}(C_{\text{Lip}} + C_{\text{fmax}})\tau \leq \epsilon,
\end{aligned}
$$

where the last inequality is by the condition that if $\tau \leq C_3 \epsilon (3-\beta)^{-1}(C_{\text{Lip}} + C_{\text{fmax}})^{-1} e^{-6C_{\text{Lip}}}$ for some absolute constant $C_3$. Lastly, setting $C_1 = \max\{C_2, C_3\}$ and noting that $(3-\beta)^{-1} < (1+\beta)^{-1}$ completes the proof. $\qquad\square$

**Lemma 8.** *For any $\epsilon, \delta, R > 0$ such that $\delta' := 2L\exp(-m/2) + \delta \in (0,1)$, there exists:*

$$m^\star = \mathcal{O}(\mathrm{poly}(R, L, C_{\mathrm{Lip}}, \beta)) \cdot \epsilon^{-2} e^{12C_{\mathrm{Lip}}} \log(1/\delta')$$

*such that if $m \geq m^\star$, then for any $W^\star \in \mathcal{B}(W^{[1]}, Rm^{-1/2})$, under the following choice of step-size $\gamma = \nu\epsilon/[C_{\mathrm{Lip}} C_{\mathrm{fmax}}^2 mLe^{12C_{\mathrm{Lip}}}]$ and $N = L^2 R^2 C_{\mathrm{Lip}} C_{\mathrm{fmax}}^2 e^{12C_{\mathrm{Lip}}}/(2\varepsilon^2\nu)$ for some small enough absolute constant $\nu$, the cumulative loss can be upper bounded with probability at least $1 - \delta'$ by:*

$$\frac{1}{N}\sum_{i=1}^{N}\mathcal{L}(\boldsymbol{x}_i, \boldsymbol{W}^{[i]}) \leq \frac{1}{N}\sum_{i=1}^{N}\mathcal{L}(\boldsymbol{x}_i, \boldsymbol{W}^\star) + 3\epsilon,$$

$$\frac{1}{N}\sum_{i=1}^{N}\mathcal{L}(\mathcal{A}_\rho(\boldsymbol{x}_i, \boldsymbol{W}^{[i]}), \boldsymbol{W}^{[i]}) \leq \frac{1}{N}\sum_{i=1}^{N}\mathcal{L}(\mathcal{A}_\rho(\boldsymbol{x}_i, \boldsymbol{W}^{[i]}), \boldsymbol{W}^\star) + 3\epsilon.$$

*Proof.* We set $\tau = C_1\epsilon(1 + \beta)^{-1}(C_{\mathrm{Lip}} + C_{\mathrm{fmax}})^{-1}e^{-6C_{\mathrm{Lip}}}$ where $C_1$ is a small enough absolute constant so that the requirements on $\tau$ in Lemmas 6 and 7 can be satisfied. Let $Rm^{-1/2} \leq \tau$, then we obtain the condition for $W^\star \in \mathcal{B}(W^{[1]}, \tau)$, i.e., $m \geq R^2 C_1^{-2}\epsilon^{-2}(1+\beta)^2(C_{\mathrm{Lip}} + C_{\mathrm{fmax}})^2 e^{12C_{\mathrm{Lip}}}$. We now show that $W^{[1]}, ..., W^{[N]}$ are inside $\mathcal{B}(W^{[1]}, \tau)$ as well. The proof follows by induction. Clearly, we have $W^{[1]} \in \mathcal{B}(W^{[1]}, \tau)$. Suppose that $W^{[1]}, ..., W^{[i]} \in \mathcal{B}(W^{[1]}, \tau)$, then with probability at least $1 - \delta'$, we have:

$$\left\|W_l^{[i+1]} - W_l^{[1]}\right\|_{\mathrm{F}} \leq \sum_{j=1}^{i}\left\|W_l^{[j+1]} - W_l^{[j]}\right\|_{\mathrm{F}}$$

$$= \sum_{j=1}^{i}\gamma\left\|(1-\beta)\nabla_{\boldsymbol{W}_l}\mathcal{L}(\boldsymbol{x}_j, \boldsymbol{W}^{[j]}) + \beta\nabla_{\boldsymbol{W}_l}\mathcal{L}(\mathcal{A}_\rho(\boldsymbol{x}_j, \boldsymbol{W}^{[j]}), \boldsymbol{W}^{[j]})\right\|_{\mathrm{F}}$$

$$\leq \gamma(1-\beta)N\left\|\nabla_{\boldsymbol{W}_l}\mathcal{L}(\boldsymbol{x}_j, \boldsymbol{W}^{[j]})\right\|_{\mathrm{F}} + \gamma\beta N\left\|\nabla_{\boldsymbol{W}_l}\mathcal{L}(\mathcal{A}_\rho(\boldsymbol{x}_j, \boldsymbol{W}^{[j]}), \boldsymbol{W}^{[j]})\right\|_{\mathrm{F}}$$

$$\leq \gamma NC_{\mathrm{Lip}}\|\boldsymbol{w}_L\|_2\|\boldsymbol{f}_{l-1}\|_2\prod_{r=l+1}^{L-1}\left(\frac{C_{\mathrm{Lip}}}{L}\|\boldsymbol{W}_r\|_2 + 1\right)\left\|\widetilde{\boldsymbol{W}}_l - \boldsymbol{W}_l\right\|_{\mathrm{F}}$$

$$\leq \gamma NC_{\mathrm{Lip}}(3+\tau)C_{\mathrm{fmax}}\left(\frac{C_{\mathrm{Lip}}}{L}(3+\tau) + 1\right)^{L-l-1}$$

$$\leq 6\gamma NC_{\mathrm{Lip}}C_{\mathrm{fmax}}e^{6C_{\mathrm{Lip}}}.$$

Plugging in our parameter choice for $\gamma$ and $N$ leads to:

$$\left\|W_l^{[i+1]} - W_l^{[1]}\right\|_{\mathrm{F}} \leq 3C_{\mathrm{Lip}}C_{\mathrm{fmax}}e^{6C_{\mathrm{Lip}}}LR^2\epsilon^{-1}m^{-1} \leq \tau,$$

where the last inequality holds as long as $m \geq 3C_{\mathrm{Lip}}C_{\mathrm{fmax}}e^{12C_{\mathrm{Lip}}}LR^2C_1^{-1}\epsilon^{-2}$. Therefore by induction we see that $W^{[1]}, ..., W^{[N]} \in \mathcal{B}(W^{[1]}, \tau)$. Now, we are ready to prove the first inequality in the lemma. We provide an upper bound for the cumulative loss as follows:

$$\mathcal{L}(\boldsymbol{x}_i, \boldsymbol{W}^{[i]}) - \mathcal{L}(\boldsymbol{x}_i, \boldsymbol{W}^\star)$$

$$\leq \left\langle(1-\beta)\nabla_{\boldsymbol{W}}\mathcal{L}(\boldsymbol{x}_i, \boldsymbol{W}^{[i]}) + \beta\nabla_{\boldsymbol{W}}\mathcal{L}(\mathcal{A}_\rho(\boldsymbol{x}_i, \boldsymbol{W}^{[i]}), \boldsymbol{W}^{[i]}), \boldsymbol{W}^{[i]} - \boldsymbol{W}^\star\right\rangle + \epsilon \quad \text{(By Lemma 7)}$$

$$= \frac{\left\langle\boldsymbol{W}^{[i]} - \boldsymbol{W}^{[i+1]}, \boldsymbol{W}^{[i]} - \boldsymbol{W}^\star\right\rangle}{\gamma} + \epsilon$$

$$= \frac{\|\boldsymbol{W}^{[i]} - \boldsymbol{W}^{[i+1]}\|_{\mathrm{F}}^2 + \|\boldsymbol{W}^{[i]} - \boldsymbol{W}_l^\star\|_{\mathrm{F}}^2 - \|\boldsymbol{W}^{[i+1]} - \boldsymbol{W}^\star\|_{\mathrm{F}}^2}{2\gamma} + \epsilon$$

$$= \frac{\|\boldsymbol{W}^{[i]} - \boldsymbol{W}^\star\|_{\mathrm{F}}^2 - \|\boldsymbol{W}^{[i+1]} - \boldsymbol{W}^\star\|_{\mathrm{F}}^2 + \gamma^2\left\|(1-\beta)\nabla_{\boldsymbol{W}}\mathcal{L}(\boldsymbol{x}_i, \boldsymbol{W}^{[i]}) + \beta\nabla_{\boldsymbol{W}}\mathcal{L}(\mathcal{A}_\rho(\boldsymbol{x}_i, \boldsymbol{W}^{[i]}), \boldsymbol{W}^{[i]})\right\|_{\mathrm{F}}^2}{2\gamma} + \epsilon$$

$$\leq \frac{\|\boldsymbol{W}^{[i]} - \boldsymbol{W}^\star\|_{\mathrm{F}}^2 - \|\boldsymbol{W}^{[i+1]} - \boldsymbol{W}^\star\|_{\mathrm{F}}^2}{2\gamma} + \frac{6^2\gamma^2 C_{\mathrm{Lip}}^2 C_{\mathrm{fmax}}^2 e^{12C_{\mathrm{Lip}}}}{2\gamma} + \epsilon.$$

Telescoping over $i = 1,...,N$, we obtain:

$$\frac{1}{N}\sum_{i=1}^{N}\mathcal{L}(\boldsymbol{x}_i,\boldsymbol{W}^{[i]})$$

$$\leq \frac{1}{N}\sum_{i=1}^{N}\mathcal{L}(\boldsymbol{x}_i,\boldsymbol{W}^{\star}) + \frac{\|\boldsymbol{W}^{(1)}-\boldsymbol{W}^{\star}\|_{\mathrm{F}}^2}{2N\gamma} + 18\gamma C_{\mathrm{Lip}}^2 C_{\mathrm{fmax}}^2 e^{12C_{\mathrm{Lip}}} + \epsilon$$

$$\leq \frac{1}{N}\sum_{i=1}^{N}\mathcal{L}(\boldsymbol{x}_i,\boldsymbol{W}^{\star}) + \frac{LR^2}{2\gamma mN} + 18\gamma C_{\mathrm{Lip}}^2 C_{\mathrm{fmax}}^2 e^{12C_{\mathrm{Lip}}} + \epsilon$$

$$\leq \frac{1}{N}\sum_{i=1}^{N}\mathcal{L}(\boldsymbol{x}_i,\boldsymbol{W}^{\star}) + 3\epsilon,$$

where in the first inequality we simply remove the term $-\|\boldsymbol{W}^{[N+1]}-\boldsymbol{W}^{\star}\|_{\mathrm{F}}^2/(2\gamma)$ to obtain an upper bound, the second inequality follows by the assumption that $\boldsymbol{W}^{\star} \in \mathcal{B}(\boldsymbol{W}^{[1]},Rm^{-1/2})$, the third inequality is by the parameter choice of $\gamma$ and $N$. Lastly, we denote $\delta' := 2L\exp(-m/2)+\delta \in (0,1)$, which requires $m \geq \tilde{\Omega}(1)$ satisfied by taking $m = \Omega(N/\delta)$. One can follow the same procedure to prove the second inequality in the lemma that is based on Lemma 7. □

## C.2 PROOF OF THEOREM 1

*Proof.* Since $\bar{\boldsymbol{W}}$ is uniformly sampled from $\{\boldsymbol{W}^{[1]},\cdots,\boldsymbol{W}^{[N]}\}$, by Hoeffding's inequality, with probability at least $1-\delta''$:

$$\mathbb{E}_{\bar{\boldsymbol{W}}}(\mathcal{L}_{0-1}^{\mathrm{clean}}(\bar{\boldsymbol{W}})) \leq \frac{1}{N}\sum_{i=1}^{N}\mathbb{1}\Big[y_i \cdot f(\boldsymbol{x}_i,\boldsymbol{W}^{[i]}) < 0\Big] + \sqrt{\frac{2\log(1/\delta'')}{N}}. \tag{12}$$

Since the cross-entropy loss $\ell(z) = \log(1+\exp(-z))$ satisfies $\mathbb{1}\{z \leq 0\} \leq 4\ell(z)$, we have:

$$\mathbb{1}\Big[y_i \cdot f(\boldsymbol{x}_i,\boldsymbol{W}^{[i]}) < 0\Big] \leq 4\mathcal{L}(\boldsymbol{x}_i,\boldsymbol{W}^{[i]}), \tag{13}$$

with $\mathcal{L}(\boldsymbol{x}_i,\boldsymbol{W}^{[i]}) := \ell(y_i \cdot f(\boldsymbol{x}_i,\boldsymbol{W}^{[i]}))$. Next, setting $\epsilon = LR\sqrt{C_{\mathrm{Lip}}}C_{\mathrm{fmax}}e^{6C_{\mathrm{Lip}}}/\sqrt{2\nu N}$ in Lemma 8 leads to step-size $\gamma = \sqrt{\nu}R/(\sqrt{2C_{\mathrm{Lip}}}C_{\mathrm{fmax}}e^{6C_{\mathrm{Lip}}}m\sqrt{N})$, then by combining it with Eqs. (12) and (13), with probability at least $1-\delta'-\delta''$, we have:

$$\mathbb{E}_{\bar{\boldsymbol{W}}}\big(\mathcal{L}_{0-1}^{\mathrm{clean}}(\bar{\boldsymbol{W}})\big) \leq \frac{4}{N}\sum_{i=1}^{N}(\mathcal{L}(\boldsymbol{x}_i,\boldsymbol{W}^{\star})) + \frac{12}{\sqrt{2\nu}} \cdot \frac{LR}{\sqrt{N}} + \sqrt{\frac{2\log(1/\delta'')}{N}}, \tag{14}$$

for all $\boldsymbol{W}^{\star} \in \mathcal{B}(\boldsymbol{W}^{[1]},Rm^{-1/2})$.

Define the linearized network around initialization $\boldsymbol{W}^{[1]}$ as

$$F_{\boldsymbol{W}^{[1]},\boldsymbol{W}^{\star}}(\boldsymbol{x}) := f(\boldsymbol{x},\boldsymbol{W}^{[1]}) + \langle(1-\beta)\nabla f_{\boldsymbol{W}}(\boldsymbol{x},\boldsymbol{W}^{[1]}) + \beta\nabla f_{\boldsymbol{W}}(\mathcal{A}_\rho^*(\boldsymbol{x},\boldsymbol{W}^{[1]}),\boldsymbol{W}^{[1]}),\boldsymbol{W}^{\star}-\boldsymbol{W}^{[1]}\rangle. \tag{15}$$

We now compare the loss induced by the original network with its linearized network:

$$\mathcal{L}(\boldsymbol{x}_i,\boldsymbol{W}^{\star}) - \ell(y_i \cdot F_{\boldsymbol{W}^{[1]},\boldsymbol{W}^{\star}}(\boldsymbol{x}_i)) = \ell(y_i \cdot f(\boldsymbol{x}_i,\boldsymbol{W}^{\star})) - \ell(y_i \cdot F_{\boldsymbol{W}^{[1]},\boldsymbol{W}^{\star}}(\boldsymbol{x}_i))$$

$$\leq y_i(f(\boldsymbol{x}_i,\boldsymbol{W}^{\star}) - F_{\boldsymbol{W}^{[1]},\boldsymbol{W}^{\star}}(\boldsymbol{x}_i)) \leq 18e^{6C_{\mathrm{Lip}}}(1+\beta)(C_{\mathrm{Lip}}+C_{\mathrm{fmax}})Rm^{-1/2} \leq LRN^{-1/2},$$

where the first inequality is by the 1-Lipschitz continuity of $\ell$, the second inequality is by Lemma 6 with $Rm^{-1/2} \leq 3$, i.e., $m \geq R^2/9$, the third inequality holds when $m \geq 18^2 e^{12C_{\mathrm{Lip}}}(1+\beta)^2(C_{\mathrm{Lip}}+C_{\mathrm{fmax}})^2 NL^{-2}$. Plugging the inequality above back to Eq. (14), we obtain:

$$\mathbb{E}_{\bar{\boldsymbol{W}}}\big(L_{\mathcal{D}}^{0-1}(\bar{\boldsymbol{W}})\big) \leq \frac{4}{N}\sum_{i=1}^{N}\ell(y_i \cdot F_{\boldsymbol{W}^{[1]},\boldsymbol{W}^{\star}}(\boldsymbol{x}_i)) + (\frac{12}{\sqrt{2\nu}}+1) \cdot \frac{LR}{\sqrt{N}} + \sqrt{\frac{2\log(1/\delta'')}{N}}. \tag{16}$$

Next, we will upper bound the RHS of the inequality above. For cross-entropy loss we have: $\ell(z) \leq N^{-1/2}$ for $z \geq B := \log\{1/[\exp(N^{-1/2})-1]\} = \mathcal{O}(\log N)$. We define:

$$B' = \max_{i \in [N]}|f(\boldsymbol{x}_i,\boldsymbol{W}^{[1]})|, \quad \boldsymbol{y}' = (B+B') \cdot \boldsymbol{y}.$$

Then for any $i \in [N]$, we have:
$$y_i \cdot [y_i' + f(\boldsymbol{x}_i, \boldsymbol{W}^{[1]})] = y_i \cdot [(B + B')y_i + f(\boldsymbol{x}_i, \boldsymbol{W}^{[1]})] \geq B + B' - B' = B.$$
As a result, we have:
$$\ell\{y_i \cdot (y_i' + f(\boldsymbol{x}_i, \boldsymbol{W}^{[1]}))\} \leq N^{-1/2}, i \in [N].$$
Denote by $\boldsymbol{J}_{\text{all}} := (1 - \beta)\boldsymbol{J} + \beta\hat{\boldsymbol{J}}$, where
$$\boldsymbol{J} = m^{-1/2} \cdot [\text{vec}(\nabla f_{\boldsymbol{W}}(\boldsymbol{x}_1, \boldsymbol{W}^{[1]})), ..., \text{vec}(\nabla f_{\boldsymbol{W}}(\boldsymbol{x}_N, \boldsymbol{W}^{[1]}))],$$
$$\hat{\boldsymbol{J}} = m^{-1/2} \cdot [\text{vec}(\nabla f_{\boldsymbol{W}}(\mathcal{A}_\rho^*(\boldsymbol{x}_1, \boldsymbol{W}^{[1]}), \boldsymbol{W}^{[1]})), ..., \text{vec}\nabla f_{\boldsymbol{W}}(\mathcal{A}_\rho^*(\boldsymbol{x}_N, \boldsymbol{W}^{[1]}), \boldsymbol{W}^{[1]}))].$$

Let $\boldsymbol{P}\boldsymbol{\Lambda}\boldsymbol{Q}^\top$ be the singular value decomposition of $\boldsymbol{J}_{\text{all}}$, where $\boldsymbol{P} \in \mathbb{R}^{(md+(L-2)m^2+m) \times N}, \boldsymbol{Q} \in \mathbb{R}^{N \times N}, \boldsymbol{\Lambda} \in \mathbb{R}^{N \times N}$. Let us define $\boldsymbol{w}_{\text{vec}} := \boldsymbol{P}\boldsymbol{\Lambda}^{-1}\boldsymbol{Q}^\top \boldsymbol{y}'$, then we have:
$$\boldsymbol{J}_{\text{all}}^\top \boldsymbol{w}_{\text{vec}} = \boldsymbol{Q}\boldsymbol{\Lambda}\boldsymbol{P}^\top \boldsymbol{P}\boldsymbol{\Lambda}^{-1}\boldsymbol{Q}^\top \boldsymbol{y}' = \boldsymbol{y}',$$
which implies $((1 - \beta)\boldsymbol{J}^\top + \beta\hat{\boldsymbol{J}}^\top)\boldsymbol{w}_{\text{vec}} = \boldsymbol{y}'$. Moreover, we have:
$$\begin{aligned} \|\boldsymbol{w}_{\text{vec}}\|_2^2 &= \left\|\boldsymbol{P}\boldsymbol{\Lambda}^{-1}\boldsymbol{Q}^\top \boldsymbol{y}'\right\|_2^2 = \boldsymbol{y}'^\top \boldsymbol{Q}\boldsymbol{\Lambda}^{-2}\boldsymbol{Q}^\top \boldsymbol{y}' = \boldsymbol{y}'^\top (\boldsymbol{J}_{\text{all}}^\top \boldsymbol{J}_{\text{all}})^{-1}\boldsymbol{y}'. \\ &= \boldsymbol{y}'^\top [(\boldsymbol{J}_{\text{all}}^\top \boldsymbol{J}_{\text{all}})^{-1} - (\boldsymbol{K}_{\text{all}})^{-1}]\boldsymbol{y}' + \boldsymbol{y}'^\top (\boldsymbol{K}_{\text{all}})^{-1}\boldsymbol{y}' \\ &\leq N(B + B')^2 \|(\boldsymbol{J}_{\text{all}}^\top \boldsymbol{J}_{\text{all}})^{-1} - (\boldsymbol{K}_{\text{all}})^{-1}\|_2 + (B + B')^2 \boldsymbol{y}'^\top (\boldsymbol{K}_{\text{all}})^{-1}\boldsymbol{y}'. \end{aligned} \tag{17}$$

By Lemma 3.8 in Cao & Gu (2019) and standard matrix perturbation bound, there exists $m^*(\delta, L, N, \lambda_{\min}(\boldsymbol{J}_{\text{all}}), \beta)$, such that, if $m \geq m^*$, then with probability at least $1 - \delta$, $\boldsymbol{J}_{\text{all}}^\top \boldsymbol{J}_{\text{all}}$ is positive-definite and
$$\|(\boldsymbol{J}_{\text{all}}^\top \boldsymbol{J}_{\text{all}})^{-1} - \boldsymbol{K}_{\text{all}}^{-1}\|_2 \leq \frac{\boldsymbol{y}^\top \boldsymbol{K}_{\text{all}}^{-1}\boldsymbol{y}}{N}.$$

Therefore, Eq. (17) can be further upper bounded by: $\|\boldsymbol{w}_{\text{vec}}\|_2^2 \leq \tilde{\mathcal{O}}(\boldsymbol{y}^\top \boldsymbol{K}_{\text{all}}^{-1}\boldsymbol{y})$. Plugging it back Eq. (16), with probability at least $1 - \delta - \delta' - \delta''$, we obtain:
$$\begin{aligned} \mathbb{E}_{\bar{\boldsymbol{W}}}\big(\mathcal{L}_{\mathcal{D}}^{0-1}(\bar{\boldsymbol{W}})\big) &\leq \frac{4}{N}\sum_{i=1}^N \ell(B) + (\frac{12}{\sqrt{2\nu}} + 1) \cdot \frac{L\|\boldsymbol{w}_{\text{vec}}\|_2}{\sqrt{N}} + \sqrt{\frac{\log(1/\delta'')}{N}} \\ &\leq \tilde{\mathcal{O}}\left(\sqrt{\frac{L^2\boldsymbol{y}^\top \boldsymbol{K}_{\text{all}}^{-1}\boldsymbol{y}}{N}}\right) + \mathcal{O}\left(\sqrt{\frac{\log(1/\delta'')}{N}}\right), \end{aligned}$$
which finishes the proof for generalization guarantees on clean accuracy.

In the next, we present the proof for generalization guarantees on robust accuracy. The proof technique is the same as that of clean accuracy.

Since $\bar{W}$ is uniformly sampled from $\{\boldsymbol{W}^{[1]}, \cdots, \boldsymbol{W}^{[N]}\}$, by Hoeffding's inequality, with probability at least $1 - \delta''$:
$$\mathbb{E}_{\bar{\boldsymbol{W}}}(\mathcal{L}_{0-1}^{\text{robust}}(\bar{\boldsymbol{W}})) \leq \frac{1}{N}\sum_{i=1}^N \mathbb{1}\Big[y_i \cdot f(\mathcal{A}_\rho(\boldsymbol{x}_i, \boldsymbol{W}^{[i]}), \boldsymbol{W}^{[i]}) < 0\Big] + \sqrt{\frac{2\log(1/\delta'')}{N}}. \tag{18}$$

Since the cross-entropy loss $\ell(z) = \log(1 + \exp(-z))$ satisfies $\mathbb{1}\{z \leq 0\} \leq 4\ell(z)$, we have:
$$\mathbb{1}\Big[y_i \cdot f(\mathcal{A}_\rho(\boldsymbol{x}_i, \boldsymbol{W}^{[i]}), \boldsymbol{W}^{[i]}) < 0\Big] \leq 4\mathcal{L}(\mathcal{A}_\rho(\boldsymbol{x}_i, \boldsymbol{W}^{[i]}), \boldsymbol{W}^{[i]}). \tag{19}$$

Next, setting $\epsilon = LR\sqrt{C_{\text{Lip}}}C_{\text{fmax}}e^{6C_{\text{Lip}}}/\sqrt{2\nu N}$ in Lemma 8 leads to step-size $\gamma = \sqrt{\nu}R/(\sqrt{2C_{\text{Lip}}}C_{\text{fmax}}e^{6C_{\text{Lip}}}m\sqrt{N})$, then by combining it with Eqs. (18) and (19), with probability at least $1 - \delta' - \delta''$, we have:
$$\begin{aligned} \mathbb{E}_{\bar{\boldsymbol{W}}}\big(\mathcal{L}_{0-1}^{\text{robust}}(\bar{\boldsymbol{W}})\big) &\leq \frac{4}{N}\sum_{i=1}^N (\mathcal{L}(\mathcal{A}_\rho(\boldsymbol{x}_i, \boldsymbol{W}^{[i]}), \boldsymbol{W}^\star)) + \frac{12}{\sqrt{2\nu}} \cdot \frac{LR}{\sqrt{N}} + \sqrt{\frac{2\log(1/\delta'')}{N}} \\ &\leq \frac{4}{N}\sum_{i=1}^N (\mathcal{L}(\mathcal{A}_\rho^*(\boldsymbol{x}_i, \boldsymbol{W}^\star), \boldsymbol{W}^\star)) + \frac{12}{\sqrt{2\nu}} \cdot \frac{LR}{\sqrt{N}} + \sqrt{\frac{2\log(1/\delta'')}{N}}, \end{aligned} \tag{20}$$

for all $\boldsymbol{W}^\star \in \mathcal{B}(\boldsymbol{W}^{[1]}, Rm^{-1/2})$, where the second inequality is by the definition of the worst-case adversary $\mathcal{A}_\rho^*(\cdot)$.

Next, we compare the loss induced by the original network with its linearized network defined in Eq. (15):

$$
\begin{aligned}
&\mathcal{L}(\mathcal{A}_\rho^*(\boldsymbol{x}_i, \boldsymbol{W}^\star), \boldsymbol{W}^\star) - \ell(y_i \cdot F_{\boldsymbol{W}^{[1]}, \boldsymbol{W}^\star}(\mathcal{A}_\rho^*(\boldsymbol{x}_i, \boldsymbol{W}^{[1]}))) \\
&= \ell(y_i \cdot f(\mathcal{A}_\rho^*(\boldsymbol{x}_i, \boldsymbol{W}^\star), \boldsymbol{W}^\star)) - \ell(y_i \cdot F_{\boldsymbol{W}^{[1]}, \boldsymbol{W}^\star}(\mathcal{A}_\rho^*(\boldsymbol{x}_i, \boldsymbol{W}^{[1]}))) \\
&\leq f(\mathcal{A}_\rho^*(\boldsymbol{x}_i, \boldsymbol{W}^\star), \boldsymbol{W}^\star) - F_{\boldsymbol{W}^{[1]}, \boldsymbol{W}^\star}(\mathcal{A}_\rho^*(\boldsymbol{x}_i, \boldsymbol{W}^{[1]}))) \\
&\leq 18e^{6C_{\mathrm{Lip}}}(1+\beta)(C_{\mathrm{Lip}} + C_{\mathrm{fmax}})(Rm^{-1/2} + 6\rho + 2Rm^{-1/2}\rho) \\
&\leq LRN^{-1/2},
\end{aligned}
$$

where the first inequality is by the 1-Lipschitz continuity of $\ell$, the second inequality is by Lemmas 4 and 6 with $Rm^{-1/2} + 6\rho + 2R\rho m^{-1/2} \leq 3$, i.e., $m \geq R(1+2\rho)/[3(1-2\rho)]$, the last inequality holds when $m \geq 18e^{6C_{\mathrm{Lip}}}(1+\beta)(C_{\mathrm{Lip}} + C_{\mathrm{fmax}})R(1+\rho)^2(LR/N - 18e^{6C_{\mathrm{Lip}}}(1+\beta)(C_{\mathrm{Lip}} + C_{\mathrm{fmax}}))^{-2}$. Plugging the inequality above back to Eq. (20), we obtain:

$$
\mathbb{E}_{\bar{\boldsymbol{W}}}\left(\mathcal{L}_{0-1}^{\mathrm{robust}}(\bar{\boldsymbol{W}})\right) \leq \frac{4}{N}\sum_{i=1}^{N}\ell(y_i \cdot F_{\boldsymbol{W}^{[1]}, \boldsymbol{W}^\star}(\mathcal{A}_\rho^*(\boldsymbol{x}_i, \boldsymbol{W}^{[1]}))) + (\frac{12}{\sqrt{2\nu}}+1)\cdot\frac{LR}{\sqrt{N}} + \sqrt{\frac{2\log(1/\delta'')}{N}}.
\tag{21}
$$

Lastly, noticing that based on Eq. (15):

$$
\begin{aligned}
&F_{\boldsymbol{W}^{[1]}, \boldsymbol{W}^\star}(\mathcal{A}_\rho^*(\boldsymbol{x}_i, \boldsymbol{W}^{[1]})) \\
&= f(\mathcal{A}_\rho^*(\boldsymbol{x}_i, \boldsymbol{W}^{[1]}), \boldsymbol{W}^{[1]}) + \langle (1-\beta)\nabla f_{\boldsymbol{W}}(\mathcal{A}_\rho^*(\boldsymbol{x}_i, \boldsymbol{W}^{[1]}), \boldsymbol{W}^{[1]}) + \beta\nabla f_{\boldsymbol{W}}(\mathcal{A}_\rho^*(\mathcal{A}_\rho^*(\boldsymbol{x}_i, \boldsymbol{W}^{[1]}), \boldsymbol{W}^{[1]}), \boldsymbol{W}^{[1]}), \boldsymbol{W}^\star - \boldsymbol{W}^{[1]} \rangle.
\end{aligned}
$$

Then we can define the Jacobian $\widetilde{\boldsymbol{J}}_{\mathrm{all}} := (1-\beta)\hat{\boldsymbol{J}}_\rho + \beta\hat{\boldsymbol{J}}_{2\rho}$, where

$$
\hat{\boldsymbol{J}}_\rho = m^{-1/2}\cdot[\mathrm{vec}(\nabla f_{\boldsymbol{W}}(\mathcal{A}_\rho^*(\boldsymbol{x}_1, \boldsymbol{W}^{[1]}), \boldsymbol{W}^{[1]})), ..., \mathrm{vec}\nabla f_{\boldsymbol{W}}(\mathcal{A}_\rho^*(\boldsymbol{x}_N, \boldsymbol{W}^{[1]}), \boldsymbol{W}^{[1]}))]
$$

$$
\hat{\boldsymbol{J}}_{2\rho} = m^{-1/2}\cdot[\mathrm{vec}(\nabla f_{\boldsymbol{W}}(\mathcal{A}_\rho^*(\mathcal{A}_\rho^*(\boldsymbol{x}_i, \boldsymbol{W}^{[1]}), \boldsymbol{W}^{[1]}), \boldsymbol{W}^{[1]})), ..., \mathrm{vec}\nabla f_{\boldsymbol{W}}(\mathcal{A}_\rho^*(\mathcal{A}_\rho^*(\boldsymbol{x}_i, \boldsymbol{W}^{[1]}), \boldsymbol{W}^{[1]}), \boldsymbol{W}^{[1]}))].
$$

Lastly, by replacing $\boldsymbol{J}_{\mathrm{all}}$ by $\widetilde{\boldsymbol{J}}_{\mathrm{all}}$ as in the step on clean generalization bound, we can finish the proof.

$\square$

## D   THE LOWER BOUND OF THE MINIMUM EIGENVALUE OF ROBUST NTK

We have shown that the minimum eigenvalue of robust NTK significantly affects both clean and robust generalizations. Hence we provide a lower bound estimation for its minimum eigenvalue.

**Assumption 3.** *We assume the perturbation $\rho < \mathcal{O}(C/d)$. For any data $\langle \boldsymbol{x}_i, \boldsymbol{x}_j \rangle + 2\rho + \rho^2 < 1$, in other words,*

$$
\langle \boldsymbol{x}_i, \boldsymbol{x}_j \rangle \leq 1 - \frac{C}{d} - o(1/d), \forall \boldsymbol{x}_i, \boldsymbol{x}_j,
$$

*where $C$ is some constant independent of the number of data points $N$ and the dimensional of the input feature $d$.*

**Remark:** For example, we can set $C := 2C_{\max}$. Here the $o(1/d)$ means a high-order small term of $1/d$ and can be omitted. This assumption holds for a large $d$ in practice, e.g., $d = 796$ for MNIST dataset and $d = 3072$ for CIFAR10/100 dataset.

**Theorem 2.** *Under Assumption 1 and 3, we have the following lower bound estimation for the minimum eigenvalue of $\hat{\boldsymbol{K}}_\rho$*

$$
\lambda_{\min}(\widetilde{\boldsymbol{K}}_{\mathrm{all}}) \geq 2\mu_r(\sigma_1)^2\left(1 - (N-1)\max_{i \neq j}(|\langle \boldsymbol{x}_i, \boldsymbol{x}_j \rangle| + 2\rho + \rho^2)^r\right).
$$

*where $\mu(\sigma)$ is $r$-Hermite coefficient of the activation at the first layer.*

*Proof.*

$$\lambda_{\min}(\hat{\boldsymbol{K}}_\rho) \geq \lambda_{\min}\left(2\mathbb{E}_{\boldsymbol{w}\sim\mathcal{N}(0,\mathbb{I}_d)}[\sigma_1(\hat{\boldsymbol{X}}\boldsymbol{w})\sigma_1(\hat{\boldsymbol{X}}\boldsymbol{w})^\top]\right)$$

$$= 2\lambda_{\min}\left(\sum_{s=0}^{\infty}\mu_s(\sigma_1)^2\bigcirc_{i=1}^s(\hat{\boldsymbol{X}}\hat{\boldsymbol{X}}^\top)\right)\quad\text{(Nguyen \& Mondelli, 2020, Lemma D.3)}$$

$$\geq 2\mu_r(\sigma_1)^2\lambda_{\min}(\bigcirc_{i=1}^r\hat{\boldsymbol{X}}\hat{\boldsymbol{X}}^\top)\quad\left(\text{taking } r \geq -\log_{\max_{i\neq j}(|\langle\boldsymbol{x}_i,\boldsymbol{x}_j\rangle|+2\rho+\rho^2)}^{N-1}\right)$$

$$\geq 2\mu_r(\sigma_1)^2\left(\min_{i\in[N]}\|\hat{\boldsymbol{x}}_i\|_2^{2r} - (N-1)\max_{i\neq j}|\langle\hat{\boldsymbol{x}}_i,\hat{\boldsymbol{x}}_j\rangle|^r\right)\quad\text{(Gershgorin circle theorem)}$$

$$\geq 2\mu_r(\sigma_1)^2\left(1 - (N-1)\max_{i\neq j}(|\langle\boldsymbol{x}_i,\boldsymbol{x}_j\rangle|+|\langle\boldsymbol{x}_i,\boldsymbol{\Delta}_j\rangle|+|\langle\boldsymbol{x}_j,\boldsymbol{\Delta}_i\rangle|+|\langle\boldsymbol{\Delta}_i,\boldsymbol{\Delta}_j\rangle|)^r\right)$$

$$\geq 2\mu_r(\sigma_1)^2\left(1 - (N-1)\max_{i\neq j}(|\langle\boldsymbol{x}_i,\boldsymbol{x}_j\rangle|+2\rho+\rho^2)^r\right).$$

$\square$

## E  PROOF OF THE GENERALIZATION OF RESIDUAL CNN

In this section, we provide proof for residual CNN. Firstly, we reformulate the network in Appendix E.1. Secondly, we introduce several lemmas in Appendix E.2 in order to show the upper bound for the cumulative loss (Lemma 14). The remaining step is the same as FCNN.

### E.1  REFORMULATION OF THE NETWORK

Firstly, we will rewrite the definition of CNN in Eq. (2) in a way to facilitate the proof. Specifically, we define an operator $\phi_1(\cdot)$ that divides its input into $p$ patches. The dimension of each patch is $kd$. For example, when the size of filter $k=3$, we have:

$$\phi_1(\boldsymbol{X}) = \begin{bmatrix} (\boldsymbol{X}^{(1,0:2)})^\top, & \dots & ,(\boldsymbol{X}^{(1,p-1:p+1)})^\top \\ \dots, & \dots, & \dots \\ (\boldsymbol{X}^{(d,0:2)})^\top, & \dots, & (\boldsymbol{X}^{(d,p-1:p+1)})^\top \end{bmatrix} \in \mathbb{R}^{3d\times p}.$$

Similarly, we define $\phi_l(\boldsymbol{F}_l)\in\mathbb{R}^{\kappa m\times p}$ for the subsequent layers. Then we can re-write the formula of CNN as follows:

$$\boldsymbol{F}_1 = \sigma_1(\boldsymbol{W}_1\phi_1(\boldsymbol{X})),$$
$$\boldsymbol{F}_l = \frac{1}{L}\sigma_l(\boldsymbol{W}_l\phi_l(\boldsymbol{F}_{l-1})) + \alpha_{l-1}\boldsymbol{F}_{l-1},\quad 2\leq l\leq L-1,$$
$$\boldsymbol{F}_L = \langle\boldsymbol{W}_L,\boldsymbol{F}_{L-1}\rangle,$$
$$f(\boldsymbol{X},\boldsymbol{W}) = \boldsymbol{F}_L,$$

where learnable weights are $\boldsymbol{W}_1\in\mathbb{R}^{m\times kd}$, $\boldsymbol{W}_l\in\mathbb{R}^{m\times\kappa m}$, $l=2,\dots,L-1$, and $\boldsymbol{W}_L\in\mathbb{R}^{m\times p}$.

### E.2  SOME AUXILIARY LEMMAS

**Lemma 9** (Upper bound of spectral norms of initial weight)**.** *With probability at least* $1 - 2\exp(-m/2) - 2(L-2)\exp(-\kappa m/2) - 2p\exp(-m/2)$*, the norm of the weight of residual CNN has the following upper bound:*

$$\|\boldsymbol{W}_l^{[1]}\|_2 \leq 3, \text{for } l\in[L-1], \quad \|\boldsymbol{W}_L^{[1]}\|_F \leq 3\sqrt{p}.$$

*Proof.* The bound of $\|\boldsymbol{W}_l^{[1]}\|_2$, for $l\in[L-1]$ can be obtained by directly applying Lemma 2. Regarding $\|\boldsymbol{W}_L^{[1]}\|_F$, note that $\boldsymbol{W}_L^{[1]}\in\mathbb{R}^{m\times p}$, then we bound its norm by Lemma 2 as follows

$$\left\|\boldsymbol{W}_L^{[1]}\right\|_F = \sqrt{\sum_{i=1}^p\left\|(\boldsymbol{W}_L^{[1]})^{(:,i)}\right\|_2^2} \leq 3\sqrt{p},$$

with probability at least $1-2p\exp(-m/2)$.

$\square$

**Lemma 10** (The order of the network output at initialization). *Fix any $l \in [1, L-1]$ and $\boldsymbol{X}$, when the width satisfies $m = \Omega(p^2 N/\delta)$, with probability at least $1-2l\exp(-m/2)-\delta$ over the randomness of $\{\boldsymbol{W}_i^{[1]}\}_{i=1}^l$, we have:*

$$C_{\text{fmin}} \leq \left\| \boldsymbol{F}_l(\boldsymbol{X}, \boldsymbol{W}^{[1]}) \right\|_{\text{F}} \leq C_{\text{fmax}},$$

*where $C_{\text{fmax}}$ and $C_{\text{fmax}}$ are some $\Theta(1)$ constant.*

*Proof.* The result can be obtained by simply applying Lemma 2 for the initial weights and Lemma D.1 in Du et al. (2019a). $\square$

**Lemma 11.** *For $\widetilde{\boldsymbol{W}} \in \mathcal{B}(\boldsymbol{W}^{[1]}, \tau)$ with $\tau \leq 3$, when the width satisfies $m = \Omega(p^2 N/\delta)$ and $\rho \leq 1$, with probability at least $1-2(L-1)\exp(-\kappa m/2)-\delta$, one has:*

$$\left\| \boldsymbol{F}_{L-1}(\boldsymbol{X}, \widetilde{\boldsymbol{W}}) - \boldsymbol{F}_{L-1}(\boldsymbol{X}, \boldsymbol{W}^{[1]}) \right\|_{\text{F}} \leq e^{6C_{\text{Lip}}\sqrt{\kappa}}(\sqrt{\kappa}C_{\text{Lip}} + C_{\text{fmax}})\tau.$$

*Proof.* To simplify the notation, in the following proof, the variable with $\widetilde{\cdot}$ is related to $\widetilde{\boldsymbol{W}}$, and without $\widetilde{\cdot}$ is related to $\boldsymbol{W}^{[1]}$. For the output of the first layer, we have:

$$\left\| \widetilde{\boldsymbol{F}}_1 - \boldsymbol{F}_1 \right\|_{\text{F}} = \left\| \sigma_1(\widetilde{\boldsymbol{W}}_1 \phi_1(\boldsymbol{X})) - \sigma_1(\boldsymbol{W}_1 \phi_1(\boldsymbol{X})) \right\|_{\text{F}} \leq C_{\text{Lip}} \left\| \widetilde{\boldsymbol{W}}_1 - \boldsymbol{W}_1 \right\|_{\text{F}} \|\phi_1(\boldsymbol{X})\|_{\text{F}}$$

$$\leq C_{\text{Lip}} \left\| \widetilde{\boldsymbol{W}}_1 - \boldsymbol{W}_1 \right\|_{\text{F}} \sqrt{\kappa} \|\boldsymbol{X}\|_{\text{F}} \leq \tau \sqrt{\kappa} C_{\text{Lip}}.$$

For the $l$-th layer ($l \in \{2, 3, ..., L-1\}$), we have:

$$\left\| \widetilde{\boldsymbol{F}}_l - \boldsymbol{F}_l \right\|_{\text{F}}$$

$$= \left\| \frac{1}{L}\sigma_l(\widetilde{\boldsymbol{W}}_l\phi_l(\widetilde{\boldsymbol{F}}_{l-1})) + \alpha_{l-1}\widetilde{\boldsymbol{F}}_{l-1} - \frac{1}{L}\sigma_l(\boldsymbol{W}_l\phi_l(\boldsymbol{F}_{l-1})) - \alpha_{l-1}\boldsymbol{F}_{l-1} \right\|_{\text{F}}$$

$$\leq \left\| \frac{1}{L}\sigma_l(\widetilde{\boldsymbol{W}}_l\phi_l(\widetilde{\boldsymbol{F}}_{l-1})) - \frac{1}{L}\sigma_l(\boldsymbol{W}_l\phi_l(\boldsymbol{F}_{l-1})) \right\|_{\text{F}} + \alpha_{l-1}\left\| \widetilde{\boldsymbol{F}}_{l-1} - \boldsymbol{F}_{l-1} \right\|_{\text{F}} \quad \text{(By Triangle inequality)}$$

$$\leq \frac{C_{\text{Lip}}}{L}\left\| \widetilde{\boldsymbol{W}}_l\phi_l(\widetilde{\boldsymbol{F}}_{l-1}) - \boldsymbol{W}_l\phi_l(\boldsymbol{F}_{l-1}) \right\|_{\text{F}} + \left\| \widetilde{\boldsymbol{F}}_{l-1} - \boldsymbol{F}_{l-1} \right\|_{\text{F}} \quad \text{(By the Lipschitz continuity of } \sigma_l)$$

$$\leq \frac{C_{\text{Lip}}}{L}\left\{ \|\boldsymbol{W}_l\|_2 \left\| \phi_l(\widetilde{\boldsymbol{F}}_{l-1}) - \phi_l(\boldsymbol{F}_{l-1}) \right\|_{\text{F}} + \left\| \widetilde{\boldsymbol{W}}_l - \boldsymbol{W}_l \right\|_2 \left\| \phi_l(\widetilde{\boldsymbol{F}}_{l-1}) \right\|_{\text{F}} \right\} + \left\| \widetilde{\boldsymbol{F}}_{l-1} - \boldsymbol{F}_{l-1} \right\|_{\text{F}}$$

$$\leq \frac{C_{\text{Lip}}}{L}\sqrt{\kappa}\left\{ \|\boldsymbol{W}_l\|_2 \left\| \widetilde{\boldsymbol{F}}_{l-1} - \boldsymbol{F}_{l-1} \right\|_{\text{F}} + \left\| \widetilde{\boldsymbol{W}}_l - \boldsymbol{W}_l \right\|_2 \left\| \widetilde{\boldsymbol{F}}_{l-1} \right\|_{\text{F}} \right\} + \left\| \widetilde{\boldsymbol{F}}_{l-1} - \boldsymbol{F}_{l-1} \right\|_{\text{F}}$$

$$\leq \left( \frac{C_{\text{Lip}}}{L}\sqrt{\kappa}(\|\boldsymbol{W}_l\|_2 + \tau) + 1 \right)\left\| \widetilde{\boldsymbol{F}}_{l-1} - \boldsymbol{F}_{l-1} \right\|_{\text{F}} + \frac{C_{\text{Lip}}}{L}\sqrt{\kappa}\tau\|\boldsymbol{F}_{l-1}\|_{\text{F}}.$$

Therefore, by applying the inequality recursively and Lemmas 2 and 10, with probability at least $1-2(L-1)\exp(-m/2)-\delta$, we have:

$$\left\| \widetilde{\boldsymbol{F}}_{L-1} - \boldsymbol{F}_{L-1} \right\|_{\text{F}}$$

$$\leq [\frac{C_{\text{Lip}}}{L}\sqrt{\kappa}(3+\tau)+1]\left\| \widetilde{\boldsymbol{F}}_{L-2} - \boldsymbol{F}_{L-2} \right\|_{\text{F}} + \frac{C_{\text{Lip}}}{L}\sqrt{\kappa}\tau C_{\text{fmax}} \quad \text{(By Lemmas 2 and 10)}$$

$$\leq [\frac{C_{\text{Lip}}}{L}\sqrt{\kappa}(3+\tau)+1]^{L-2}\left\| \widetilde{\boldsymbol{F}}_1 - \boldsymbol{F}_1 \right\|_{\text{F}} + \sum_{i=0}^{L-3}[\frac{C_{\text{Lip}}}{L}\sqrt{\kappa}(3+\tau)+1]^i\frac{C_{\text{Lip}}}{L}\sqrt{\kappa}\tau C_{\text{fmax}} \quad \text{(By recursion)}$$

$$\leq (6\frac{C_{\text{Lip}}}{L}\sqrt{\kappa}+1)^{L-2}\tau\sqrt{\kappa}C_{\text{Lip}} + \sum_{i=0}^{L-3}(6\frac{C_{\text{Lip}}}{L}\sqrt{\kappa}+1)^i\frac{C_{\text{Lip}}}{L}\sqrt{\kappa}\tau C_{\text{fmax}}$$

$$\leq e^{6C_{\text{Lip}}\sqrt{\kappa}}(\sqrt{\kappa}C_{\text{Lip}} + C_{\text{fmax}})\tau.$$

$\square$

**Lemma 12.** *For $\boldsymbol{W},\widetilde{\boldsymbol{W}} \in \mathcal{B}(\boldsymbol{W}^{[1]},\tau)$ with $\tau \leq 3$, when the width satisfies $m = \Omega(p^2 N/\delta)$ and $\rho \leq 1$, with probability at least $1 - 2\exp(-m/2) - 2(L-2)\exp(-\kappa m/2) - 2p\exp(-m/2) - \delta$, we have:*

$$\left| f(\boldsymbol{X},\widetilde{\boldsymbol{W}}) - f(\boldsymbol{X},\boldsymbol{W}) - f(\boldsymbol{x},\boldsymbol{W}) - \left\langle (1-\beta)\nabla_{\boldsymbol{W}} f(\boldsymbol{x},\boldsymbol{W}) + \beta\nabla_{\boldsymbol{W}} f(\mathcal{A}_\rho(\boldsymbol{x},\boldsymbol{W}),\boldsymbol{W}),,\widetilde{\boldsymbol{W}} - \boldsymbol{W}\right\rangle \right|$$

$$\leq 18\sqrt{p}e^{6C_{\mathrm{Lip}}\sqrt{\kappa}}(1+\beta)(\sqrt{\kappa}C_{\mathrm{Lip}} + C_{\mathrm{fmax}})\tau,$$

$$\left| f(\mathcal{A}_\rho(\boldsymbol{x},\boldsymbol{W}),\widetilde{\boldsymbol{W}}) - f(\mathcal{A}_\rho(\boldsymbol{x},\boldsymbol{W}),\boldsymbol{W}) - \left\langle (1-\beta)\nabla_{\boldsymbol{W}} f(\boldsymbol{x},\boldsymbol{W}) + \beta\nabla_{\boldsymbol{W}} f(\mathcal{A}_\rho(\boldsymbol{x},\boldsymbol{W}),\boldsymbol{W}),\widetilde{\boldsymbol{W}} - \boldsymbol{W}\right\rangle \right|$$

$$\leq 18\sqrt{p}e^{6C_{\mathrm{Lip}}\sqrt{\kappa}}(2-\beta)(\sqrt{\kappa}C_{\mathrm{Lip}} + C_{\mathrm{fmax}})\tau.$$

*Proof.* To simplify the notation, in the following proof, the variable with $\tilde{\cdot}$ is related to $\widetilde{\boldsymbol{W}}$, and without $\tilde{\cdot}$ is related to $\boldsymbol{W}$. Then, let us prove the first inequality.

$$\left| \widetilde{f} - f - \left\langle (1-\beta)\nabla_{\boldsymbol{W}} f + \beta\nabla_{\boldsymbol{W}}\hat{f},\widetilde{\boldsymbol{W}} - \boldsymbol{W}\right\rangle \right|$$

$$= \left| \left\langle \widetilde{\boldsymbol{w}}_L,\widetilde{\boldsymbol{f}}_{L-1}\right\rangle - \left\langle \boldsymbol{w}_L,\boldsymbol{f}_{L-1}\right\rangle - \sum_{l=1}^{L}\left\langle (1-\beta)\nabla_{\boldsymbol{W}_l} f + \beta\nabla_{\boldsymbol{W}_l}\hat{f},\widetilde{\boldsymbol{W}}_l - \boldsymbol{W}_l\right\rangle \right|$$

$$= \left| \left\langle \widetilde{\boldsymbol{w}}_L,\widetilde{\boldsymbol{f}}_{L-1} - \boldsymbol{f}_{L-1}\right\rangle + \left\langle \boldsymbol{f}_{L-1},\widetilde{\boldsymbol{w}}_L - \boldsymbol{w}_L\right\rangle - \sum_{l=1}^{L-1}\left\langle (1-\beta)\nabla_{\boldsymbol{W}_l} f + \beta\nabla_{\boldsymbol{W}_l}\hat{f},\widetilde{\boldsymbol{W}}_l - \boldsymbol{W}_l\right\rangle \right. \tag{22}$$

$$\left. - \left\langle (1-\beta)\boldsymbol{f}_{L-1} + \beta\hat{\boldsymbol{f}}_{L-1},\widetilde{\boldsymbol{w}}_L - \boldsymbol{w}_L\right\rangle \right|$$

$$\leq \|\widetilde{\boldsymbol{w}}_L\|_2 \left\|\widetilde{\boldsymbol{f}}_{L-1} - \boldsymbol{f}_{L-1}\right\|_2 + \beta\left\|\boldsymbol{f}_{L-1} - \hat{\boldsymbol{f}}_{L-1}\right\|_2 \|\widetilde{\boldsymbol{w}}_L - \boldsymbol{w}_L\|_2$$

$$+ \left| \sum_{l=1}^{L-1}\left\langle (1-\beta)\nabla_{\boldsymbol{W}_l} f,\widetilde{\boldsymbol{W}}_l - \boldsymbol{W}_l\right\rangle \right| + \left| \sum_{l=1}^{L-1}\left\langle \beta\nabla_{\boldsymbol{W}_l}\hat{f},\widetilde{\boldsymbol{W}}_l - \boldsymbol{W}_l\right\rangle \right|.$$

The first term can be bounded by Lemmas 2 and 11 as follows:

$$\|\widetilde{\boldsymbol{w}}_L\|_2 \left\|\widetilde{\boldsymbol{f}}_{L-1} - \boldsymbol{f}_{L-1}\right\|_2$$

$$\leq \left(\left\|\boldsymbol{w}_L^{[1]}\right\|_2 + \left\|\widetilde{\boldsymbol{w}}_L - \boldsymbol{w}_L^{[1]}\right\|_2\right)\left(\left\|\widetilde{\boldsymbol{f}}_{L-1} - \boldsymbol{f}_{L-1}^{[1]}\right\|_2 + \left\|\boldsymbol{f}_{L-1} - \boldsymbol{f}_{L-1}^{[1]}\right\|_2\right)$$

$$\leq (3\sqrt{p} + \tau)2e^{6C_{\mathrm{Lip}}\sqrt{\kappa}}(\sqrt{\kappa}C_{\mathrm{Lip}} + C_{\mathrm{fmax}})\tau.$$

The second term can be bounded by Lemmas 3, 4 and 11 with $\rho \leq 1$ as follows:

$$\beta\left\|\hat{\boldsymbol{f}}_{L-1} - \boldsymbol{f}_{L-1}\right\|_2 \|\widetilde{\boldsymbol{w}}_L - \boldsymbol{w}_L\|_2$$

$$\leq \beta\left(\left\|\boldsymbol{f}_{L-1}^{[1]} - \boldsymbol{f}_{L-1}\right\|_2 + \left\|\hat{\boldsymbol{f}}_{L-1} - \hat{\boldsymbol{f}}_{L-1}^{[1]}\right\|_2 + \left\|\boldsymbol{f}_{L-1}^{[1]} - \hat{\boldsymbol{f}}_{L-1}^{[1]}\right\|_2\right)\left(\left\|\boldsymbol{w}_L - \boldsymbol{w}_L^{[1]}\right\|_2 + \left\|\widetilde{\boldsymbol{w}}_L - \boldsymbol{w}_L^{[1]}\right\|_2\right)$$

$$\leq \beta e^{6C_{\mathrm{Lip}}\sqrt{\kappa}}(\sqrt{\kappa}C_{\mathrm{Lip}} + C_{\mathrm{fmax}})(2\tau + 3\rho)2\tau. \tag{23}$$

The third term can be bounded by Lemmas 2 and 10 as follows:

$$(1-\beta)\left| \sum_{l=1}^{L-1}\left\langle \nabla_{\boldsymbol{W}_l} f,\widetilde{\boldsymbol{W}}_l - \boldsymbol{W}_l\right\rangle \right|$$

$$\leq (1-\beta)\sum_{l=1}^{L-1}C_{\mathrm{Lip}}\|\boldsymbol{W}_L\|_{\mathrm{F}}\|\phi_l(\boldsymbol{F}_{l-1})\|_{\mathrm{F}}\prod_{r=l+1}^{L-1}\left(\frac{C_{\mathrm{Lip}}}{L}\sqrt{\kappa}\|\boldsymbol{W}_r\|_2 + 1\right)\left\|\widetilde{\boldsymbol{W}}_l - \boldsymbol{W}_l\right\|_{\mathrm{F}} \tag{24}$$

$$\leq (1-\beta)\sum_{l=1}^{L-1}C_{\mathrm{Lip}}(3\sqrt{p} + \tau)\kappa C_{\mathrm{fmax}}\left(\frac{C_{\mathrm{Lip}}}{L}\sqrt{\kappa}(3+\tau) + 1\right)^{L-l-1}\tau$$

$$\leq (1-\beta)\sqrt{p}C_{\mathrm{fmax}}e^{6\sqrt{\kappa}C_{\mathrm{Lip}}}\tau.$$

Similarly, the fourth term can be upper bounded by $\beta\sqrt{p}C_{\text{fmax}}e^{6\sqrt{\kappa}C_{\text{Lip}}}\tau$. Plugging back Eq. (22) yields:

$$\left|f(\boldsymbol{x},\widetilde{\boldsymbol{W}})-f(\boldsymbol{x},\boldsymbol{W})-\left\langle(1-\beta)\nabla_{\boldsymbol{W}}f+\beta\nabla_{\boldsymbol{W}}\hat{f},\widetilde{\boldsymbol{W}}-\boldsymbol{W}\right\rangle\right|$$

$$\leq(3\sqrt{p}+\tau)2e^{6\sqrt{\kappa}C_{\text{Lip}}}(\sqrt{\kappa}C_{\text{Lip}}+C_{\text{fmax}})\tau+\beta e^{6\sqrt{\kappa}C_{\text{Lip}}}(\sqrt{\kappa}C_{\text{Lip}}+C_{\text{fmax}})(2\tau+3\rho)2\tau+\sqrt{p}C_{\text{fmax}}e^{6\sqrt{\kappa}C_{\text{Lip}}}\tau$$

$$\leq18\sqrt{p}e^{6\sqrt{\kappa}C_{\text{Lip}}}(1+\beta)(\sqrt{\kappa}C_{\text{Lip}}+C_{\text{fmax}})\tau,$$

which completes the proof of the first inequality in the lemma. The second inequality can be proved by the same method as in Lemma 6.

$\square$

**Lemma 13.** *There exists an absolute constant $C_1$ such that, for any $\epsilon>0$ and any $\boldsymbol{W},\widetilde{\boldsymbol{W}}\in\mathcal{B}(\boldsymbol{W}^{[1]},\tau)$ with $\tau\leq C_1\epsilon(\sqrt{\kappa}C_{\text{Lip}}+C_{\text{fmax}})^{-1}p^{-1/2}e^{-6C_{\text{Lip}}\sqrt{\kappa}}$, with probability at least $1-2\exp(-m/2)-2(L-2)\exp(-\kappa m/2)-2p\exp(-m/2)-\delta$, one has:*

$$\mathcal{L}(\boldsymbol{x},\widetilde{\boldsymbol{W}})\geq\mathcal{L}(\boldsymbol{x},\boldsymbol{W})+\left\langle(1-\beta)\nabla_{\boldsymbol{W}}\mathcal{L}(\boldsymbol{x},\boldsymbol{W})+\beta\nabla_{\boldsymbol{W}}\mathcal{L}(\mathcal{A}_\rho(\boldsymbol{x},\boldsymbol{W}),\boldsymbol{W}),\widetilde{\boldsymbol{W}}-\boldsymbol{W}\right\rangle-\epsilon,$$

$$\mathcal{L}(\mathcal{A}_\rho(\boldsymbol{x},\boldsymbol{W}),\widetilde{\boldsymbol{W}})\geq\mathcal{L}(\mathcal{A}_\rho(\boldsymbol{x},\boldsymbol{W}),\boldsymbol{W})+\left\langle(1-\beta)\nabla_{\boldsymbol{W}}\mathcal{L}(\boldsymbol{x},\boldsymbol{W})+\beta\nabla_{\boldsymbol{W}}\mathcal{L}(\mathcal{A}_\rho(\boldsymbol{x},\boldsymbol{W}),\boldsymbol{W}),\widetilde{\boldsymbol{W}}-\boldsymbol{W}\right\rangle-\epsilon.$$

*Proof.* Firstly, we will prove the first inequality. Recall that the cross-entropy loss is written as $\ell(z)=\log(1+\exp(-z))$ and we denote by $\mathcal{L}(\boldsymbol{x},\boldsymbol{W}):=\ell[y\cdot f(\boldsymbol{x},\boldsymbol{W})]$. Then one has:

$$\mathcal{L}(\boldsymbol{x},\widetilde{\boldsymbol{W}})-\mathcal{L}(\boldsymbol{x},\boldsymbol{W})$$
$$=\ell[y\tilde{f}]-\ell[yf]$$
$$\geq\ell'[yf]\cdot y\cdot(\tilde{f}-f)\quad\text{(By convexity of }\ell(z)\text{)}.$$
$$\geq\ell'[yf]\cdot y\cdot\left\langle(1-\beta)\nabla f,\widetilde{\boldsymbol{W}}-\boldsymbol{W}\right\rangle+\ell'[y\hat{f}]\cdot y\cdot\left\langle\beta\nabla\hat{f},\widetilde{\boldsymbol{W}}-\boldsymbol{W}\right\rangle-\kappa_1$$
$$=\left\langle(1-\beta)\nabla_{\boldsymbol{W}}\mathcal{L}(\boldsymbol{x},\boldsymbol{W})+\beta\nabla_{\boldsymbol{W}}\mathcal{L}(\mathcal{A}_\rho(\boldsymbol{x},\boldsymbol{W}),\boldsymbol{W}),\widetilde{\boldsymbol{W}}-\boldsymbol{W}\right\rangle-\kappa_1\quad\text{(By chain rule)},$$

where we define:

$$\kappa_1:=\left|\ell'[yf]\cdot y\cdot(\tilde{f}-f)-\ell'[yf]\cdot y\cdot\left\langle(1-\beta)\nabla f,\widetilde{\boldsymbol{W}}-\boldsymbol{W}\right\rangle-\ell'[y\hat{f}]\cdot y\cdot\left\langle\beta\nabla\hat{f},\widetilde{\boldsymbol{W}}-\boldsymbol{W}\right\rangle\right|.$$

Thus, it suffices to show that $\kappa_1$ can be upper bounded by $\epsilon$:

$$\kappa_1\leq\left|\ell'[yf]\cdot y\left\{\tilde{f}-f-\left\langle(1-\beta)\nabla_{\boldsymbol{W}}f+\beta\nabla_{\boldsymbol{W}}\hat{f},\boldsymbol{W}),\widetilde{\boldsymbol{W}}-\boldsymbol{W}\right\rangle\right\}\right|+\left|\left\{\ell'[yf]\cdot y-\ell'[y\hat{f}]\cdot y\right\}\beta\left\langle\nabla_{\boldsymbol{W}}\hat{f},\widetilde{\boldsymbol{W}}-\boldsymbol{W}\right\rangle\right|$$

$$\leq18\sqrt{p}e^{6\sqrt{\kappa}C_{\text{Lip}}}(1+\beta)(\sqrt{\kappa}C_{\text{Lip}}+C_{\text{fmax}})\tau+2\beta\left|\left\langle\nabla_{\boldsymbol{W}}\hat{f},\widetilde{\boldsymbol{W}}-\boldsymbol{W}\right\rangle\right|$$

$$\leq18\sqrt{p}e^{6\sqrt{\kappa}C_{\text{Lip}}}(1+\beta)(\sqrt{\kappa}C_{\text{Lip}}+C_{\text{fmax}})\tau+2\beta\left|\sum_{l=1}^{L-1}\left\langle\nabla_{\boldsymbol{W}_l}f(\boldsymbol{x},\boldsymbol{W}),\widetilde{\boldsymbol{W}}_l-\boldsymbol{W}_l\right\rangle\right|+2\beta|\langle\boldsymbol{f}_{L-1},\widetilde{\boldsymbol{w}_L}-\boldsymbol{w}_L\rangle|$$

$$\leq18\sqrt{p}e^{6\sqrt{\kappa}C_{\text{Lip}}}(1+\beta)(\sqrt{\kappa}C_{\text{Lip}}+C_{\text{fmax}})\tau+2\beta\sqrt{p}C_{\text{fmax}}e^{6\sqrt{\kappa}C_{\text{Lip}}}\tau+4\beta C_{\text{fmax}}\tau$$

$$\leq24\sqrt{p}(1+\beta)e^{6\kappa C_{\text{Lip}}}(\kappa C_{\text{Lip}}+C_{\text{fmax}})\tau$$

$$\leq\epsilon,$$

$$(25)$$

where the first and the third inequality is by triangle inequality, the second inequality is by Lemma 12 and the fact that $|\ell'[yf(\boldsymbol{x},\boldsymbol{W})]\cdot y|\leq1$, , the fourth inequality follows the same proof as in Eqs. (23) and (24), and the last inequality is by the condition that if $\tau\leq C_2\epsilon(1+\beta)^{-1}p^{-1/2}(\sqrt{\kappa}C_{\text{Lip}}+C_{\text{fmax}})^{-1}e^{-6C_{\text{Lip}}\sqrt{\kappa}}$ for some absolute constant $C_2$. Following the same method in Lemma 7, we can prove the second inequality if $\tau\leq C_3\epsilon(3-\beta)^{-1}p^{-1/2}(\sqrt{\kappa}C_{\text{Lip}}+C_{\text{fmax}})^{-1}e^{-6\sqrt{\kappa}C_{\text{Lip}}}$ for some absolute constant $C_3$. Lastly, setting $C_1=\max\{C_2,C_3\}$ and noting that $(3-\beta)^{-1}<(1+\beta)^{-1}$ completes the proof. $\square$

**Lemma 14.** *For any $\epsilon,\delta,R>0$, there exists: $m^{\star}=\mathcal{O}(\mathrm{poly}(R,L,k,C_{\mathrm{Lip}},p,\beta))\cdot\epsilon^{-2}e^{12C_{\mathrm{Lip}}\sqrt{\kappa}}\log(1/\delta)$ such that if $m\geq m^{\star}$, then for any $\boldsymbol{W}^{*}\in\mathcal{B}(\boldsymbol{W}^{[1]},Rm^{-1/2})$, under the following choice of step-size $\gamma=\nu\epsilon/[C_{\mathrm{Lip}}pkC_{\mathrm{fmax}}^{2}mLe^{12C_{\mathrm{Lip}}\sqrt{\kappa}}]$ and iterations $N=L^{2}R^{2}C_{\mathrm{Lip}}pkC_{\mathrm{fmax}}^{2}e^{12C_{\mathrm{Lip}}\sqrt{\kappa}}/(2\varepsilon^{2}\nu)$ for some small enough absolute constant $\nu$, the cumulative loss can be upper bounded with probability at least $1-\delta$ by:*

$$\frac{1}{N}\sum_{i=1}^{N}\mathcal{L}(\boldsymbol{x}_{i},\boldsymbol{W}^{[i]})\leq\frac{1}{N}\sum_{i=1}^{N}\mathcal{L}(\boldsymbol{x}_{i},\boldsymbol{W}^{\star})+3\epsilon,$$

$$\frac{1}{N}\sum_{i=1}^{N}\mathcal{L}(\mathcal{A}_{\rho}(\boldsymbol{x}_{i},\boldsymbol{W}^{[i]}),\boldsymbol{W}^{[i]})\leq\frac{1}{N}\sum_{i=1}^{N}\mathcal{L}(\mathcal{A}_{\rho}(\boldsymbol{x}_{i},\boldsymbol{W}^{[i]}),\boldsymbol{W}^{\star})+3\epsilon.$$

*Proof.* We set $\tau=C_{1}(1+\beta)^{-1}\epsilon(\sqrt{\kappa}C_{\mathrm{Lip}}+C_{\mathrm{fmax}})^{-1}p^{-1/2}e^{-6C_{\mathrm{Lip}}\sqrt{\kappa}}$ where $C_{1}$ is a small enough absolute constant so that the requirements on $\tau$ in Lemmas 12 and 13 can be satisfied.

We set $\tau=C_{1}\epsilon(1+\beta)^{-1}(C_{\mathrm{Lip}}+C_{\mathrm{fmax}})^{-1}e^{-6C_{\mathrm{Lip}}}$ where $C_{1}$ is a small enough absolute constant so that the requirements on $\tau$ in Lemmas 6 and 7 can be satisfied. Let $Rm^{-1/2}\leq\tau$, then we obtain the condition for $\boldsymbol{W}^{\star}\in\mathcal{B}(\boldsymbol{W}^{[1]},\tau)$, i.e., $m\geq R^{2}C_{1}^{-2}\epsilon^{-2}(1+\beta)^{2}p(\sqrt{\kappa}C_{\mathrm{Lip}}+C_{\mathrm{fmax}})^{2}e^{12C_{\mathrm{Lip}}\sqrt{\kappa}}$. We now show that $\boldsymbol{W}^{[1]},...,\boldsymbol{W}^{[N]}$ are inside $\mathcal{B}(\boldsymbol{W}^{[1]},\tau)$ as well. The proof follows by induction. Clearly, we have $\boldsymbol{W}^{[1]}\in\mathcal{B}(\boldsymbol{W}^{[1]},\tau)$. Suppose that $\boldsymbol{W}^{[1]},...,\boldsymbol{W}^{[i]}\in\mathcal{B}(\boldsymbol{W}^{[1]},\tau)$, then with probability at least $1-\delta'$, we have:

$$\left\|\boldsymbol{W}_{l}^{[i+1]}-\boldsymbol{W}_{l}^{[1]}\right\|_{\mathrm{F}}\leq\sum_{j=1}^{i}\left\|\boldsymbol{W}_{l}^{[j+1]}-\boldsymbol{W}_{l}^{[j]}\right\|_{\mathrm{F}}$$

$$=\sum_{j=1}^{i}\gamma\left\|(1-\beta)\nabla_{\boldsymbol{W}_{l}}\mathcal{L}(\boldsymbol{x}_{j},\boldsymbol{W}^{[j]})+\beta\nabla_{\boldsymbol{W}_{l}}\mathcal{L}(\mathcal{A}_{\rho}(\boldsymbol{x}_{j},\boldsymbol{W}^{[j]}),\boldsymbol{W}^{[j]})\right\|_{\mathrm{F}}$$

$$\leq\gamma(1-\beta)N\left\|\nabla_{\boldsymbol{W}_{l}}\mathcal{L}(\boldsymbol{x}_{j},\boldsymbol{W}^{[j]})\right\|_{\mathrm{F}}+\gamma\beta N\left\|\nabla_{\boldsymbol{W}_{l}}\mathcal{L}(\mathcal{A}_{\rho}(\boldsymbol{x}_{j},\boldsymbol{W}^{[j]}),\boldsymbol{W}^{[j]})\right\|_{\mathrm{F}}$$

$$\leq\gamma NC_{\mathrm{Lip}}\|\boldsymbol{W}_{L}\|_{\mathrm{F}}\|\phi_{l}(\boldsymbol{F}_{l-1})\|_{F}\prod_{r=l+1}^{L-1}(\frac{C_{\mathrm{Lip}}}{L}\sqrt{k}\|\boldsymbol{W}_{r}\|_{2}+1)\left\|\widetilde{\boldsymbol{W}}_{l}-\boldsymbol{W}_{l}\right\|_{\mathrm{F}}$$

$$\leq\gamma NC_{\mathrm{Lip}}(3\sqrt{p}+\tau)\sqrt{k}C_{\mathrm{fmax}}\left(\frac{C_{\mathrm{Lip}}}{L}\sqrt{k}(3+\tau)+1\right)^{L-l-1}$$

$$\leq 6\gamma NC_{\mathrm{Lip}}\sqrt{p\kappa}C_{\mathrm{fmax}}e^{6\sqrt{\kappa}C_{\mathrm{Lip}}}.$$

Plugging in our parameter choice for $\gamma$ and $N$ leads to:

$$\left\|\boldsymbol{W}_{l}^{[i+1]}-\boldsymbol{W}_{l}^{[1]}\right\|_{\mathrm{F}}\leq 3C_{\mathrm{Lip}}\sqrt{p\kappa}C_{\mathrm{fmax}}e^{6\sqrt{\kappa}C_{\mathrm{Lip}}}LR^{2}\epsilon^{-1}m^{-1}\leq\tau,$$

where the last inequality holds as long as $m\geq 3C_{\mathrm{Lip}}\sqrt{p\kappa}C_{\mathrm{fmax}}e^{12\sqrt{\kappa}C_{\mathrm{Lip}}}LR^{2}C_{1}^{-1}\epsilon^{-2}$. Therefore by induction we see that $\boldsymbol{W}^{[1]},...,\boldsymbol{W}^{[N]}\in\mathcal{B}(\boldsymbol{W}^{[1]},\tau)$. Now, we are ready to prove the first inequality in the lemma. We provide an upper bound for the cumulative loss as follows:

$$\mathcal{L}(\boldsymbol{x}_{i},\boldsymbol{W}^{[i]})-\mathcal{L}(\boldsymbol{x}_{i},\boldsymbol{W}^{\star})$$

$$\leq\left\langle(1-\beta)\nabla_{\boldsymbol{W}}\mathcal{L}(\boldsymbol{x}_{i},\boldsymbol{W}^{[i]})+\beta\nabla_{\boldsymbol{W}}\mathcal{L}(\mathcal{A}_{\rho}(\boldsymbol{x}_{i},\boldsymbol{W}^{[i]}),\boldsymbol{W}^{[i]}),\boldsymbol{W}^{[i]}-\boldsymbol{W}^{\star}\right\rangle+\epsilon\quad\text{(By Lemma 7)}$$

$$=\frac{\left\langle\boldsymbol{W}^{[i]}-\boldsymbol{W}^{[i+1]},\boldsymbol{W}^{[i]}-\boldsymbol{W}^{\star}\right\rangle}{\gamma}+\epsilon$$

$$=\frac{\|\boldsymbol{W}^{[i]}-\boldsymbol{W}^{[i+1]}\|_{\mathrm{F}}^{2}+\|\boldsymbol{W}^{[i]}-\boldsymbol{W}_{l}^{\star}\|_{\mathrm{F}}^{2}-\|\boldsymbol{W}^{[i+1]}-\boldsymbol{W}^{\star}\|_{\mathrm{F}}^{2}}{2\gamma}+\epsilon$$

$$=\frac{\|\boldsymbol{W}^{[i]}-\boldsymbol{W}^{\star}\|_{\mathrm{F}}^{2}-\|\boldsymbol{W}^{[i+1]}-\boldsymbol{W}^{\star}\|_{\mathrm{F}}^{2}+\gamma^{2}\left\|(1-\beta)\nabla_{\boldsymbol{W}}\mathcal{L}(\boldsymbol{x}_{i},\boldsymbol{W}^{[i]})+\beta\nabla_{\boldsymbol{W}}\mathcal{L}(\mathcal{A}_{\rho}(\boldsymbol{x}_{i},\boldsymbol{W}^{[i]}),\boldsymbol{W}^{[i]})\right\|_{\mathrm{F}}^{2}}{2\gamma}+\epsilon$$

$$\leq\frac{\|\boldsymbol{W}^{[i]}-\boldsymbol{W}^{\star}\|_{\mathrm{F}}^{2}-\|\boldsymbol{W}^{[i+1]}-\boldsymbol{W}^{\star}\|_{\mathrm{F}}^{2}}{2\gamma}+\frac{6^{2}\gamma^{2}C_{\mathrm{Lip}}^{2}p\kappa C_{\mathrm{fmax}}^{2}e^{12\sqrt{\kappa}C_{\mathrm{Lip}}}}{2\gamma}+\epsilon.$$

Telescoping over $i = 1,...,N$, we obtain:

$$\frac{1}{N}\sum_{i=1}^{N}\mathcal{L}(\boldsymbol{x}_i,\boldsymbol{W}^{[i]})$$

$$\leq \frac{1}{N}\sum_{i=1}^{N}\mathcal{L}(\boldsymbol{x}_i,\boldsymbol{W}^{\star}) + \frac{\|\boldsymbol{W}^{(1)}-\boldsymbol{W}^{\star}\|_{\mathrm{F}}^2}{2N\gamma} + 18\gamma C_{\mathrm{Lip}}^2 p\kappa C_{\mathrm{fmax}}^2 e^{12\sqrt{\kappa}C_{\mathrm{Lip}}} + \epsilon$$

$$\leq \frac{1}{N}\sum_{i=1}^{N}\mathcal{L}(\boldsymbol{x}_i,\boldsymbol{W}^{\star}) + \frac{LR^2}{2\gamma m N} + 18\gamma C_{\mathrm{Lip}}^2 p\kappa C_{\mathrm{fmax}}^2 e^{12\sqrt{\kappa}C_{\mathrm{Lip}}} + \epsilon$$

$$\leq \frac{1}{N}\sum_{i=1}^{N}\mathcal{L}(\boldsymbol{x}_i,\boldsymbol{W}^{\star}) + 3\epsilon,$$

where in the first inequality we simply remove the term $-\|\boldsymbol{W}^{[N+1]}-\boldsymbol{W}^{\star}\|_{\mathrm{F}}^2/(2\gamma)$ to obtain an upper bound, the second inequality follows by the assumption that $\boldsymbol{W}^{\star} \in \mathcal{B}(\boldsymbol{W}^{[1]}, Rm^{-1/2})$, the third inequality is by the parameter choice of $\gamma$ and $N$. One can follow the same procedure to prove the second inequality in the lemma that is based on Lemma 13. $\square$

## F    ADDITIONAL EXPERIMENTS

### F.1    CORRELATION BETWEEN FGSM-NTK AND PGD-NTK SCORES AND ACCURACY

We present a comprehensive analysis of the correlation between NTK scores and accuracy in Figure 6. The following 10 NTKs are selected: clean NTK, robust NTK with $\rho \in \{3/255, 8/255\}$ subjected to FGSM/PGD attack, the corresponding robust twice NTK subjected to FGSM/PGD attack. The result demonstrates that robust NTK has a higher correlation compared with standard NTK under adversarial training. Moreover, our analysis indicates a general trend wherein NTKs with FGSM attacks display a higher correlation with accuracy than those subjected to PGD attacks. These results suggest that employing NTKs with FGSM attacks is preferable for guiding robust architecture searches.

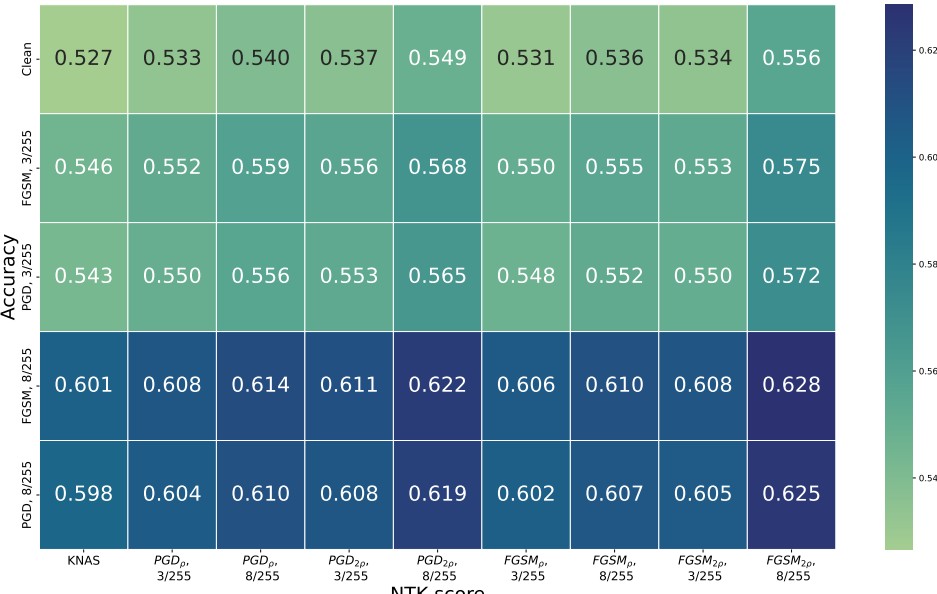

Figure 6: Spearman coefficient between NTK-score, and various metrics. The label with subscript $\rho$ in the x-axis indicates the score w.r.t the robust NTK while the one with subscript $\rho$ indicates the score w.r.t the twice robust NTK.

## F.2 EVALUATION RESULTS ON NAS APPROACHES

In Table 7, we provide additional evaluation results on NAS approaches, e.g., AdvRush (Mok et al., 2021), KNAS (Xu et al., 2021), EigenNAS (Zhu et al., 2022a), NASI (Shu et al., 2022), NASWOT (Mellor et al., 2021). For KNAS, we use the data with PGD attack to construct the kernel. We follow the same set-up in Xu et al. (2021) to efficiently compute the NTK. Specifically, we randomly select 50 samples and estimate the minimum eigenvalue of NTK by its Frobenius norm. Similarly, in EigenNAS, we choose the trace of the NTK as proposed in Zhu et al. (2022a). Regarding NASI, NASWOT, and AdvRush, we use the official open-source implementation to produce the result. For DARTS, we use the first-order implementation, i.e., DARTS-V1, from the library of Dong & Yang (2020).

## F.3 GRADIENT OBFUSCATION IN SPARSE AND DENSE ARCHITECTURES

Originally pointed out by Athalye et al. (2018), gradient obfuscation is a cause for the overestimated robustness of several adversarial defense mechanisms, e.g., parametric noise injection defense (He et al., 2019) and ensemble defenses (Gao et al., 2022). Such gradient obfuscation might exist in the defense of several dense or sparse architectures (Kundu et al., 2021). In this work, we employ vanilla adversarial training as a defense approach, which does not suffer from gradient obfuscation, as originally demonstrated in (Athalye et al., 2018). Given the huge considered search space with 6466 architectures, we investigate whether the sparsity of the architecture design might have an impact on gradient obfuscation. Specifically, based on the search space in Figure 1, we group all architectures in terms of the number of operators "Zeroize", which can represent the sparsity of the network. Next, we select the one with the highest robust accuracy in each group and check whether this optimal architecture satisfies the characteristics of non-gradient obfuscation in Table 4.

Table 4: Characteristics of non-gradient obfuscation Athalye et al. (2018).

| |
| --- |
| a) One-step attack performs worse than iterative attacks. |
| b) Black-box attacks perform worse than white-box attacks. |
| c) Unbounded attacks fail to obtain $100\%$ attack success rate. |
| d) Increasing the perturbation radius $\rho$ does not increase the attack success rate. |
| e) Adversarial examples can not be found by random sampling if gradient-based attacks do not. |

Table 5 presents the evaluation result of these architectures under FGSM, Square Attack, and PGD attack with varying radius. We can see that all of these architectures clearly satisfy a), c), d). Regarding e), all of the networks still satisfy it because gradient-based attacks can find adversarial examples. For b), only the sparse network with 4 "Zeroize" does not satisfy. Therefore, we can see that these architectures can be considered without suffering gradient obfuscation, which is consistent with the original conclusion for adversarial training in (Athalye et al., 2018).

Table 5: Testing of gradient obfuscation for architectures with different sparsity.

| Number of "Zeroize" | Clean | FGSM (3/255) | Square (3/255) | PGD (3/255) | PGD (8/255) | PGD (16/255) | PGD (32/255) | PGD (64/255) | PGD (128/255) | PGD (255/255) |
| --- | --- | --- | --- | --- | --- | --- | --- | --- | --- | --- |
| 0 | 0.7946 | 0.6975 | 0.6929 | 0.6924 | 0.4826 | 0.1774 | 0.0083 | 0.0000 | 0.0000 | 0.0000 |
| 1 | 0.7928 | 0.6933 | 0.6884 | 0.6874 | 0.4798 | 0.1744 | 0.0077 | 0.0000 | 0.0000 | 0.0000 |
| 2 | 0.7842 | 0.6832 | 0.6786 | 0.6779 | 0.4711 | 0.1739 | 0.0080 | 0.0000 | 0.0000 | 0.0000 |
| 3 | 0.7456 | 0.6515 | 0.6482 | 0.6472 | 0.4442 | 0.1599 | 0.0079 | 0.0000 | 0.0000 | 0.0000 |
| 4 | 0.5320 | 0.4560 | 0.4427 | 0.4528 | 0.3191 | 0.1393 | 0.0193 | 0.0010 | 0.0000 | 0.0000 |

## F.4 ROBUSTNESS TOWARDS DISTRIBUTION SHIFT

To make the benchmark more comprehensive in a realistic setting, we evaluate each architecture in the search space under distribution shift. We choose CIFAR-10-C (Hendrycks & Dietterich, 2019),

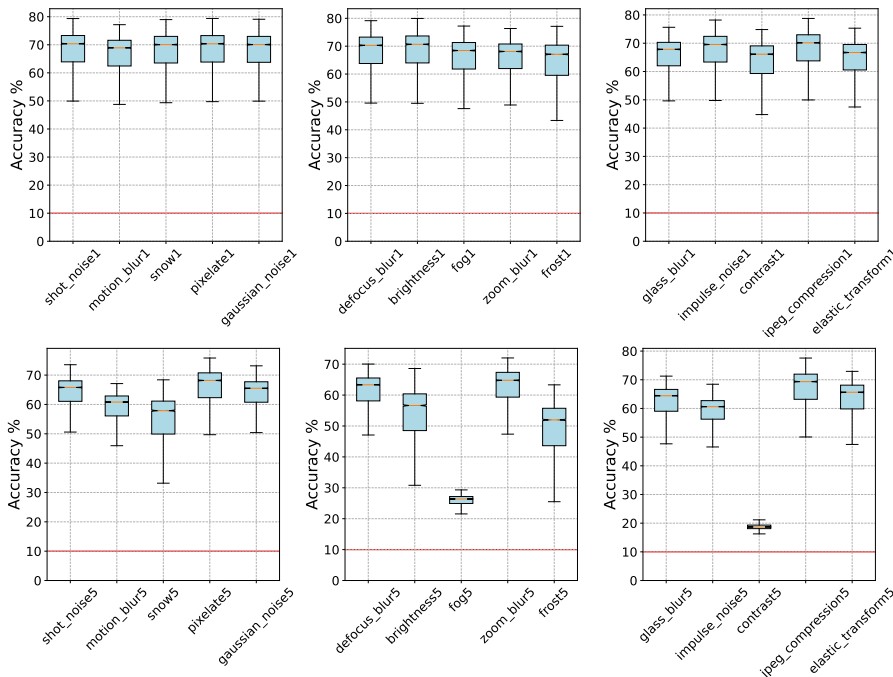

Figure 7: Boxplots for accuracy under various corruptions and severity levels (1 and 5) of all 6466 architectures in the considered search space. Red line indicates the accuracy of a random guess.

Table 6: Spearman coefficient between various accuracies on CIFAR-10-C and adversarial robust accuracy on CIFAR-10. We can see that the adversarial robust accuracy has a relatively high correlation with all corruptions except for contrast corruptions.

|  | PGD 3/255 | FGSM 3/255 | PGD 8/255 | FGSM 8/255 |
|---|---|---|---|---|
| Shot noise | 0.984 | 0.984 | 0.965 | 0.968 |
| Motion blur | 0.990 | 0.990 | 0.980 | 0.982 |
| Snow | 0.995 | 0.995 | 0.989 | 0.990 |
| Pixelate | 0.996 | 0.996 | 0.984 | 0.988 |
| Gaussian noise | 0.981 | 0.981 | 0.961 | 0.965 |
| Defocus blur | 0.993 | 0.993 | 0.983 | 0.986 |
| Brightness | 0.993 | 0.993 | 0.984 | 0.986 |
| Fog | 0.916 | 0.916 | 0.918 | 0.918 |
| Zoom blur | 0.992 | 0.992 | 0.978 | 0.982 |
| Frost | 0.991 | 0.991 | 0.985 | 0.987 |
| Glass blur | 0.993 | 0.993 | 0.981 | 0.984 |
| Impulse noise | 0.947 | 0.947 | 0.936 | 0.938 |
| Contrast | -0.555 | -0.555 | -0.574 | -0.571 |
| Jpeg compression | 0.996 | 0.997 | 0.984 | 0.987 |
| Elastic transform | 0.995 | 0.995 | 0.984 | 0.987 |

which includes 15 visual corruptions with 5 different severity levels, resulting in 75 new metrics. In Figure 7, we show the boxplots of the accuracy. All architectures are robust towards corruption with lower severity levels. When increasing the severity levels to five, the models are no longer robust to fog and contrast architectures. Similar to robust accuracy under FGSM and PGD attacks, we can see a non-negligible gap between the performance of different architectures, which motivates robust architecture design. Moreover, in Table 6, we plot the Spearman coefficient between various accuracies under corruptions with severity level 5 on CIFAR-10-C and adversarial robust accuracy on CIFAR-10. Interestingly, we can see that the adversarial robust accuracy has a relatively high correlation with all corruptions except for contrast corruptions. As a result, performing NAS on the adversarially trained architectures in the benchmark can obtain a robust guarantee of distribution shift.

Table 7: Result of different NAS algorithms on our NAS-RobBench-201.

| | Method | Clean | FGSM (3/255) | PGD (3/255) | FGSM (8/255) | PGD (8/255) |
|---|---|---|---|---|---|---|
| | Optimal | 0.794 | 0.698 | 0.692 | 0.537 | 0.482 |
| Clean | Regularized Evolution | 0.791 | 0.693 | 0.688 | 0.530 | 0.476 |
| | Random Search | 0.779 | 0.682 | 0.676 | 0.520 | 0.470 |
| | Local Search | 0.796 | 0.697 | 0.692 | 0.533 | 0.478 |
| FGSM (8/255) | Regularized Evolution | 0.790 | 0.693 | 0.688 | 0.532 | 0.478 |
| | Random Search | 0.774 | 0.679 | 0.674 | 0.521 | 0.471 |
| | Local Search | 0.794 | 0.695 | 0.689 | 0.535 | 0.481 |
| PGD (8/255) | Regularized Evolution | 0.789 | 0.692 | 0.686 | 0.531 | 0.478 |
| | Random Search | 0.771 | 0.676 | 0.671 | 0.520 | 0.471 |
| | Local Search | 0.794 | 0.695 | 0.689 | 0.535 | 0.481 |
| Train-free | KNAS | 0.767 | 0.675 | 0.67 | 0.521 | 0.472 |
| | EigenNAS | 0.766 | 0.674 | 0.668 | 0.52 | 0.471 |
| | NASI | 0.666 | 0.571 | 0.567 | 0.410 | 0.379 |
| | NASWOT | 0.766 | 0.674 | 0.668 | 0.52 | 0.471 |
| Differentiable search | AdvRush | 0.587 | 0.492 | 0.489 | 0.352 | 0.330 |
| | DARTS | 0.332 | 0.286 | 0.285 | 0.215 | 0.213 |

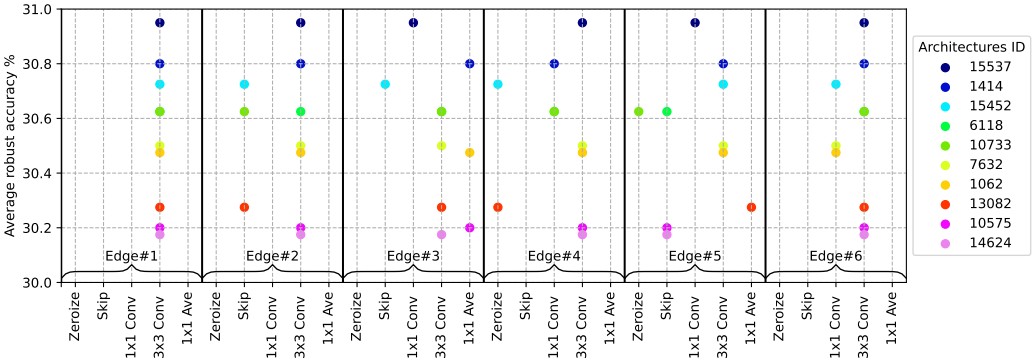

Figure 8: The operators of each edge in the top-10 (average robust accuracy of PGD-3/255, PGD-8/255 FGSM-3/255 FGSM-8/255) architecture in Jung et al. (2023). We can see a notable difference from the one present in Figure 4 in terms of architectures id and selected nodes. Additionally, the best architecture in NAS-RobBench-201 has a higher accuracy ($\approx 60\%$) than that of the best architecture ($\approx 30\%$) in Jung et al. (2023).

## G  LIMITATIONS

Our benchmark and theoretical results have a primary limitation, as they currently pertain exclusively to Fully Connected Neural Networks (FCNN) and Convolutional Neural Networks (CNN), with no exploration of their applicability to state-of-the-art vision Transformers. Furthermore, our study is constrained by a search space encompassing only 6466 architectures, leaving room for future research to consider a more expansive design space.

Our analysis framework could be adapted for a vision Transformer but the analysis for Transformer is still difficult because of different training procedures and more complicated architectures. The optimization guarantees of Transformer under a realistic setting (e.g., scaling, architecture framework) in theory is still an open question. Secondly, similar to the NTK-based analysis in the community, we study the neural network in the linear regime, which can not fully explain the success of practical neural networks (Allen-Zhu et al., 2019; Yang & Hu, 2021). However, NAS can still benefit from the NTK result. For example, the empirical NTK-based NAS algorithm allows for feature learning by taking extra metrics (Mok et al., 2022). Additionally, though our theoretical result uses the NTK tool under

the lazy training regime (Chizat et al., 2019), NAS can still benefit from the NTK result. For example, the empirical NTK-based NAS algorithm allows for feature learning by taking extra metrics, see Mok et al. (2022) for details.

## H  SOCIETAL IMPACT

Firstly, this work releases a NAS benchmark that includes the evaluation result of $6466$ architectures under adversarial training, which facilitates researchers to efficiently and fairly evaluate various NAS algorithms. Secondly, this work provides the generalization guarantee for searching robust architecture for the NAS community, which paves the way for practitioners to develop train-free NAS algorithms. Hence, we do not expect any negative societal bias from this perspective. However, we encourage the community to explore further the general societal bias from machine learning models into real-world applications.

