# OpenReview forum: "Robust NAS under adversarial training: benchmark, theory, and beyond"
_ICLR.cc/2024/Conference — ICLR 2024 poster_

### Official Review · Reviewer_AZRS · 2023-10-22

**Soundness:** 3 good
**Presentation:** 3 good
**Contribution:** 3 good
**Rating:** 6
**Confidence:** 3

**Summary:**

For fast and standardized evaluation of Neural Architecture Search algorithms, it is important to develop various benchmarks that contain a large set of architectures and their quality measured with diverse performance metrics. In this vein, the paper builds and releases a new NAS benchmark that considers not only the standard accuracy of evaluated architectures but also their robustness against adversarial attacks. The authors extend the existing NAS-Bench-201 benchmark by adversarially training the NAS-Bench-201 architectures and evaluating them on clean and perturbed data. The proposed robust NAS benchmark is named NAS-RobBench-201. Additionally, the authors theoretically characterize the robust generalization performance of architectures using the Neural Tangent Kernel (NTK) framework.

**Strengths:**

- As the authors note, adversarially training takes considerably more time than standard training with the cross-entropy loss. Given the enormous computational cost of adversarial training and evaluating a large number of architectures, I believe the authors really have done an impressive set of experiments.
- I generally agree with the necessity to create a NAS benchmark that targets robustness and generalization beyond standard test data.
- I do not feel confident enough to review and comment on the more theoretical parts of the paper. The proofs were not reviewed thoroughly, and thus I cannot vouch for its correctness. I am, however, familiar with the NTK theory, and I find the idea of extending the NTK-based analysis to explain adversarial robustness interesting.

**Weaknesses:**

- Currently, the robust accuracy is measured only under FGSM and PGD attacks. I think the performance under stronger attack methods (e.g., AutoAttack) is necessary to make the benchmark more reliable.
- The authors observe that the standard and robust accuracies exhibit a meaningful level of correlation. If they are, do we really need a separate robustness benchmark that includes robust accuracies? Wouldn’t searching for the optimal architecture in a conventional sense work as a good proxy and naturally result in a more robust architecture?
- I believe that NAS-Bench-201 is too constrained of a search space to accurately reflect the vastness of the potential architecture pool. For instance, [1] shows that many NTK-based methods fail outside the boundary of NAS-Bench-201. While this is a good first step towards building a more comprehensive robustness benchmark, it would be great if the authors could show that the experimental and theoretical analyses hold outside the tested search space.
- It would be nice to include the performance of trained architectures on out-of-distribution data and/or datasets with distribution shift in the benchmark. Incorporating these other types of robustness would make the benchmark more comprehensive, especially considering that in real deployment scenarios, the architectures will encounter generic OOD data more often than adversarially-perturbed data.
- In section 4.4, the authors study the correlation between the minimum eigenvalue of the NTK matrix and robust accuracy. I understand that this NTK-score is derived from theoretical resultss, but have the authors studied the relationship between other NTK-based scores (e.g., condition number [2]) and robust accuracy?

[1] Mok, Jisoo, et al. "Demystifying the neural tangent kernel from a practical perspective: Can it be trusted for neural architecture search without training?." Proceedings of the IEEE/CVF Conference on Computer Vision and Pattern Recognition. 2022.
[2] Chen, Wuyang, Xinyu Gong, and Zhangyang Wang. "Neural Architecture Search on ImageNet in Four GPU Hours: A Theoretically Inspired Perspective." International Conference on Learning Representations. 2020.

**Questions:**

Please refer to the weaknesses section.

---

> ### Author Response · Authors · 2023-11-19
> **Response (1 out of 2) to reviewer AZRS**
>
> We thank the reviewer AZRS for the insightful feedback. We address the concerns below and add necessary revisions highlighted in red color in the updated version.
>
> ---
>
> > **Q1:** [The performance under stronger attack methods (e.g., AutoAttack), out-of-distribution data and/or datasets with distribution shift.]
>
> **A1:** Thanks for the suggestion, the discussion on the AutoAttack can be found in [general response](https://openreview.net/forum?id=cdUpf6t6LZ&noteId=FRw76YCx2K) for details.
>
> Furthermore, to make the benchmark more comprehensive in a realistic setting, we evaluate 6466 architectures in the search space under distribution shift. We choose CIFAR-10-C [3], which includes 15 visual corruptions with 5 different severity levels, resulting in 75 new metrics. In the supplementary material, we have provided the result in a JSON file.
>
> We add a new section Appendix F.4 to describe the evaluation and discuss the result. In Fig.7 of the revised version (or the [link](https://imgur.com/a/iVolRzr)), we show the boxplots of the accuracy. We can find that all architectures are robust towards corruption with lower severity levels. When increasing the severity levels to five, the models are no longer robust to fog and contrast architectures. Moreover, similar to robust accuracy under FGSM and PGD attacks, we can see a non-negligible gap between the performance of different architectures, which motivates robust architecture design. In Table 6 of the revised version (or the [link](https://imgur.com/a/iVolRzr)),  we show the Spearman coefficient between various accuracies on CIFAR-10-C and adversarial robust accuracy on CIFAR-10. Interestingly, we can see that the adversarial robust accuracy has a relatively high correlation with all corruptions except for contrast corruptions. As a result, performing NAS on the adversarially trained architectures in the benchmark can achieve a robust guarantee of distribution shift.
>
> ---
>
> > **Q2:** [The authors observe that the standard and robust accuracies exhibit a meaningful level of correlation. If they are, do we really need a separate robustness benchmark that includes robust accuracies? Wouldn’t searching for the optimal architecture in a conventional sense work as a good proxy and naturally result in a more robust architecture?]
>
> **A2:**  Let us explain why we do need a separate benchmark. We agree with the reviewer that there exists a partial correlation between clean accuracy and robust accuracy, as suggested by our benchmark. This is one of the findings from our benchmark: demonstrating that the correlation is not extremely low under adversarial training, compared to the benchmark under standard training [6]. We have included the discussion in Appendix B.2 of the paper.
>
> Notice that before building the benchmark, we did not know the relationship between clean accuracy and robust accuracy under adversarial training. To be specific, in the existing benchmark [6] that was constructed in the same space but under standard training, the correlation was not high. This correlation was enhanced only when adversarial training was involved. This is one finding from our benchmark. Moreover, we can see a notable difference between these two benchmarks in terms of top architecture id and selected nodes, see Fig. 4 and Fig. 8. This suggests that employing the benchmark under standard training to identify a resilient architecture may not guarantee its robustness under adversarial training.
>
> ---
>
> > **Q3:** [NAS-Bench-201 is too constrained of a search space to accurately reflect the vastness of the potential architecture pool. For instance, [1] shows that many NTK-based methods fail outside the boundary of NAS-Bench-201. While this is a good first step towards building a more comprehensive robustness benchmark, it would be great if the authors could show that the experimental and theoretical analyses hold outside the tested search space.]
>
> **A3:** We thank the reviewer for pointing out this interesting question. When checking previous literature on NTK train-free algorithms, we find that NTK-based methods perform well on different search spaces beyond NAS-Bench-201, e.g., TE-NAS [2] and  EigenNAS [5] on DARTS search spaces. Nevertheless, the applicability of such results under adversarial training remains uncertain on other search spaces, as it requires expensive construction of the benchmark under adversarial training.

---

> > ### Author Response · Authors · 2023-11-19
> > **Response (2 out of 2) to reviewer AZRS**
> >
> > > **Q4:** [Evaluation on other NTK-based scores (e.g., condition number [2]).]
> >
> > **A4:**  According to the reviewer's suggestion, we show the correlation between accuracy and several NTK scores based on their condition number, which is calculated by using all of the network parameters following [2]. Interestingly, using the condition number of robust NTK does not improve the correlation compared to using the condition number of clean NTK, but using twice-robust NTK is still better than robust NTK. Given that the condition number can describe the curvature of NTK, and can reveal the convergence speed of the neural network [8],  it would be interesting to theoretically analyze the effect of the condition number on the robust generalization bound in the future.
> >
> > ||Clean | FGSM, 3/255 | PGD, 3/255|  APGD, 3/255 | FGSM, 8/255| PGD, 8/255| APGD, 8/255|
> > |--------|--------|-------|-------|-------|-------|-------|-------|
> > | Cond($K$) | 0.739 | 0.742 | 0.739 | 0.738 | 0.765 | 0.755 | 0.748 |
> > |Cond($\hat{K}_{\rho}$), 3/255  | 0.505 | 0.512 | 0.509 | 0.508 | 0.546 | 0.536 | 0.531 |
> > |Cond($\hat{K}_{\rho}$), 8/255 | 0.721 | 0.729 | 0.725 | 0.724 | 0.762 | 0.753 | 0.747 |
> > | Cond($\hat{K}_{2\rho}$), 3/255  |  0.558 | 0.566 | 0.562 | 0.561 | 0.600 | 0.589 | 0.584 |
> > | Cond($\hat{K}_{2\rho}$), 8/255  | 0.724 | 0.730 | 0.727 | 0.726 | 0.762 | 0.752 | 0.746 |
> >
> > ---
> >
> > If the reviewer AZRS has any remaining concerns, we are happy to clarify further.
> >
> > ---
> >
> >
> > ### Refs
> >
> > [1] "Demystifying the neural tangent kernel from a practical perspective: Can it be trusted for neural architecture search without training?." CVPR, 2022.
> >
> > [2] "Neural architecture search on imagenet in four gpu hours: A theoretically inspired perspective." ICLR, 2021.
> >
> > [3] "Benchmarking Neural Network Robustness to Common Corruptions and Perturbations." ICLR, 2019.
> >
> > [4] "On feature learning in neural networks with global convergence guarantees." ICLR, 2022.
> >
> > [5] "Generalization properties of NAS under activation and skip connection search." NeurIPS, 2022.
> >
> > [6] "Neural Architecture Design and Robustness: A Dataset." ICLR, 2023.
> >
> > [7] "KNAS: green neural architecture search." ICML, 2021.
> >
> > [8] "Toward a theory of optimization for over-parameterized systems of non-linear equations: the lessons of deep learning." Arxiv, 2020.

---

### Official Review · Reviewer_dbFp · 2023-11-01

**Soundness:** 3 good
**Presentation:** 3 good
**Contribution:** 2 fair
**Rating:** 6
**Confidence:** 4

**Summary:**

The paper first presents an adversarially trained benchmark based on NASBench 201, and present NASRobBench 201.

**Strengths:**

There is not much literature that presents a unified robust NAS benchmark, in that sense the work targets a unique space of research that needs exploration.

The NTK score vs accuracy correlation sounds interesting.

**Weaknesses:**

1. It has been well known that adversarial robustness often may occur due to gradient obfuscation and is applicable for different sparse and dense model architectures [1,2]. Thus, a discussion on that would be necessary. In specific, is there a way to benchmark based on a subnet's susceptibility to be more prone towards obfuscation?

2. Di you use ImageNet or ImageNet-16-120? As the Fig. 2 and the contribution section has mention of each.

3. Please demonstrate the efficacy of the NASRobBench on Autoattack.

4. The related work of Adversarial example generation is not up to date, the author should discuss more about the auto attacks and other contemporary attacks.

[1] Obfuscated gradients give a false sense of security: Circumventing defenses to adversarial examples, ICML 2018.

[2] DNR: A Tunable Robust Pruning Framework Through Dynamic Network Rewiring of DNNs, ASP-DAC 2021.

**Questions:**

1. Apart from the doubts in weakness, I have the following question:

a. How you add noise "twice"?

b. Please extend the NTK score vs accuracy correlation with more recent and stronger attack scenarios.


### Post rebuttal:
-------------------------

Thanks authors for rebuttal.

Despite the limited novelty of the work, which the work did not commit anyway, I believe the work is good enough with enough empirical evaluations. Thus, I increase the score to 6! Though a 7 might have been the right score for this. I refrained from giving it a 8, just due to limited novelty scope, however, I think this work could be a useful add to the adversarial network benchmarking community.

---

> ### Author Response · Authors · 2023-11-17
> **Response (1 out of 2) to reviewer dbFp**
>
> We thank the reviewer dbFp for the constructive feedback. We address the concerns below and incorporate necessary revisions highlighted in red color in the updated version.
>
> ---
>
> > **Q1:** [It has been well known that adversarial robustness often may occur due to gradient obfuscation and is applicable for different sparse and dense model architectures [1,2]. Thus, a discussion on that would be necessary. In specific, is there a way to benchmark based on a subnet's susceptibility to be more prone towards obfuscation?]
>
> **A1:** We are thankful to the reviewer for the suggestion of this ablation experiment. Upon the suggestion of the reviewer, we add a new section in Appendix F.3 to discuss this.
>
> We agree with the reviewer that gradient obfuscation might be a cause for the overestimated robustness of several adversarial defense mechanisms, e.g., parametric noise injection defense [3] and ensemble defenses [6], and can exist in the defense of several architectures [2]. However, in this work, we employ vanilla adversarial training as a defense approach, which does not suffer from gradient obfuscation, as originally demonstrated in [1] and mentioned in later literature [3].
>
>
> We investigate whether the sparsity of the architecture design might have an impact on gradient obfuscation. Specifically, based on the search space in Figure 1 of the paper, we group all architectures in terms of the number of operators “Zeroize’’, which can represent the sparsity of the network (the more  “Zeroize” operations, the more sparse the resulting network). Next, we select the network with the highest robust accuracy in each group and check whether this optimal architecture satisfies the following characteristics of non-gradient obfuscation [1]:
> -  a) One-step attack performs worse than iterative attacks.
> -  b) Black-box attacks perform worse than white-box attacks.
> -  c) Unbounded attacks fail to obtain $100\%$ attack success rate.
> - d) Increasing the perturbation radius $\rho$ does not increase attack success rate.
> - e) Adversarial examples can not be found by random sampling if gradient-based attacks do not.
>
> The result below is the evaluation of these architectures under FGSM, Square attack, and PGD attack with varying perturbation radius. We can see that all of these architectures clearly satisfy a), c), and d). Regarding e), all of the networks still satisfy it because gradient-based attacks can find adversarial examples. For b), only the sparse network with 4 ``Zeroize" does not satisfy the condition. Therefore, we can see that these architectures can not be considered as suffering gradient obfuscation, which is consistent with the original conclusion for adversarial training in [1].
>
> |Number of “Zeroize"|Clean | FGSM, 3/255 | Square, 3/255 |PGD, 3/255|  PGD, 8/255 | PGD, 16/255| PGD, 32/255| PGD, 64/255|PGD, 128/255|PGD, 255/255|
> |--------|-------|-------|-------|-------|-------|-------|-------|-------|-------|-------|
> | 0  |0.7946|0.6975|0.6929|0.6924|0.4826|0.1774|0.0083|0.0000|0.0000|0.0000|
> | 1  |0.7928|0.6933|0.6884|0.6874|0.4798|0.1744|0.0077|0.0000|0.0000|0.0000|
> | 2 | 0.7842|0.6832|0.6786|0.6779|0.4711|0.1739|0.0080|0.0000|0.0000|0.0000|
> | 3 | 0.7456|0.6515|0.6482|0.6472|0.4442|0.1599|0.0079|0.0000|0.0000|0.0000|
> | 4 |0.5320|0.4560|0.4427|0.4528|0.3191|0.1393|0.0193|0.0010|0.0000|0.0000|
>
> Lastly, under the aforementioned defense mechanisms e.g., noise injection defense and ensemble defenses, some of these architectures in the search space might be more prone to gradient obfuscation. Thus, we believe it is possible to further develop robust NAS benchmarks under other defense mechanisms, but this is out of the scope of the current work.
>
>
> ---
>
>
> > **Q2:** [Did you use ImageNet or ImageNet-16-120? As the Fig. 2 and the contribution section has mention of each.]
>
> A2: Following NAS-Bench-201 [4], we conduct an evaluation on ImageNet-16-120, which is a well-established benchmark. In the updated version, we revised the contribution on page 2 as follows:
>
> "107k GPU hours are required to build the benchmark on three datasets (CIFAR-10/100, ImageNet-16-120) under adversarial training."
>
>
> ---
>
> > **Q3:** [Please demonstrate the efficacy of the NASRobBench on Autoattack.]
>
> **A3:**  Thanks for the suggestion, we have provided more evaluation of the benchmark, see [general response](https://openreview.net/forum?id=cdUpf6t6LZ&noteId=FRw76YCx2K) for details.

---

> ### Author Response · Authors · 2023-11-17
> **Response (2 out of 2) to reviewer dbFp**
>
> ---
>
> > **Q4:** [The related work of Adversarial example generation is not up to date, the author should discuss more about the auto attacks and other contemporary attacks.]
>
> **A4:** In the revised version, we update the related work from the classical adversarial attack to up-to-date ways as follows:
>
> To mitigate the effect of hyper-parameters in PGD and the overestimation of robustness [1], [7]  propose two variants of parameter-free PGD attack, namely, APGD_CE and APGD_DLR, where CE stands for cross-entropy loss and DLR indicates difference of logits ratio (DLR) loss. Both APGD_CE and APGD_DLR attacks dynamically adapt the step-size of PGD based on the loss at each step. Furthermore, to enhance the diversity of robust evaluation, [7] introduce Auto-attack, which is the integration of APGD_CE, APGD_DLR, Adaptive Boundary Attack (FAB) [4], and black-box Square Attack [5].
>
> ---
>
> > **Q5:** [How you add noise "twice"?]
>
> **A5:** We perform one PGD/FGSM attack on the raw image data to generate the adversarial data, and then we perform the same attack on this adversarial data again. We have added such clarification in page 9 of the revised version.
>
> ---
>
> > **Q6:** [Please extend the NTK score vs accuracy correlation with more recent and stronger attack scenarios.]
>
> **A6:**  We test the correlation between the robust accuracy under APGD_CE and the various NTK metrics in CIFAR-10. On one hand, we can still see that the robust NTK enables the increasing correlation under adversarial training, which is consistent with our observation in the original version for FGSM and PGD attacks. On the other hand, we can see the correlation decrease only ~0.5% for APGD accuracy compared to PGD accuracy. We have updated the result on the APGD attack In the revised version, see Fig.5 in page 9 or [the anonymous link](https://imgur.com/a/WcnEzOR).
>
> |Clean | FGSM, 3/255 | PGD, 3/255|  APGD, 3/255 | FGSM, 8/255| PGD, 8/255| APGD, 8/255|
> |--------|-------|-------|-------|-------|-------|-------|
> | KNAS, clean | 0.527 | 0.546 | 0.543 | 0.541 | 0.601 | 0.598 | 0.593 |
> | KNAS, PGD, 3/255 | 0.533 | 0.552 | 0.550 | 0.548 | 0.608 | 0.604 | 0.599 |
> | KNAS, PGD, 8/255 | 0.540 | 0.559 | 0.556 | 0.554 | 0.614 | 0.610 | 0.605 |
> | KNAS, twice-PGD, 3/255| 0.537 | 0.556 | 0.553 | 0.551 | 0.611 | 0.608 | 0.602 |
> | KNAS, twice-PGD,  8/255 | 0.549 | 0.568 | 0.565 | 0.564 | 0.622 | 0.619 | 0.615 |
>
>
> ---
>
> If the reviewer dbFp has any remaining concerns, we are happy to clarify further.
>
> ---
>
> ### Refs
>
> [1] Athalye, et al. "Obfuscated gradients give a false sense of security: Circumventing defenses to adversarial examples." ICLM, 2018.
>
> [2] Kundu, et al. "Dnr: A tunable robust pruning framework through dynamic network rewiring of dnns." ASP-DAC, 2021.
>
> [3] He, et al. "Parametric noise injection: Trainable randomness to improve deep neural network robustness against adversarial attack." CVPR, 2019.
>
> [4] Dong, et al. "NAS-Bench-201: Extending the Scope of Reproducible Neural Architecture Search." ICLR, 2020.
>
> [5] Andriushchenko, et al. "Square attack: a query-efficient black-box adversarial attack via random search." ECCV, 2020.
>
> [6] Gao ,et al. "MORA: Improving Ensemble Robustness Evaluation with Model Reweighing Attack." NeurIPS, 2022.
>
> [7] Croce, et al. "Reliable evaluation of adversarial robustness with an ensemble of diverse parameter-free attacks." ICML, 2020.
>
> [8] Croce, et al. "Minimally distorted adversarial examples with a fast adaptive boundary attack." ICML, 2020.

---

> ### Author Response · Authors · 2023-11-20
> **Response to reviewer dbFp**
>
> We appreciate the constructive feedback from the reviewer dbFp. During the rebuttal, we have already:
> - evaluated all 6466 architectures under the APGD_CE attack on CIFAR-10 and plotted their correlation with the NTK score. Additionally, we evaluated 2000 architectures using the complete AutoAttack. Based on the suggestion from reviewer AZRS, we also assessed 6466 architectures using 75 additional corruption metrics on the CIFAR-10-C dataset and analyzed the results.
>
>
> - added a new section to analyze gradient obfuscation in the proposed benchmark.
>
>
> - incorporated more attacks in the related work, clarified the usage of ImageNet-16-120, and explained the implementation of adding noise twice.
>
>
> We hope that our additional experiments and clarifications address the reviewer’s concerns. We sincerely appreciate it if the reviewer could reconsider the evaluation.
>
>
> Best regards,
>
>
> Authors

---

### Official Review · Reviewer_QErk · 2023-11-07

**Soundness:** 2 fair
**Presentation:** 3 good
**Contribution:** 2 fair
**Rating:** 6
**Confidence:** 4

**Summary:**

This paper presents two primary contributions. Firstly, the authors have developed a search space encompassing over 6,000 adversarially trained architectures, addressing the limitation of prior studies that lacked such a comprehensive search space. Secondly, the authors demonstrate that robust architectures can indeed be identified within this search space by employing robust NTK.

**Strengths:**

1. The paper's principal strength lies in its creation of an adversarially trained search space, which required 107k GPU hours to construct. This significant advancement is poised to benefit researchers focusing on robust architecture search in the future immensely.
2. The analyses conducted within this adversarial search space are considered to be novel. It is particularly noteworthy to observe cross-sectional evidence in the NAS search space, confirming the expectation that a 3x3 kernel size CNN layer contributes to robustness. Additionally, the findings concerning the correlation between clean accuracy and robustness in the face of adversarial attacks are intriguing.
3. The paper is well-written, offering clarity and ease of comprehension.

**Weaknesses:**

1. The paper lacks a detailed explanation and justification for the necessity of employing twice perturbation, as mentioned at the end of page 7. It remains unclear why robust NTK requires a double perturbation when conventional adversarial training typically examines generalization from a single perturbation.
2. While it is posited that robust accuracy is influenced by the adversarial term, the theoretical analysis provided appears to be a reiteration of what was presented by Zhu et al. and Cao et al. in the context of clean NTK. Consequently, the contribution in this area seems minimal. Notably, the robust term A*(x,W), which is contingent on W and x, was not accounted for in the theoretical framework and was instead treated as an independent input variable. This oversight could impact the applicability of the theoretical analysis.
3. The paper employs an evaluation metric that is not standard, raising questions about its normalization. The last part of page 4, in the dataset paragraph, should clarify whether the metric is normalized before the attack and then inputted into the model. Additionally, it would be beneficial if the authors specified whether the same perturbation radius (32x32) used for ImageNet was applied during training.

* All of the previous concerns have been resolved through the rebuttal. Thanks for the clear and explicit response to my initial concerns and questions. Therefore, I have revised my score to 6. Regarding normalization in evaluation, I would like the author to double-check whether normalization is also applied inside the attack functions, specifically after adding the perturbation and before forwarding it to the model.

**Questions:**

1. Could you please clarify what 'optimal' refers to in Table 2?
2. In the first column of Table 2, what is the definition of 'criterion' used for?
3. It would be interesting to learn about the correlation results when measured with the existing clean NTK, such as in previous works by Zhu et al. and Cao et al. (which could correspond to beta being 0).
4. Referring to Figure 3, it appears that there is a high correlation between clean and robust accuracy. However, I am curious about the implications of using just clean NTK values for adversarial robust search and how that might affect the interpretation of the results.


Minor Comments
1. In Table 2, the entries in the first column are not center-aligned. This misalignment could be corrected for improved readability and a more professional presentation of the data.
2. The interpretations of Figure 3, specifically parts (b) and (c), pose some difficulties. Enhanced specificity in the explanations accompanying these figures would greatly facilitate understanding.

---

> ### Author Response · Authors · 2023-11-15
> **Response (1 out of 3) to reviewer QErk**
>
> We thank the reviewer QErk for the insightful feedback. Below, we address the concerns pointed out by the reviewer QErk and revise the text in red in the updated version.
>
> ---
> > **Q1:** [Explanation for employing twice perturbation. Why robust NTK requires a double perturbation when conventional adversarial training typically examines generalization from a single perturbation.]
>
> **A1:** We explain the usage of twice-robust NTK both theoretically and empirically.
>
> - **Theoretically**, In the neural tangent kernel theory, we need to define a linearized neural network around the initialization of its weight. Specifically, in equation 15, we consider the linearized network
> $F_{\bf W^{[1]},\bf W^\star}(\bf x) :=
> f(\bf x,\bf W^{[1]})  + \langle
> (1-\beta)\nabla f_{\bf W}(\bf x,\bf W^{[1]}) +
> \beta \nabla f_{\bf W}(\mathcal{A}^*_\rho(\bf x,\bf W^{[1]}),\bf W^{[1]})  , \bf W^\star - \bf W^{[1]} \rangle$.
> Then at the end of the proof of Theorem 1, we have
> $F_{\bf W^{[1]},\bf W^\star}(\mathcal{A}^*_\rho(\bf x_i,\bf W^{[1]}))=    f(\mathcal{A}^*_\rho(\bf x_i,\bf W^{[1]}),\bf W^{[1]})  + \langle
>    (1-\beta)\nabla f_{\bf W}(\mathcal{A}^*_\rho(\bf x_i,\bf W^{[1]}),\bf W^{[1]}) +  \beta \nabla f_{\bf W}(\mathcal{A}^*_\rho(\mathcal{A}^*_\rho(\bf x_i,\bf W^{[1]}),\bf W^{[1]}),\bf W^{[1]})  , \bf W^\star - \bf W^{[1]} \rangle. $
> As a result, the Jacobian is given by
> $\widetilde{\bf J}\_\mathrm{all}  := (1-\beta)\hat{\bf J}\_\rho+\beta \hat{\bf J}\_{2\rho}$, where the twice-NTK appears.
>
>
> -  **Empirically**, we have demonstrated the **superiority of twice-robust NTK when compared to the (single perturbation) robust NTK in Figure 5**, which shows that the correlation between the accuracy ranking and twice-robust NTK is higher than the robust NTK.
>
> We agree with the reviewer that the intrinsic motivation of twice perturbation is not as straightforward as classical adversarial training works in the single perturbation style. However, this twice perturbation approach shares a similar style with adding noise before the adversarial attack [5]. More precisely, [5] demonstrate that augmenting clean data with noise before executing adversarial attack has the benefit of avoiding catastrophic overfitting, compared to a classical adversarial attack on the clean data. The proposed Noise-FGSM achieves promising performance across several datasets (e..g, CIFAR-10, and SVHN)  and mitigates robust overfitting. This demonstrates that introducing a stronger attack is beneficial to obtain better performance in practice. We believe that the development of algorithms based on the twice perturbation style will be a viable and beneficial avenue.
>
>
> ---
>
> > **Q2:** [Not standard evaluation raises questions about its normalization. Whether the metric is normalized before the attack and then inputted into the model. Whether the same perturbation radius (32x32) used for ImageNet was applied during training.]
>
> **A2:** In this work, the data is normalized with zero-mean and unit-variance before the attack. We have added such clarification in the dataset paragraph in the revised version. We emphasize that **we are using standard tools and evaluation metrics**, as used also in several ML papers [4,11] and open-source robustness library [3] developed by MadryLab. Regarding the implementation detail, we employ the Torchattacks library [7]. Lastly, we will release the pre-trained weight of 6444* 3* 3 architectures to encourage the practitioners to evaluate with more metrics.
>
> We emphasize the same perturbation radius $\rho = 8/255$ is used for ImageNet in the training procedure of the revised version.
>
> ---
>
> > **Q3:** [What 'optimal' refers to in Table 2.]
>
> **A3:** Optimal refers to the architecture with the highest average robust accuracy among the benchmarks. We have updated the caption of Table 2 in the revised version.
>
> ---
>
> > **Q4:** [The definition of 'criterion' in the first column of Table 2.]
>
> **A4:**  The first column of Table 2 is referred to as the "attack scheme" and we updated it in the revised version to avoid misunderstanding. The "criterion" used in our previous version indicates how the accuracy is measured during the searching phase of these baseline methods. We have already updated it in the caption of Table 2 for a better understanding in our new version. We are thankful to the reviewer for the attentive study of our work.

---

> ### Author Response · Authors · 2023-11-15
> **Response (2 out of 3) to reviewer QErk**
>
> > **Q5:** [The theoretical analysis provided appears to be a reiteration of what was presented by [1,2] in the context of clean NTK. Consequently, the contribution in this area seems minimal. The robust term was not accounted for in the theoretical framework. ]
>
> **A5:**  Let us explain why the previous work cannot be directly applied to our setting. How to handle the **multi-objective** trade-off between clean/robust accuracy under the more general robust architecture search framework is still unclear in theory. Here, we clarify the technical difficulties behind this.
>
> Previous work [2] builds the generalization bound of fully-connected neural networks via NTK and [1] extends this result under the activation function search and skip connection search. Both of them focus on the standard training. Instead, our work studies the robust generalization bound under *multi-objective adversarial training* of the searched architecture. Here we list the following reasons why previous work cannot be employed to our setting under multi-objective training (including adversarial training):
> -   **Trade-off and analysis of multi-objective training** are unclear: We focus on a multi-objective training scheme by introducing a regularization parameter to balance the standard training and adversarial training. In this case, the coupling relationship during training as well as the clean/robust accuracy trade-off makes the analysis difficult. For example, how to tackle the robust term $A^\star(x,W)$ for analysis is questionable from previous work [1,2].
> -   **NTK formulation under multiple-objective training** is unclear: To our knowledge, only [8] consider an NTK-based analysis in adversarial training. However, [8] only involves the optimization convergence under adversarial training without generalization analysis. Previous works [1,2] are based on the sole NTK for generalization guarantees. Therefore, we need a new way to build different NTKs to connect clean/robust accuracy.
>
> Our analysis addressed the following technical difficulties: 1) how to build the proof framework under multi-objective training by the well-designed joint of NTKs; 2) how to tackle the coupling relationship among several NTKs and derive the lower bound of the minimum eigenvalue of NTKs. 3) how to properly tackle the robust term $A^\star(x,W)$  between the input and weight perturbation; (See Lemma 4 for detail).
> Accordingly, our result demonstrates that, under adversarial training, the generalization performance (clean accuracy and robust accuracy) is affected by different NTKs. Concretely, the clean accuracy is determined by one clean NTK and robust NTK; while robust accuracy is determined by robust NTK and its “twice” perturbation version. Our results demonstrate the effect of different search schemes, perturbation radius, and the balance parameter, which doesn’t exist in previous literature.
>
> ---
>
> > **Q6:** [It would be interesting to learn about the correlation results when measured with the existing clean NTK, such as in previous works by [1,2] (which could correspond to beta being 0).]
>
> **A6:** We are thankful to the reviewer for the suggestion. In the first column of Figure 5, we have provided the result for KNAS [10], which relies on the minimum eigenvalue of the clean NTK, and we can see the correlation with respect to clean NTK is lower than robust NTK.  [3, Zhu et al] propose EigenNAS, which also uses the clean NTK but with different estimations for the minimum eigenvalue of NTK. Below we provide the result for EigenNAS.  Following the original version of the paper, we plot the Spearman correlation between different NTK scores and various accuracy metrics. Specifically, we use adversarial data with PGD attacks to construct the robust NTK. We can see that robust NTK still has a higher correlation than clean NTK.
>
> | |Clean | FGSM, 3/255 | PGD, 3/255| FGSM, 8/255| PGD, 8/255|
> |--------|-------|-------|--------|------|--------|
> | EigenNAS, clean| 0.527|  0.546|  0.543|  0.601|  0.598|
> | EigenNAS, PGD, 3/255|0.534|  0.553|  0.550|  0.608|  0.605 |
> | EigenNAS, PGD, 8/255| 0.540|  0.559|  0.556|  0.614|  0.610|
>
>
> ---
>
> > **Q7:** [There is a high correlation between clean and robust accuracy. The implications of using just clean NTK values for adversarial robust search and how that might affect the interpretation of the results.]
>
> **A7:** We agree with the reviewer that there exists a partial correlation between clean accuracy and robust accuracy. However, we can still see from the results of Figure 5 (or the response for Q6) that the correlation between clean NTK and robust accuracy is lower than that of the robust NTK. This implies that using clean NTK for robust NAS is worse than using robust NTK under adversarial training.
>
> ---
>
> > **Q8:** [In Table 2, the entries in the first column are not center-aligned.]
>
> **A8:**  We are thankful to the reviewer for the attentive reading. We fixed the vertical alignment issue in the revised version.

---

> ### Author Response · Authors · 2023-11-15
> **Response (3 out of 3) to reviewer QErk**
>
> > **Q9:** [The interpretations of Figure 3, specifically parts (b) and (c), pose some difficulties. Enhanced specificity in the explanations accompanying these figures would greatly facilitate understanding.]
>
> **A9:** We have updated the caption in Figure 3 (b) and (c) as follows: (b) Overall, the architecture ranking sorted by robust metric and clean metric correlate well for lower ranking (when the value of the x-axis is larger, i.e., ranking of those bad architectures) but there still exists a difference for higher ranking. This motivates the NAS for robust architecture in terms of robust accuracy instead of clean accuracy. (c): The result reveals a high correlation across various datasets, thereby inspiring the exploration of searching on a smaller dataset to find a robust architecture for larger datasets.
>
> ---
>
> If the reviewer QErk has any remaining concerns, we are happy to clarify further.
>
> ---
>
> ### Refs
>
> [1] "Generalization properties of NAS under activation and skip connection search." NeurIPS, 2022.
>
> [2] "Generalization bounds of stochastic gradient descent for wide and deep neural networks." NeurIPS, 2019.
>
> [3] https://github.com/MadryLab/robustness.
>
> [4] "Overfitting in adversarially robust deep learning." ICML, 2020.
>
> [5] "Make some noise: Reliable and efficient single-step adversarial training." NeurIPS, 2022.
>
> [6] "Exploring the loss landscape in neural architecture search." Uncertainty in Artificial Intelligence, 2021.
>
> [7] "Torchattacks: A pytorch repository for adversarial attacks." ArXiv, 2020.
>
> [8] "Convergence of adversarial training in overparametrized neural networks." NeurIPS, 2019.
>
> [9] "Explaining and harnessing adversarial examples." ICLR, 2015.
>
> [10] "KNAS: green neural architecture search." ICML, 2021.
>
> [11] "Theoretically principled trade-off between robustness and accuracy." ICML, 2019.

---

> ### Author Response · Authors · 2023-11-20
> **Response to reviewer QErk**
>
> We are grateful for the valuable feedback from the reviewer QErk. During the rebuttal, we have:
>
> - Explained the twice robust NTK theoretically and empirically.
>
> - Clarified the theoretical contribution of our work and its contrast against the prior work.
>
> - Reported the correlation results when measured with clean NTK.
>
> - Polished the writing of Table 2, clarified the evaluation metrics, and explained the implications of using clean NTK values for robust NAS.
>
> We would appreciate it if the reviewer could provide feedback on our previous responses.
>
> Best regards,
>
> Authors

---

### Author Response · Authors · 2023-11-17
**General response to feedback on AutoAttack**

Dear reviewers,

We appreciate your insightful comments. During the rebuttal period, we are launching the evaluation on AutoAttack as reviewers dbFp and AZRS suggested. Since the evaluation is time-consuming (e.g., **10-20 minutes for one architecture** in the benchmark in a single RTX 4090), currently we have released the following results:

- all 6466 architectures under the APGD_CE attack (*one typical type of AutoAttack*) on CIFAR-10, see the json file in the supplementary material. Besides, we exhibit its correlation with the NTK score, see Fig. 5 of page 9 or [the anonymous link](https://imgur.com/a/WcnEzOR).
- 3379 architectures evaluated on the complete AutoAttack, see the json file in the supplementary material.

We will continue running these experiments under AutoAttack and appreciate the reviewers’ comments on this, which facilitates the development of a robust NAS algorithm in the community.

We promise to release all  pre-trained weight of 6466 * 3 * 3 architectures to encourage future practitioners to use our pre-trained weight for further evaluation under more attacks and further experiments. In the supplementary material, we provide some examples of the pre-trained weight.

Best regards,

Authors

---

### Author Response · Authors · 2023-11-22
**Appreciate the constructive reviews; Make updates during the rebuttal; Eager for feedback**

Dear reviewers,

We are truly thankful for all your insightful reviews. Your comments have inspired us to make a number of updates to the paper, which we believe will improve over the original version. In detail, we have made the following effort:
-  **800 additional GPU hours** are invested to make the benchmark more comprehensive. See the result in the JSON file of the supplementary material.

-  **Distribution shift assessment**. We evaluate 6466 architectures in the search space under distribution shift with 75 new metrics. In the supplementary material, we add a new section Appendix F.4 to discuss our additional ablation experiments.

-   **AutoAttack assessment**. We evaluate all 6466 architectures under the APGD_CE attack (one typical type of AutoAttack) on CIFAR-10. Besides, we exhibit its correlation with the NTK score, see Fig. 5 on page 9 of the paper. We evaluate 3379 architectures on the complete AutoAttack. The remaining experiment is still running and will be included in the final version of the paper.

- **Clarification on the theoretical contribution** of our work and its contrast against the prior work.

-  **Additional exploration of the correlation between NTK and accuracy**, e.g.,  the correlation results when measured with EigenNAS.

-  **Gradient obfuscation analysis**. A new section in Appendix F.3 has been added to scrutinize gradient obfuscation within our proposed benchmark.

Lastly, **given the rebuttal deadline is within the next few hours**, we would greatly appreciate it if the reviewers could give any feedback based on our response.

Best regards,

Authors

---

### Meta-Review · Area_Chair_oDAU · 2023-12-11

**Metareview:**

In this paper, the authors first propose a robust NAS benchmark that consists of both clean and robust accuracies for a large number of adversarially trained architectures from the NAS-Bench-201 search space. Then, they further proposes a neural tangent kernel (NTK)-based approach to search for the architectures with both high clean and robust accuracies, with theoretically-grounded scoring measure. The experimental results show the effectiveness of the method on the proposed benchmark, with some findings on the architectural components that help with robustness and the correlations between the clean and the robust accuracy.

All reviewers agreed that building a benchmark focused on the robustness of the architecture is an important contribution and found the proposed NTK-based NAS approach as compelling. However, initially some of the reviewers leaned toward rejection due to some of the unresolved issues regarding the novelty of the analysis over the existing analysis on the NTK-based NAS methods that only focus on the clean accuracy, lack of experimental validation on other types of attacks (e.g. AutoAttack), lack of ablation studies, and some potential for gradient obfuscation. However, these concerns were addressed away by the detailed responses from the authors that are backed up by additional experimental results, which led to the consensus among the reviewers to accept the paper.

Given the importance of the benchmark focused on the robustness of the NAS, and difficulty of building such a dataset for each individual researcher due to the amount of computation necessary, as well as the effectiveness of the proposed multi-objective NTK-based NAS method, the paper will be of interest to a large number of audiences in the NAS community.

**Justification For Why Not Higher Score:**

Despite the practical impact the benchmark would bring when it becomes accessible to the NAS community, the findings are not quite novel (e.g. correlation between clean and robust accuracy) or minor. The multi-objective NTK-based NAS approach also is more of an incremental extension of an existing approach.

**Justification For Why Not Lower Score:**

Introducing a robustness-centered benchmark would benefit the researchers from the NAS community and further promote research into that direction.

---

### Decision · Program_Chairs · 2024-01-16

Accept (poster)